# Optimal Guarantees for Algorithmic Reproducibility and Gradient Complexity in Convex Optimization

**Liang Zhang**\*
ETH Zurich & Max Planck Institute
liang.zhang@inf.ethz.ch

**Junchi Yang**\*
ETH Zurich
junchi.yang@inf.ethz.ch

**Amin Karbasi**
Yale University & Google Research
amin.karbasi@yale.edu

**Niao He**
ETH Zurich
niao.he@inf.ethz.ch

## Abstract

Algorithmic reproducibility measures the deviation in outputs of machine learning algorithms upon minor changes in the training process. Previous work suggests that first-order methods would need to trade-off convergence rate (gradient complexity) for better reproducibility. In this work, we challenge this perception and demonstrate that both optimal reproducibility and near-optimal convergence guarantees can be achieved for smooth convex minimization and smooth convex-concave minimax problems under various error-prone oracle settings. Particularly, given the inexact initialization oracle, our regularization-based algorithms achieve the best of both worlds – optimal reproducibility and near-optimal gradient complexity – for minimization and minimax optimization. With the inexact gradient oracle, the near-optimal guarantees also hold for minimax optimization. Additionally, with the stochastic gradient oracle, we show that stochastic gradient descent ascent is optimal in terms of both reproducibility and gradient complexity. We believe our results contribute to an enhanced understanding of the reproducibility-convergence trade-off in the context of convex optimization.

## 1 Introduction

In the realm of machine learning, improving model performance remains a primary focus; however, this alone falls short when it comes to the practical deployment of algorithms. There has been a growing emphasis on the development of machine learning systems that prioritize trustworthiness and reliability. Central to this pursuit is the concept of reproducibility [38, 64], which requires algorithms to yield consistent outputs, in the face of minor changes to the training environment. Unfortunately, a lack of reproducibility has been reported across various domains [10, 40, 41, 64], posing significant challenges to the integrity and dependability of scientific research. Notably, empirical studies in Henderson et al. [43] have revealed that reproducing baseline algorithms in reinforcement learning is a formidable task due to both inherent sources (e.g., random seeds, environment properties) and external sources (e.g., hyperparameters, codebases) of non-determinism. These findings underscore the criticality of having access to the relevant code and data, as well as sufficient documentation of experimental details, to ensure reproducibility in machine learning algorithms.

Instead of considering the irreproducibility issue solely from an empirical perspective, Ahn et al. [1] initiated the theoretical study of reproducibility in machine learning as an inherent characteristic of the algorithms themselves. They focus on first-order algorithms for convex minimization problems and

---

\*Equal Contribution.

37th Conference on Neural Information Processing Systems (NeurIPS 2023).

Table 1: Algorithmic reproducibility (Def. 3) and gradient complexity for algorithms in the smooth convex minimization setting given inexact deterministic oracles (Def. 1). Here, "LB" stands for lower-bound and $a \wedge b$ denotes $\min\{a, b\}$. For the inexact gradient oracle, $\delta \leq \mathcal{O}(\epsilon)$ is required for GD to be $\epsilon$-optimal and $\delta \leq \mathcal{O}(\epsilon^{5/4})$ is required for Algo. 1.

| Algorithm | Inexact Initialization | | Inexact Gradient | |
| --- | --- | --- | --- | --- |
| | Convergence | Reproducibility | Convergence | Reproducibility |
| GD [1] | $\mathcal{O}(1/\epsilon)$ | $\mathcal{O}(\delta^2)$ | $\mathcal{O}(1/\epsilon)$ | $\mathcal{O}(\delta^2/\epsilon^2)$ |
| AGD [6] | $\mathcal{O}(1/\sqrt{\epsilon})$ | $\mathcal{O}(\delta^2 e^{1/\sqrt{\epsilon}})$ | - | - |
| Algo. 1 (Thm. 3.3, 3.5) | $\tilde{\mathcal{O}}(1/\sqrt{\epsilon})$ | $\mathcal{O}(\delta^2)$ | $\tilde{\mathcal{O}}(1/\sqrt{\epsilon})$ | $\mathcal{O}(\delta^2/\epsilon^{2.5})$ |
| LB [61, 1] | $\Omega(1/\sqrt{\epsilon})$ | $\Omega(\delta^2)$ | $\Omega(1/\sqrt{\epsilon})$ | $\Omega(\delta^2/\epsilon^2)$ |

define reproducibility as the deviation in outputs of independent runs of the algorithms, accounting for sources of irreproducibility captured by inexact or noisy oracles. In particular, they consider three practical error-prone operations, including inexact initialization, inexact gradient computation due to numerical errors, and stochastic gradient computation due to sampling or shuffling. When restricting the outputs to be $\epsilon$-optimal and assuming the level of inexactness that could cause irreproducibility is bounded by $\delta$, they establish both lower and upper reproducibility bounds of (stochastic) gradient descent for all three settings. The lower-bounds indicate the existence of intrinsic irreproducibility for any first-order algorithms, while the matching upper-bounds suggest that (stochastic) gradient descent already achieves optimal reproducibility.

An important question arises regarding whether there is a fundamental trade-off between reproducibility and convergence speed in algorithms. For example, in the case of inexact initialization, the optimally reproducible algorithm [1], gradient descent (GD), is known to be strictly sub-optimal in terms of gradient complexity for smooth convex minimization problems [61]. On the other hand, the optimally convergent algorithm, Nesterov's accelerated gradient descent (AGD) [60], suffers from a worse reproducibility bound [6]. The situation becomes more intricate in the case of inexact gradient computation. A natural question that we aim to address in this paper is: *Can we achieve the best of both worlds – optimal convergence and reproducibility?*

On another front, while minimization problems can effectively model and explain the behavior of many traditional machine learning systems, recent years have witnessed a surge of applications that are formulated as minimax optimization problems. Important examples include generative adversarial networks (GANs) [37], robust optimization [54], and reinforcement learning [25]. Despite a wealth of convergence theory for various minimax optimization algorithms, extensive empirical evidence suggests that these algorithms can be hard to train in practice [67, 4, 53]: the training procedure can be very unstable [23] and highly sensitive to changes of hyper-parameters. Motivated by such issues, we initiate the theoretical study of algorithmic reproducibility in minimax optimization. The second question that we aim to address in this paper is: *What are the fundamental limits of reproducibility for minimax optimization algorithms and their convergence-reproducibility trade-offs?* We will focus on smooth convex-concave minimax optimization as a first step, where the irreproducibility issue comes from either inexact initialization, inexact gradient computation, or stochastic gradient computation.

## 1.1 Our Contributions

Our main contributions are two-fold:

First, we propose Algorithm 1, which solves a regularized version of the smooth convex minimization problem. This algorithm achieves both optimal algorithmic reproducibility of $\mathcal{O}(\delta^2)$ and near-optimal gradient complexity of $\tilde{\mathcal{O}}(1/\sqrt{\epsilon})^2$ under the $\delta$-inexact initialization oracle. Table 1 provides a comparison with GD and AGD. Our results rely on the key observation that solutions to strongly-

---

[2]Throughout the paper, $\tilde{\mathcal{O}}$ hides additional logarithmic factors. We claim near-optimality of the result when it is optimal up to logarithmic terms.

Table 2: Algorithmic reproducibility (Def. 6) and gradient complexity for algorithms in the smooth convex-concave minimax setting given inexact deterministic oracles (Def. 4). Here, "LB" stands for lower-bound and $a \wedge b$ denotes $\min\{a, b\}$. For the inexact gradient oracle, $\delta \leq \mathcal{O}(\epsilon)$ is required for GDA, EG, and Algo. 3 to be $\epsilon$-optimal, and $\delta \leq \mathcal{O}(\epsilon^2)$ is required for Algo. 2. The diameter $D$ in Assumption 4.1 is a trivial upper-bound for reproducibility in all cases.

| Algorithm | Inexact Initialization | | Inexact Gradient | |
| --- | --- | --- | --- | --- |
| | Convergence | Reproducibility | Convergence | Reproducibility |
| GDA (Thm. 4.2) | $\mathcal{O}(1/\epsilon^2)$ | $\mathcal{O}(\delta^2)$ | $\mathcal{O}(1/\epsilon^2)$ | $\mathcal{O}(\delta^2/\epsilon^2)$ |
| EG (Thm. 4.3) | $\mathcal{O}(1/\epsilon)$ | $\mathcal{O}(\delta^2 e^{1/\epsilon} \wedge (\delta^2 + 1/\epsilon^2))$ | $\mathcal{O}(1/\epsilon)$ | $\mathcal{O}(\delta^2 e^{1/\epsilon} \wedge 1/\epsilon^2)$ |
| Algo. 2 (Thm. 4.4, 4.6) | $\tilde{\mathcal{O}}(1/\epsilon)$ | $\mathcal{O}(\delta^2)$ | $\tilde{\mathcal{O}}(1/\epsilon)$ | $\mathcal{O}(\delta^2/\epsilon^2)$ |
| Algo. 3 (Thm. 4.7, 4.8) | $\tilde{\mathcal{O}}(1/\epsilon)$ | $\mathcal{O}(\delta^2)$ | $\tilde{\mathcal{O}}(1/\epsilon)$ | $\mathcal{O}(\delta^2/\epsilon^2)$ |
| LB ([63], Lem. B.3) | $\Omega(1/\epsilon)$ | $\Omega(\delta^2)$ | $\Omega(1/\epsilon)$ | $\Omega(\delta^2/\epsilon^2)$ |

convex regularized problems are unique, allowing algorithms that converge close to the minimizers to be reproducible. This highlights the effectiveness of regularization in achieving near-optimal convergence without compromising reproducibility.

Second, we extend the notion of reproducibility to smooth convex-concave minimax optimization (1) under inexact initialization and inexact gradient oracles. We establish the first reproducibility analysis for commonly-used minimax optimization algorithms such as gradient descent ascent (GDA) and Extragradient (EG) [48]. Our results indicate that they are either sub-optimal in terms of convergence or reproducibility. To address this, we propose two new algorithms (Algorithm 2 and 3) which utilize regularization techniques to achieve optimal algorithmic reproducibility and near-optimal gradient complexity. The summarized results are presented in Table 2. Additional numerical experiments showcasing the effectiveness of our algorithms can be found in Appendix D. Although smooth convex-concave minimax optimization is nonsmooth in its primal form, our results indicate an improved reproducibility compared to the result of general nonsmooth convex problems [1] by leveraging the additional minimax structure. Lastly, in the case of stochastic gradient oracle, we show stochastic GDA can simultaneously attain both optimal convergence and optimal reproducibility.

## 1.2 Related Works

**Related Notions.** *(Reproducibility)* Previous works that study reproducibility in machine learning are mostly on the empirical side. They either conduct experiments to report irreproducibility issues in the community [40, 43, 18, 64], or propose practical tricks to improve reproducibility [69, 79, 56, 19]. Ahn et al. [1] initiated the theoretical study of reproducibility in convex minimization problems as a property of the algorithm itself. *(Replicability)* In an independent work, Impagliazzo et al. [45] proposed the notion of replicability in statistical learning, where an algorithm is replicable if its outputs on two i.i.d. datasets are exactly the same with high probability. Its connection to generalization and differential privacy [29] is established in Bun et al. [21] and Kalavasis et al. [47]. Replicable algorithms are proposed in the context of stochastic bandits [30] and clustering [31]. *(Stability)* Depending on the context, the term stability may have different meanings. In empirical studies [4, 5, 22], instability often refers to issues such as oscillations or failure to converge during training. In learning theory, algorithmic stability [17] measures the deviation in an algorithm's outputs for finite-sum problems when a single item in the input dataset is replaced by an i.i.d. in-distribution sample. The concept receives increasing attention as it implies dimension-independent generalization bounds of gradient-based methods for both minimization [42, 11, 6] and minimax [33, 49, 16] problems. In the area of differential equations [13] and variational inequalities [32], stability is also examined as a property of the solution set in response to perturbations in the problem conditions.

In this work, we consider the notion of reproducibility that characterizes the behavior of algorithms upon slight perturbations in the training. We defer the task of establishing intrinsic connections

among related notions to future work. The most closely related concept is algorithmic stability, where the analysis is similar to reproducibility under the inexact deterministic gradient oracle. Attia and Koren [6] showed the stability of AGD [60] grows exponentially with the number of iterations. Later, this is improved to quadratic dependence [7] based on a similar idea as ours that leverages stability of solutions to strongly-convex minimization problems [68, 34]. However, since there is no inexactness of the gradients in their setting, it is possible to ensure outputs that are arbitrarily close to the optimal solution. Given the presence of inexact gradients in our case, the convergence is only limited to a neighborhood of the optimal solution, which makes the problem more challenging. The trade-off between stability and convergence was investigated in Chen et al. [24]. Their results suggest that a faster algorithm has to be less stable, and vice versa. However, we show the feasibility of achieving both optimal reproducibility and near-optimal convergence simultaneously in the setting we considered.

**Minimax Optimization.** Existing literature on minimax optimization primarily focuses on convergence analysis across various settings. For instance, there are studies on the strongly-convex–strongly-concave case [72, 57], convex-concave case [59, 63], and nonconvex–(strongly)-concave case [52, 71]. The lower complexity bounds have also been established for these settings [80, 50, 77]. Our work aims to design reproducible algorithms while maintaining the optimal oracle complexities achieved in these previous works.

**Inexact Gradient Oracles.** A series of works investigate the convergence properties of first-order methods under deterministic inexact oracles for minimization [26, 27, 28] and minimax [70] problems. However, their inexact oracles differ from ours, and our focus is more on reproducibility. In recent years, there has been increasing interest in studying biased stochastic gradient oracles as well, where the bias arises from various sources such as problem structure [44], compression [14] or Byzantine failure [15] in distributed learning, and gradient-free optimization [62]. These biases can also contribute to irreproducibility, and this direction would be an interesting avenue of research.

**Regularization Technique.** The central algorithmic insight driving our improvements towards obtaining both optimal convergence and reproducibility is the regularization technique, which is commonly used in the optimization literature. One important use case is to boost convergence by leveraging known and good convergence properties of algorithms on smooth strongly-convex functions for solving convex and nonsmooth problems, see e.g., [51, 3, 74], just to name a few. In addition, the regularization technique has also been demonstrated to be useful in improving stability and generalization [76, 7], enhancing sensitivity and privacy guarantees [34, 78], etc. In this paper, we provide another important use case by showing an improved convergence-reproducibility trade-off.

## 2 Preliminaries in Algorithmic Reproducibility

**Notation.** We use $\|\cdot\|$ to represent the Euclidean norm. $\Pi_{\mathcal{C}}(x)$ denotes the projection of $x$ onto the set $\mathcal{C}$. A function $h : S \to \mathbb{R}$ is $\ell$-smooth if it is differentiable and its gradient $\nabla h$ satisfies $\|\nabla h(x_1) - \nabla h(x_2)\| \leq \ell \|x_1 - x_2\|$ for any $x_1, x_2$ in the domain $S \in \mathbb{R}^d$. A function $g : S \to \mathbb{R}$ is convex if $g(\alpha x_1 + (1 - \alpha)x_2) \leq \alpha g(x_1) + (1 - \alpha)g(x_2)$ for any $\alpha \in [0, 1]$ and $x_1, x_2 \in S$. If $g$ satisfies $g(x) - (\mu/2)\|x\|^2$ being convex with $\mu > 0$, then it is $\mu$-strongly-convex. Similarly, a function $g : S \to \mathbb{R}$ is concave if $-g$ is convex, and $\mu$-strongly-concave if $-g$ is $\mu$-strongly-convex.

Ahn et al. [1] studied the algorithmic reproducibility for convex minimization problems $\min_{x \in \mathcal{X}} F(x)$, measured by the $(\epsilon, \delta)$-deviation bound of an algorithm $\mathcal{A}$. Here, $\delta$ denotes the size of errors in the oracles that can lead to different outputs in independent runs of the same algorithm. The notion of reproducibility also requires $\mathcal{A}$ to produce $\epsilon$-optimal solutions, avoiding trivial outputs.

**Definition 1.** Three different inexact oracle models are considered: $(i)$ a $\delta$-*inexact initialization oracle* that returns a starting point $x_0 \in \mathcal{X}$ such that $\|x_0 - u_0\|^2 \leq \delta^2/4$ for some reference point $u_0 \in \mathcal{X}$, $(ii)$ a $\delta$-*inexact deterministic gradient oracle* that returns an inexact gradient $G(x)$ such that $\|\nabla F(x) - G(x)\|^2 \leq \delta^2$ for the true gradient $\nabla F(x)$, $(iii)$ a $\delta$-*inexact stochastic gradient oracle* that returns an unbiased gradient estimate $\nabla f(x; \xi)$ such that $\mathbb{E}\|\nabla f(x; \xi) - \nabla F(x)\|^2 \leq \delta^2$.

**Definition 2.** A point $\hat{x} \in \mathcal{X}$ is an $\epsilon$-optimal solution if $F(\hat{x}) - \min_{x \in \mathcal{X}} F(x) \leq \epsilon$ in the deterministic setting, or $\mathbb{E}[F(\hat{x})] - \min_{x \in \mathcal{X}} F(x) \leq \epsilon$ in the stochastic setting, where the expectation is taken over all the randomness in the gradient oracle and in the algorithm that outputs $\hat{x}$.

**Definition 3.** The $(\epsilon, \delta)$-deviation $\|\hat{x} - \hat{x}'\|^2$ is used to measure the reproducibility of an algorithm $\mathcal{A}$ with $\epsilon$-optimal solutions $\hat{x}$ and $\hat{x}'$, where $\hat{x}$ and $\hat{x}'$ are outputs of two independent runs of the algorithm $\mathcal{A}$ given a $\delta$-inexact oracle in Definition 1.

We expand the definitions of reproducibility to encompass minimax optimization problems:

$$\min_{x \in \mathcal{X}} \max_{y \in \mathcal{Y}} F(x, y). \tag{1}$$

Our goal is to find the *saddle point* $(x^*, y^*)$ of the function $F(x, y)$, such that $F(x^*, y) \leq F(x^*, y^*) \leq F(x, y^*)$ holds for all $(x, y) \in \mathcal{X} \times \mathcal{Y}$. The optimality of a point $(\hat{x}, \hat{y})$ can be assessed by its *duality gap*, defined as $\max_{y \in \mathcal{Y}} F(\hat{x}, y) - \min_{x \in \mathcal{X}} F(x, \hat{y})$. In the minimax setting, we analyze reproducibility under the following inexact oracle models.

**Definition 4.** Three different inexact oracle models are considered: $(i)$ a $\delta$-*inexact initialization oracle* that returns a starting point $(x_0, y_0) \in \mathcal{X} \times \mathcal{Y}$ such that $\|x_0 - u_0\|^2 + \|y_0 - v_0\|^2 \leq \delta^2/4$ for some reference point $(u_0, v_0) \in \mathcal{X} \times \mathcal{Y}$, $(ii)$ a $\delta$-*inexact deterministic gradient oracle* that returns an inexact gradient $G(x, y) = (G_x(x, y), G_y(x, y))$ at any querying point $(x, y) \in \mathcal{X} \times \mathcal{Y}$ such that $\|\nabla F(x, y) - G(x, y)\|^2 \leq \delta^2$ for the true gradient $\nabla F(x, y) = (\nabla_x F(x, y), \nabla_y F(x, y))$, $(iii)$ a $\delta$-*inexact stochastic gradient oracle* that returns an unbiased gradient estimate $\nabla f(x, y; \xi) = (\nabla_x f(x, y; \xi), \nabla_y f(x, y; \xi))$ such that $\mathbb{E}_\xi \|\nabla f(x, y; \xi) - \nabla F(x, y)\|^2 \leq \delta^2$.

**Definition 5.** A point $(\hat{x}, \hat{y}) \in \mathcal{X} \times \mathcal{Y}$ is an $\epsilon$-saddle point solution if its duality gap satisfies that $\max_{y \in \mathcal{Y}} F(\hat{x}, y) - \min_{x \in \mathcal{X}} F(x, \hat{y}) \leq \epsilon$ in the deterministic setting, or its *weak* duality gap satisfies that $\max_{y \in \mathcal{Y}} \mathbb{E}[F(\hat{x}, y)] - \min_{x \in \mathcal{X}} \mathbb{E}[F(x, \hat{y})] \leq \epsilon$ in the stochastic setting.

**Definition 6.** The $(\epsilon, \delta)$-deviation $\|\hat{x} - \hat{x}'\|^2 + \|\hat{y} - \hat{y}'\|^2$ is used to measure the reproducibility of an algorithm $\mathcal{A}$ with $\epsilon$-saddle points $(\hat{x}, \hat{y})$ and $(\hat{x}', \hat{y}')$, where $(\hat{x}, \hat{y})$ and $(\hat{x}', \hat{y}')$ are outputs of two independent runs of the algorithm $\mathcal{A}$ given a $\delta$-inexact oracle in Definition 4.

The optimal convergence rates are well-understood for the convex optimization problems, including convex minimization [61] and convex-concave minimax optimization [63]. Ahn et al. [1] provided the theoretical lower-bounds of reproducibility for convex minimization problems, which can be extended to convex-concave minimax problems as well (Lemma B.3). We say an algorithm achieves optimal reproducibility if its reproducibility upper-bounds match the established theoretical lower-bounds.

## 3 Deterministic Gradient Oracle for Minimization Problems

In this section, we consider convex minimization problems of the form

$$\min_{x \in \mathcal{X}} F(x),$$

where $\mathcal{X}$ is a convex and closed set. We focus on the standard smooth and convex setting as detailed in Assumption 3.1. Our goal is to find an $\epsilon$-optimal point as in Definition 2. Ahn et al. [1] showed that the optimal convergence rate and reproducibility can be achieved at the same time using stochastic gradient descent (SGD) for the stochastic gradient oracle model. In the deterministic case, they showed GD achieves the optimal reproducibility, albeit with a sub-optimal convergence rate [60, 61]. Considering the instability of accelerated gradient descent (AGD) [26, 28, 6], Ahn et al. [1] conjectured that $\Omega(1/\epsilon)$ gradient complexity is necessary to attain the optimal reproducibility.

**Assumption 3.1.** *The function $F$ is convex and $\ell$-smooth. We have access to initial points $x_0$ that are $D$-close to an optimal solution, i.e., $\|x^* - x_0\|^2 \leq D^2$ for some $x^* \in \arg\min_{x \in \mathcal{X}} F(x)$.*

We introduce a generic algorithmic framework outlined in Algorithm 1, that solves a quadratically regularized auxiliary problem $(\star)$ using a base algorithm $\mathcal{A}$ with initialization $x_0$ until an accuracy of $\epsilon_r$ is reached. Our key insight is that since the optimal solution for strongly convex problems is unique, the reproducibility of the outputs from the regularized problem can be easily guaranteed. Note that the regularization parameter $r$ presents a trade-off: as $r$ increases, the auxiliary problem can be solved more efficiently, but the obtained solution deviates further from the original solution. We will show that Algorithm 1 achieves a near-optimal complexity of $\tilde{\mathcal{O}}(1/\sqrt{\epsilon})$, along with optimal reproducibility under an inexact initialization oracle and slightly sub-optimal reproducibility under an inexact deterministic gradient oracle. This finding disproves the conjecture [1] that $\Omega(1/\epsilon)$ complexity is necessary to achieve optimal reproducibility.

**Algorithm 1** Reproducible Algorithmic Framework for Convex Minimization Problems

**Input:** Regularization parameter $r > 0$, accuracy $\epsilon_r > 0$, base algorithm $\mathcal{A}$, initial point $x_0 \in \mathcal{X}$.

Apply $\mathcal{A}$ to approximately solve the $r$-strongly-convex and $(\ell + r)$-smooth problem

$$x_r \leftarrow \arg\min_{x \in \mathcal{X}} F_r(x) := F(x) + \frac{r}{2}\|x - x_0\|^2, \qquad (\star)$$

such that the optimality gap

$$F_r(x_r) - \min_{x \in \mathcal{X}} F_r(x) \leq \epsilon_r.$$

**Output:** $x_r$.

## 3.1 Inexact Initialization Oracle

We first examine the behavior of Algorithm 1 with access to exact deterministic gradients but given different initializations. Starting from two distinct initial points $x_0$ and $x_0'$ such that $\|x_0 - x_0'\|^2 \leq \delta^2$, we want to control the deviation between the final outputs $x_r$ and $x_r'$ of the algorithm. The following contraction property is essential to attain optimal reproducibility.

**Lemma 3.2.** *Let* $x_r^* = \arg\min_{x \in \mathcal{X}}\{F(x) + (r/2)\|x - x_0\|^2\}$ *and* $(x_r^*)' = \arg\min_{x \in \mathcal{X}}\{F(x) + (r/2)\|x - x_0'\|^2\}$. *When $F$ is convex, it holds that $\|x_r^* - (x_r^*)'\|^2 \leq \|x_0 - x_0'\|^2$ for any $r > 0$.*

This indicates the optimal solutions are reproducible up to $\delta^2$. Consequently, if we can solve the auxiliary problem $(\star)$ to a high accuracy $\epsilon_r$, we can ensure the final output $x_r$ is reproducible. The selection of $\epsilon_r$ exhibits a trade-off: a smaller value increases complexity, yet brings the output closer to the reproducible $x_r^*$. We characterize the complexity and reproducibility of Algorithm 1 by carefully choosing the parameters $r$ and $\epsilon_r$.

**Theorem 3.3.** *Under Assumption 3.1 and given an inexact initialization oracle, Algorithm 1 with* $r = \epsilon/D^2$, $\epsilon_r = (\epsilon/2)\min\{1, \delta^2/(4D^2)\}$ *and AGD [61] as base algorithm $\mathcal{A}$ outputs an $\epsilon$-optimal point $x_r$ with $\tilde{\mathcal{O}}(\sqrt{\ell D^2/\epsilon})$ gradient complexity, and the reproducibility is $\|x_r - x_r'\|^2 \leq 4\delta^2$.*

This theorem implies that we can simultaneously achieve the near-optimal complexity of $\tilde{\mathcal{O}}(\sqrt{\ell D^2/\epsilon})$ and optimal reproducibility of $\mathcal{O}(\delta^2)$, which improves over the $\mathcal{O}(\ell D^2/\epsilon)$ complexity of GD [1]. In fact, when combined with any base algorithm that solves the auxiliary problem, Algorithm 1 attains optimal reproducibility. However, using AGD as the base algorithm results in the best complexity. To the best of our knowledge, this is the only algorithm capable of achieving the best of both worlds. Previously, Attia and Koren [6] proved that the algorithmic reproducibility (referred to as initialization stability in their study) of Nesterov's AGD is $\Theta(\delta^2 e^{1/\sqrt{\epsilon}})$ when the initialization is $\delta^2$-apart.

**Remark 1.** Adding regularization is a common and useful technique in the optimization literature. Our algorithmic framework solves one auxiliary regularized strongly-convex problem, which is referred to as classical regularization reduction in Allen-Zhu and Hazan [3]. Algorithm 1 is biased and requires the knowledge of $\epsilon$ and $D$ to control the biased term introduced by the regularization term. The convergence guarantee also has an additional sub-optimal logarithmic term. Allen-Zhu and Hazan [3] proposed to use a double-loop algorithm, where a sequence of auxiliary regularized strongly-convex problems with decreasing regularization parameters are solved. The vanishing regularization ensures the algorithm is unbiased, and the resulting convergence guarantee requires no knowledge of $\epsilon$ and does not have an additional logarithmic term. Similar idea could apply to our case as well, and the task of bridging such gaps is deferred to future work.

## 3.2 Inexact Deterministic Gradient Oracle

We further study the algorithmic reproducibility and gradient complexity of Algorithm 1 under the inexact gradient oracle model that returns an inexact gradient $G(x) \in \mathbb{R}^d$ such that $\|G(x) - \nabla F(x)\|^2 \leq \delta^2$ at any query point $x \in \mathcal{X}$. From the inexact gradient oracle of $F$, we can construct an inexact gradient oracle for the auxiliary problem $F_r$: $G_r(x) = G(x) + r(x - x_0)$ which satisfies the condition $\|G_r(x) - \nabla F_r(x)\|^2 = \|G(x) - \nabla F(x)\|^2 \leq \delta^2$. To solve the auxiliary problem, we consider AGD with an inexact oracle (Inexact-AGD) as proposed by Devolder et al. [27]. The proposition below establishes its convergence behavior.

**Proposition 3.4.** *Consider* $\min_{x \in \mathcal{X}} F_r(x)$, *where* $F_r$ *is* $r$-*strongly-convex and* $(\ell + r)$-*smooth. Given an inexact gradient oracle that returns* $G_r(x)$ *such that* $\|G_r(x) - \nabla F_r(x)\|^2 \leq \delta^2$, *starting from* $y_0 = x_0$, *AGD with the following update rule*

$$
x_{t+1} = \Pi_{\mathcal{X}}\left(y_t - \frac{1}{2(\ell + r)} G_r(y_t)\right),
$$

$$
y_{t+1} = x_{t+1} + \frac{2 - \sqrt{r/(\ell + r)}}{2 + \sqrt{r/(\ell + r)}}(x_{t+1} - x_t), \qquad \text{(Inexact-AGD)}
$$

*for* $t = 0, 1, \cdots, T - 1$, *satisfies that*

$$
F_r(x_T) - F_r(x_r^*) \leq \exp\left(-\frac{T}{2}\sqrt{\frac{r}{2\ell}}\right)\left(F_r(x_0) - F_r(x_r^*) + \frac{r}{4}\|x_0 - x_r^*\|^2\right) + \sqrt{\frac{2\ell}{r}}\left(\frac{1}{\ell + r} + \frac{2}{r}\right)\delta^2,
$$

*where* $x_r^*$ *is the unique minimizer of* $F_r(x)$.

This proposition suggests that Inexact-AGD converges to a neighborhood with a radius of $\mathcal{O}(\delta^2/r^{3/2})$ around the optimal value. We note that convergence to the exact solution is unattainable for algorithms employing inexact gradients [27, 28], and the size of this neighborhood is important in determining the reproducibility of $x_r$.

**Theorem 3.5.** *Under Assumption 3.1 with* $0 < \epsilon \leq \ell D^2$ *and given an inexact deterministic gradient oracle in Definition 1, Algorithm 1 with* $r = \epsilon/D^2$, $\epsilon_r = 6\delta^2 D^3\sqrt{\ell/(2\epsilon^3)}$ *and Inexact-AGD as base algorithm outputs a* $(6\delta^2 D^3\sqrt{\ell/(2\epsilon^3)} + \epsilon/2)$-*optimal point* $x_r$ *with* $\tilde{\mathcal{O}}(\sqrt{\ell D^2/\epsilon})$ *gradient complexity, and the reproducibility is* $\|x_r - x_r'\|^2 \leq \mathcal{O}(\delta^2/\epsilon^{5/2})$.

Ahn et al. [1] showed that GD achieves optimal reproducibility of $\mathcal{O}(\delta^2/\epsilon^2)$ and a complexity of $\mathcal{O}(1/\epsilon)$ when $\delta \leq \mathcal{O}(\epsilon)$. Our results indicate that a reproducibility of $\mathcal{O}(\delta^2/\epsilon^{5/2})$ and a near-optimal complexity of $\tilde{\mathcal{O}}(1/\sqrt{\epsilon})$ can be attained when $\delta \leq \mathcal{O}(\epsilon^{5/4})$. We conjecture that this suboptimal reproducibility bound is inevitable for the proposed framework given the lower bound result in Devolder et al. [27] for algorithms under a $(\delta, \ell, \mu)$-inexact oracle associated with $\ell$-smooth $\mu$-strongly-convex functions. Further discussions are provided in Appendix A.2. Moreover, we point out that for minimizing $\ell$-smooth and $\mu$-strongly-convex functions, Proposition 3.4 already implies that Inexact-AGD attains the optimal reproducibility of $\mathcal{O}(\min\{\delta^2, \epsilon\})$ and the optimal complexity of $\tilde{\mathcal{O}}(\sqrt{\ell/\mu})$ when the problem is well-conditioned, improving over the $\tilde{\mathcal{O}}(\ell/\mu)$ complexity in the previous work [1].

**Remark 2.** In Appendix D, we demonstrate the effectiveness of Algorithm 1 on a quadratic minimization problem equipped with an inexact gradient oracle. The results are plotted in Figure 1 in the appendix. We observe that the reproducibility can be greatly improved when adding regularization, with only a small degradation in the convergence performance.

## 4  Deterministic Gradient Oracle for Minimax Problems

In this section, we address the minimax optimization problem of the form

$$
\min_{x \in \mathcal{X}} \max_{y \in \mathcal{Y}} F(x, y),
$$

where $\mathcal{X}$ and $\mathcal{Y}$ are convex compact sets. We focus on the standard smooth and convex-concave setting as detailed in Assumption 4.1. We aim to find an $\epsilon$-saddle point $(\hat{x}, \hat{y})$ such that its duality gap satisfies $\max_{y \in \mathcal{Y}} F(\hat{x}, y) - \min_{x \in \mathcal{X}} F(x, \hat{y}) \leq \epsilon$. Here, the assumption that the domains are convex and bounded ensures the existence of the saddle point when the objective is convex-concave [73]. We focus on minimax problems equipped with inexact initialization oracles and inexact deterministic gradient oracles as defined in Definition 4. We first show that two classical algorithms, gradient descent ascent (GDA) and Extragradient (EG) [48, 72], are either sub-optimal in convergence or sub-optimal in reproducibility, which mirrors the minimization setting. Based on the same regularization idea, we propose two new frameworks in Algorithm 2 and 3 that successfully attain near-optimal convergence and optimal reproducibility at the same time.

**Assumption 4.1.** *For all* $y \in \mathcal{Y}$, $F(\cdot, y)$ *is convex, and for all* $x \in \mathcal{X}$, $F(x, \cdot)$ *is concave. Furthermore,* $F$ *is* $\ell$-*smooth on the domain* $\mathcal{X} \times \mathcal{Y}$. *Additionally, both* $\mathcal{X}$ *and* $\mathcal{Y}$ *have a diameter of* $D$. *This means that* $\|x_1 - x_2\|^2 \leq D^2$ *and* $\|y_1 - y_2\|^2 \leq D^2$ *for all* $x_1, x_2 \in \mathcal{X}$ *and* $y_1, y_2 \in \mathcal{Y}$.

The optimal gradient complexity to find $\epsilon$-saddle point under such assumptions is $\Theta(1/\epsilon)$ [63]. Since the minimax problem reduces to a minimization problem on $\mathcal{X}$ when the domain $\mathcal{Y}$ is restricted to be a singleton, the reproducibility lower-bounds [1] for smooth convex minimization hold as lower-bounds for smooth convex-concave minimax optimization as well. That is, $\Omega(\delta^2)$ under the inexact initialization oracle, and $\Omega(\delta^2/\epsilon^2)$ under the inexact gradient oracle (see Lemma B.3). We now present the convergence rate and reproducibility bounds of GDA (see Algorithm 4) and EG (see Algorithm 5).

**Theorem 4.2.** (GDA) *Under Assumption 4.1, the average iterate $(\bar{x}_T, \bar{y}_T)$ output by GDA with stepsize $1/(\ell\sqrt{T})$ after $T = \mathcal{O}(1/\epsilon^2)$ iterations is an $\epsilon$-saddle point. Furthermore, the reproducibility of the output is* (i) $\mathcal{O}(\delta^2)$ *under $\delta$-inexact initialization oracle;* (ii) $\mathcal{O}(\delta^2/\epsilon^2)$ *under $\delta$-inexact deterministic gradient oracle if $\delta \leq \mathcal{O}(\epsilon)$.*

**Theorem 4.3.** (Extragradient) *Under Assumption 4.1, the average iterate $(\bar{x}_{T+1/2}, \bar{y}_{T+1/2})$ output by EG with stepsize $1/\ell$ after $T = \mathcal{O}(1/\epsilon)$ iterations is an $\epsilon$-saddle point. Furthermore, the reproducibility of this output is* (i) $\mathcal{O}(\min\{\delta^2 e^{1/\epsilon}, \delta^2 + 1/\epsilon^2, D^2\})$ *under $\delta$-inexact initialization oracle;* (ii) $\mathcal{O}(\min\{\delta^2 e^{1/\epsilon}, 1/\epsilon^2, D^2\})$ *under $\delta$-inexact deterministic gradient oracle if $\delta \leq \mathcal{O}(\epsilon)$.*

While GDA can achieve optimal reproducibility, it converges with a sub-optimal complexity of $\mathcal{O}(1/\epsilon^2)$. On the other hand, EG achieves an optimal $\mathcal{O}(1/\epsilon)$ complexity but is not optimally reproducible. Further details on this are provided in Appendix B. In Appendix B.3.4, we also demonstrate that EG, through an alternative parameter selection, can achieve optimal reproducibility at a sub-optimal rate $\mathcal{O}(1/\epsilon^{3/2})$. The question that remains open is how to simultaneously attain both optimal reproducibility and gradient complexity. To address this, we have developed two algorithmic frameworks with near-optimal guarantees, one based on regularization and the other based on proximal point methods [66, 12].

## 4.1 Regularization Helps!

---
**Algorithm 2** Reproducible Algorithmic Framework for Convex-Concave Minimax Problems

---
**Input:** Regularization parameter $r > 0$, accuracy $\epsilon_r > 0$, base algorithm $\mathcal{A}$, initialization $(x_0, y_0)$.
  Apply $\mathcal{A}$ to inexactly solve the $r$-strongly-convex-strongly-concave and $(\ell + r)$-smooth problem

$$(x_r, y_r) \leftarrow \min_{x \in \mathcal{X}} \max_{y \in \mathcal{Y}} F_r(x,y) := F(x,y) + \frac{r}{2}\|x - x_0\|^2 - \frac{r}{2}\|y - y_0\|^2, \qquad (*)$$

such that $\forall (x,y) \in \mathcal{X} \times \mathcal{Y}$,

$$\nabla_x F_r(x_r, y_r)^\top (x_r - x) - \nabla_y F_r(x_r, y_r)^\top (y_r - y) \leq \epsilon_r. \qquad (2)$$

**Output:** $(x_r, y_r)$.

---

We demonstrate that adding regularization is sufficient to achieve near-optimal guarantees for smooth convex-concave minimax problems. The general framework is summarized in Algorithm 2, where a base algorithm $\mathcal{A}$ is applied to solve a regularized auxiliary problem which is strongly-convex in $x$ and strongly-concave in $y$. For the inexact initialization case, we show that an optimal reproducibility bound of $\mathcal{O}(\delta^2)$ and a near-optimal convergence rate of $\tilde{\mathcal{O}}(1/\epsilon)$ can be attained simultaneously.

**Theorem 4.4.** *Under Assumption 4.1 and given an inexact initialization oracle, Algorithm 2 with $r = \epsilon/D^2$, $\epsilon_r = \epsilon \cdot \min\{1, \delta^2/(8D^2)\}$ and EG as base algorithm $\mathcal{A}$ outputs a $(2\epsilon)$-saddle point $(x_r, y_r)$ with $\tilde{\mathcal{O}}(\ell D^2/\epsilon)$ gradient complexity, and the reproducibility is $4\delta^2$.*

Consider a $\delta$-inexact deterministic gradient oracle that returns $G(x,y) = (G_x(x,y), G_y(x,y))$. First note $G_r(x,y) = (G_x(x,y) + r(x - x_0), G_y(x,y) - r(y - y_0))$ is a $\delta$-inexact gradient for the auxiliary problem $(*)$. We now characterize the convergence behavior of EG with this $\delta$-inexact gradient oracle, referred to as Inexact-EG, to solve the auxiliary problem.

**Lemma 4.5.** *Consider $\min_{x \in \mathcal{X}} \max_{y \in \mathcal{Y}} F_r(x,y)$, where $F_r(x,y)$ is $r$-strongly-convex-strongly-concave and $(\ell + r)$-smooth. Given an inexact gradient oracle that returns $G_r(x,y)$ such that $\|G_r(x,y) - \nabla F_r(x,y)\|^2 \leq \delta^2$, Inexact-EG with stepsize $1/(2(\ell + r))$ satisfies*

$$\|x_T - x_r^*\|^2 + \|y_T - y_r^*\|^2 \leq \exp\left(-\frac{T}{8}\frac{r}{\ell + r}\right)\left(\|x_0 - x_r^*\|^2 + \|y_0 - y_r^*\|^2\right) + \frac{8\delta^2}{r}\left(\frac{2}{\ell + r} + \frac{1}{r}\right).$$

*where $(x_r^*, y_r^*)$ is the unique saddle point of $F_r(x, y)$.*

This lemma implies that Inexact-EG converges linearly to a neighborhood of size $\mathcal{O}(\delta^2/r^2)$ around the saddle point, which can be translated to the inaccuracy measure in (2) with $\epsilon_r = \mathcal{O}(\delta/r)$ utilizing Lemma C.5. It is worth emphasizing that the size of this neighborhood is critical for achieving optimal reproducibility, and the dependency on $r$ in the above convergence rate is key for attaining near-optimal complexity. Stonyakin et al. [70] analyzed Mirror-Prox [59] with restarts for strongly-monotone variational inequalities under a different inexact oracle (see Devolder et al. [27] and [70, Example 6.1] for its relationship with the inexactness notion of ours). Compared to Inexact-EG, their two-loop structure of the restart scheme is more complicated to implement.

**Theorem 4.6.** *Under Assumption 4.1 with $0 < \epsilon \leq \ell D^2$ and given an inexact gradient oracle, Algorithm 2 with $r = \epsilon/D^2$, $\epsilon_r = \mathcal{O}(\delta/r)$ and Inexact-EG as base algorithm $\mathcal{A}$ outputs an $\mathcal{O}(\epsilon+\delta/\epsilon)$-saddle point with $\tilde{\mathcal{O}}(\ell D^2/\epsilon)$ gradient complexity, and the reproducibility is $\mathcal{O}(\delta^2/\epsilon^2)$.*

**Remark 3.** Some numerical experiments on a bilinear matrix game with inexact gradient information are provided in Appendix D (see Figure 2). With a small degradation in the convergence speed, the regularized framework in Algorithm 2 effectively improves the reproducibility of the base algorithm.

The theorem indicates that optimal reproducibility $\mathcal{O}(\delta^2/\epsilon^2)$ and near-optimal gradient complexity $\tilde{\mathcal{O}}(1/\epsilon)$ can be achieved when $\delta \leq \mathcal{O}(\epsilon^2)$. Note by Theorem 4.2 and 4.3, GDA and EG can find $\epsilon$-saddle points when $\delta \leq \mathcal{O}(\epsilon)$. Next, we introduce an alternative algorithmic framework that preserves the optimal reproducibility and attains the near-optimal complexity as long as $\delta \leq \mathcal{O}(\epsilon)$.

### 4.2 Inexact Proximal Point Method

We propose a two-loop inexact proximal point framework, presented in Algorithm 3, which can achieve both near-optimal gradient complexity and optimal algorithmic reproducibility. Compared to Algorithm 2, the regularization parameter $1/\alpha = \mathcal{O}(\ell)$ does not depend on the target accuracy $\epsilon$ and the diameter $D$, and the center of the regularization term is the last iterate $(x_t, y_t)$ instead of the initial point. Since the auxiliary problem is $\ell$-strongly-convex-strongly-concave and $2\ell$-smooth with condition number being $\Theta(1)$, a wider range of base algorithms can be used to achieve the optimal complexity than solving the problem in Algorithm 2 where the condition number is $\Theta(1/\epsilon)$.

---

**Algorithm 3** Inexact Proximal Point Method for Convex-Concave Minimax Problems

---

**Input:** Stepsize $\alpha > 0$, accuracy $\hat{\epsilon} > 0$, algorithm $\mathcal{A}$, initialization $(x_0, y_0)$, iteration number $T$.
**for** $t = 0, 1, \cdots T - 1$ **do**
    Apply $\mathcal{A}$ to inexactly solve the smooth strongly-convex–strongly-concave problem

$$(x_{t+1}, y_{t+1}) \leftarrow \min_{x \in \mathcal{X}} \max_{y \in \mathcal{Y}} \hat{F}_t(x, y) := F(x, y) + \frac{1}{2\alpha}\|x - x_t\|^2 - \frac{1}{2\alpha}\|y - y_t\|^2.$$

    such that $\forall (x, y) \in \mathcal{X} \times \mathcal{Y}$,

$$\nabla_x \hat{F}_t(x_{t+1}, y_{t+1})^\top (x_{t+1} - x) - \nabla_y \hat{F}_t(x_{t+1}, y_{t+1})^\top (y_{t+1} - y) \leq \hat{\epsilon}.$$

**Output:** $(\bar{x}_{T+1}, \bar{y}_{T+1}) = (1/T) \sum_{t=0}^{T-1} (x_{t+1}, y_{t+1})$.

---

**Theorem 4.7.** *Under Assumption 4.1and given a $\delta$-inexact initialization oracle in Definition 4 with $\delta \leq \mathcal{O}(1/\sqrt{\epsilon})$, Algorithm 3 with $\hat{\epsilon} \leq \delta^2/(2\alpha T^2)$ and $\alpha = 1/\ell$ outputs an $\mathcal{O}(\epsilon)$-saddle point after $T = \mathcal{O}(1/\epsilon)$ iterations, and the reproducibility is $9\delta^2$.*

**Remark 4.** The required accuracy $\hat{\epsilon}$ for the auxiliary problem is $\mathcal{O}(\delta^2\epsilon^2)$. Given that the auxiliary problem is $\ell$-strongly-convex-strongly-concave and $2\ell$-smooth, various linearly convergent algorithms such as EG, GDA, and Optimistic GDA [35] can find a point that satisfies the stopping criterion within $\mathcal{O}(\log(1/(\delta\epsilon)))$ iterations. As a result, the total gradient complexity is $\tilde{\mathcal{O}}(1/\epsilon)$. In contrast, using GDA as the base algorithm in Algorithm 2 will lead to a sub-optimal gradient complexity.

**Theorem 4.8.** *Under Assumption 4.1 and given a $\delta$-inexact deterministic gradient oracle in Definition 4 with $\delta \leq \mathcal{O}(\epsilon)$, Algorithm 3 with $\hat{\epsilon} \leq \mathcal{O}(\delta)$ and $\alpha = 1/\ell$ outputs an $\mathcal{O}(\epsilon)$-saddle point after $T = \mathcal{O}(1/\epsilon)$ iterations, and the reproducibility is $\mathcal{O}(\delta^2/\epsilon^2)$.*

**Remark 5.** This theorem requires solving the auxiliary problem with a $\delta$-inexact gradient oracle. In addition to Inexact-EG presented in Lemma 4.5, we show in Appendix C.1 that GDA with inexact gradients (Inexact-GDA) can also converge linearly to the optimal point up to a $\mathcal{O}(\delta^2)$ error. Thus the total complexity is $\mathcal{O}((1/\epsilon)\log(1/\delta))$ using both Inexact-EG and Inexact-GDA.

## 5   Stochastic Gradient Oracle for Minimax Problems

To provide a complete picture, in this section, we consider the stochastic minimax problem:

$$\min_{x\in\mathcal{X}}\max_{y\in\mathcal{Y}} F(x,y) = \mathbb{E}_\xi[f(x,y;\xi)], \tag{3}$$

where the expectation is taken over a random vector $\xi$. We have access to a $\delta$-inexact stochastic gradient oracle that can return unbiased gradients $\nabla f(x,y;\xi)$ with a bounded variance $\delta^2$ at each point $(x,y)$. We consider the popular algorithm called stochastic gradient descent ascent (SGDA). The convergence behaviors of SGDA for the stochastic minimax problem (3) are well-known in various settings. However, due to the randomness in the gradient oracle, independent runs of SGDA may lead to different outputs even with the same parameters. Following Definition 6, we further establish the $(\epsilon,\delta)$-deviation of SGDA in the theorem below.

**Theorem 5.1.** *Under Assumptions 4.1 and given an inexact stochastic gradient oracle in Definition 4 with $\delta = \mathcal{O}(1)$, the average iterates $(\bar{x}_T,\bar{y}_T) = (1/T)\sum_{t=0}^{T-1}(x_t,y_t)$ of SGDA with stepsize $1/(\ell\epsilon T)$ after $T = \Omega(1/\epsilon^2)$ iterations is an $\mathcal{O}(\epsilon)$-stationary point and the reproducibility is $\mathcal{O}\big(\delta^2/(\epsilon^2 T)\big)$.*

The $\mathcal{O}(1/\epsilon^2)$ sample complexity of SGDA is known to be optimal when the objective $F(x,y)$ is convex-concave [46]. Moreover, our results suggest that SGDA is also optimally reproducible, as the lower-bound of $\Omega\big(\delta^2/(\epsilon^2 T)\big)$ for convex minimization problems [1] is also valid for minimax optimization according to our discussions in Lemma B.3.

## 6   Conclusion

In this work, instead of solely focusing on convergence performance, we investigate another crucial property of machine learning algorithms, i.e., algorithms should be reproducible against slight perturbations. We provide the first algorithms to simultaneously achieve optimal algorithmic reproducibility and near-optimal gradient complexity for both smooth convex minimization and smooth convex-concave minimax problems under various inexact oracle models. We focus on the convex case as a first step since it is the most basic and fundamental setting in optimization. We believe a solid understanding of the reproducibility in convex optimization will shed insights for that of the more challenging nonconvex optimization. Note that some of the analysis and techniques used in this paper can be extended to the smooth nonconvex setting, aligning with the stability analysis for nonconvex objectives [42, 49]. The proposed regularized framework can be applied to nonconvex functions as well using the convergence analysis of regularization or proximal point-based methods [2, 74]. However, the non-expansiveness property in Lemma 3.2 that is essential for the reproducibility analysis will not hold any more without the convexity assumption. One potential way to alleviate it is to impose additional structural assumptions on the gradients such as negative comonotonicity [39]. We leave a detailed study of the reproducibility in nonconvex optimization to future work.

Other possible improvements of our results include deriving optimal reproducibility with an accelerated convergence rate for smooth convex minimization problems under the inexact gradient oracle, removing the additional logarithmic terms in the complexity of our algorithms using techniques in Allen-Zhu and Hazan [3], studying the reproducibility under the presence of mixed inexact oracles, and extending the results to nonsmooth settings. Another interesting direction is to design simpler and more direct methods with both optimal reproducibility and convergence guarantees. A possible way is to directly unwrap the regularized algorithmic framework 1 or 2, leading to Tikhonov regularization [8] or anchoring methods [75].

## Acknowledgments and Disclosure of Funding

We thank the anonymous reviewers for their valuable suggestions. Liang Zhang gratefully acknowledges funding from the Max Planck ETH Center for Learning Systems (CLS). Amin Karbasi

acknowledges funding in direct support of this work from NSF (IIS-1845032), ONR (N00014-19-1-2406), and the AI Institute for Learning-Enabled Optimization at Scale (TILOS). Niao He is supported by ETH research grant, NCCR Automation, and Swiss National Science Foundation Project Funding No. 200021-207343.

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

# Contents

# A  Near-optimal Guarantees in the Minimization Case

This section provides proof for the near-optimal guarantees of Algorithm 1 in the minimization case. We start with some commonly-used facts that follow from basic algebraic calculations. See Bauschke et al. [12] for an example.

**Lemma A.1.** *The following facts will be used in the analysis. For any vectors $a, b \in \mathbb{R}^d$, it holds that*

$(i)\ 2a^\top b = \|a\|^2 + \|b\|^2 - \|a - b\|^2,$

$(ii)\ 2a^\top b = \|a + b\|^2 - \|a\|^2 - \|a\|^2,$

$(iii)\ -\gamma\|a\|^2 - \dfrac{1}{\gamma}\|b\|^2 \leq 2a^\top b \leq \gamma\|a\|^2 + \dfrac{1}{\gamma}\|b\|^2, \quad \forall \gamma > 0,$

$(iv)\ \|\eta a + (1 - \eta)b\|^2 + \eta(1 - \eta)\|a - b\|^2 = \eta\|a\|^2 + (1 - \eta)\|b\|^2, \quad \forall \eta \in \mathbb{R}.$

## A.1  Inexact Initialization Oracle

This section contains proof of Lemma 3.2 and Theorem 3.3 for the near-optimal guarantees of Algorithm 1 in the inexact initialization case.

*Proof of Lemma 3.2.* By the optimality conditions of $x_r^*$ and $(x_r^*)'$, we have that for any $x, x' \in \mathcal{X}$,

$$(\nabla F(x_r^*) + r(x_r^* - x_0))^\top (x - x_r^*) \geq 0,$$
$$(\nabla F((x_r^*)') + r((x_r^*)' - x_0'))^\top (x' - (x_r^*)') \geq 0.$$

Taking $x' = x_r^*$ and $x = (x_r^*)'$ in the above equation, we obtain that

$$(x_r^* - (x_r^*)')^\top \Big( (\nabla F(x_r^*) + r(x_r^* - x_0)) - (\nabla F((x_r^*)') + r((x_r^*)' - x_0')) \Big) \leq 0.$$

Since $\nabla F$ is monotone when $F$ is convex, rearranging terms, we get

$$0 \geq (x_r^* - (x_r^*)')^\top (\nabla F(x_r^*) - \nabla F((x_r^*)')) + r\|x_r^* - (x_r^*)'\|^2 - r(x_r^* - (x_r^*)')^\top (x_0 - x_0')$$
$$\geq r\|x_r^* - (x_r^*)'\|^2 - r(x_r^* - (x_r^*)')^\top (x_0 - x_0').$$

Given $r > 0$, this means

$$\|x_r^* - (x_r^*)'\|^2 \leq (x_r^* - (x_r^*)')^\top (x_0 - x_0')$$
$$\leq \|x_r^* - (x_r^*)'\|\|x_0 - x_0'\|.$$

Dividing both sides by $\|x_r^* - (x_r^*)'\|$, the proof is complete.    □

By converging sufficiently close to the optimal solution, we can ensure Algorithm 1 is reproducible. The near-optimal convergence rate is achieved using AGD [60] as the base algorithm.

*Proof of Theorem 3.3.* We first analyze the convergence guarantee. Let $x^* \in \arg\min_{x \in \mathcal{X}} F(x)$ be one minimizer of $F(x)$, and $x_r^* = \arg\min_{x \in \mathcal{X}} F_r(x)$ be the unique minimizer of $F_r(x)$. By the definition of $F_r(x)$, we have that

$$
\begin{aligned}
F(x_r) - F(x^*) &= F_r(x_r) - \frac{r}{2}\|x_r - x_0\|^2 - F_r(x^*) + \frac{r}{2}\|x^* - x_0\|^2 \\
&\leq F_r(x_r) - F_r(x^*) + \frac{r}{2}\|x^* - x_0\|^2 \\
&\leq F_r(x_r) - F_r(x_r^*) + \frac{r}{2}\|x^* - x_0\|^2 \\
&\leq \epsilon_r + \frac{rD^2}{2}.
\end{aligned}
\tag{4}
$$

$\epsilon_r$ and $r$ will be selected later. For reproducibility, we proceed as
$$\|x_r - x_r'\| \le \|x_r - x_r^*\| + \|x_r^* - (x_r^*)'\| + \|(x_r^*)' - x_r'\|$$
$$\le \delta + 2\sqrt{\frac{2\epsilon_r}{r}}.$$
where we use the optimality condition of $x_r^*$ by $r$-strong-convexity of $F_r(x)$:
$$\frac{r}{2}\|x_r - x_r^*\|^2 \le F_r(x_r) - F_r(x_r^*)$$
$$\le \epsilon_r,$$
the optimality condition of $(x_r^*)'$ and Lemma 3.2. Setting
$$r = \frac{\epsilon}{D^2}, \quad \epsilon_r = \frac{\epsilon}{2}\min\left\{1, \frac{\delta^2}{4D^2}\right\},$$
we guarantee that $F(x_r) - F(x^*) \le \epsilon$ and $\|x_r - x_r'\| \le 2\delta$. The gradient complexity of AGD to achieve $\epsilon_r$ approximation error on the function value gap of an $\ell$-smooth and $(\ell + r)$-strongly convex function is $\mathcal{O}(\sqrt{(\ell+r)/r}\log(1/\epsilon_r)) = \tilde{\mathcal{O}}(\sqrt{\ell D^2/\epsilon})$, where $\tilde{\mathcal{O}}$ hides logarithmic terms. $\square$

## A.2 Inexact Deterministic Gradient Oracle

This section contains proof of Lemma 3.4 and Theorem 3.5 for the guarantees in the inexact deterministic gradient case. We first study the convergence behavior of AGD [60] for smooth and strongly-convex functions under the inexact gradient oracle. For the sake of simplicity and to enable a general analysis, we slightly abuse notation here to consider the optimization problem
$$\min_{x \in \mathcal{X}} f(x),$$
where $f : \mathcal{X} \to \mathbb{R}$ satisfies the following assumption.

**Assumption A.2.** $f(x)$ is $\ell$-smooth and $\mu$-strongly convex on the closed convex domain $\mathcal{X}$.

We consider the inexact gradient oracle defined below (referred to as $\delta$-oracle in this section).

**Definition 7.** ($\delta$-oracle) At any querying point $x \in \mathcal{X}$, the $\delta$-oracle returns a vector $g(x) \in \mathbb{R}^d$ such that $\|g(x) - \nabla f(x)\|^2 \le \delta^2$, where $\nabla f(x)$ is the true gradient of $f(x)$.

In previous work, Devolder et al. [27] define a different inexact oracle that is motivated by the exact first-order oracle and study the convergence behavior of first-order algorithms including AGD.

**Definition 8.** (($\delta, \ell, \mu$)-oracle [27]) At any querying point $x \in \mathcal{X}$, the ($\delta, \ell, \mu$)-oracle returns approximate first-order information $(f_{\delta,\ell,\mu}(x), g_{\delta,\ell,\mu}(x))$ such that for any $y \in \mathcal{X}$,
$$\frac{\mu}{2}\|x - y\|^2 \le f(y) - (f_{\delta,\ell,\mu}(x) + g_{\delta,\ell,\mu}(x)^\top(y - x)) \le \frac{\ell}{2}\|x - y\|^2 + \delta.$$

The lemma below characterizes that the two oracles can be transformed into each other (adapted from Devolder et al. [27, 28]).

**Lemma A.3.** Under Assumption A.2. A $\delta$-oracle can be transformed to a $(\delta', \ell', \mu')$-oracle with $\delta' = (1/(2\ell) + 1/\mu)\delta^2$, $\ell' = 2\ell$, and $\mu' = \mu/2$. A $(\delta, \ell, \mu)$-oracle can be transformed to a $\delta'$-oracle for $\delta'$ defined in (7).

*Proof.* Given a $\delta$-oracle that returns $g(x)$ at any point $x \in \mathcal{X}$, we construct a $(\delta', \ell', \mu')$-oracle as
$$f_{\delta',\ell',\mu'}(x) = f(x) - \frac{\delta^2}{\mu}, \quad g_{\delta',\ell',\mu'}(x) = g(x).$$

By $\ell$-smoothness of $f(x)$ and fact $(iii)$ in Lemma A.1, we have that
$$f(y) \le f(x) + \nabla f(x)^\top(y - x) + \frac{\ell}{2}\|x - y\|^2$$
$$= f(x) + g(x)^\top(y - x) + (\nabla f(x) - g(x))^\top(y - x) + \frac{\ell}{2}\|x - y\|^2 \tag{5}$$
$$\le f(x) + g(x)^\top(y - x) + \ell\|x - y\|^2 + \frac{\delta^2}{2\ell}.$$

Similarly by $\mu$-strong convexity of $f(x)$ and fact $(iii)$ in Lemma A.1, we have that

$$
\begin{aligned}
f(y) &\geq f(x) + \nabla f(x)^\top (y - x) + \frac{\mu}{2}\|x - y\|^2 \\
&= f(x) + g(x)^\top (y - x) + (\nabla f(x) - g(x))^\top (y - x) + \frac{\mu}{2}\|x - y\|^2 \\
&\geq f(x) + g(x)^\top (y - x) + \frac{\mu}{4}\|x - y\|^2 - \frac{\delta^2}{\mu}.
\end{aligned}
\tag{6}
$$

Combined the above two equations together, we obtain that

$$
\frac{\mu}{4}\|x - y\|^2 \leq f(y) - \left(f(x) - \frac{\delta^2}{\mu} + g(x)^\top (y - x)\right) \leq \ell\|x - y\|^2 + \left(\frac{1}{2\ell} + \frac{1}{\mu}\right)\delta^2.
$$

This concludes the proof of the first part. For the second part, given a $(\delta, \ell, \mu)$-oracle in Definition 8, we construct a $\delta'$-oracle as follows: $g(x) = g_{\delta,\ell,\mu}(x)$. Taking $y = x$ in Definition 8, we obtain $\forall x$,

$$
f_{\delta,\ell,\mu}(x) \leq f(x) \leq f_{\delta,\ell,\mu}(x) + \delta.
$$

Therefore, by strong-convexity of $f(x)$, we have that $\forall x, y$,

$$
\begin{aligned}
f(y) &\geq f(x) + \nabla f(x)^\top (y - x) + \frac{\mu}{2}\|x - y\|^2 \\
&\geq f_{\delta,\ell,\mu}(x) + \nabla f(x)^\top (y - x) + \frac{\mu}{2}\|x - y\|^2.
\end{aligned}
$$

Combined with the second part of Definition 8, we obtain that $\forall x, y$,

$$
(\nabla f(x) - g_{\delta,\ell,\mu}(x))^\top (y - x) \leq \frac{\ell - \mu}{2}\|x - y\|^2 + \delta.
$$

Then by a similar proof as for the convex case in Devolder et al. [28]. Let $\Delta(x) = \nabla f(x) - g_{\delta,\ell,\mu}(x)$ and $y = x + \min\{\sqrt{2\delta/(\ell - \mu)}, r(x)\}\Delta(x)/\|\Delta(x)\|$ for $r(x) = \max\{r \in \mathbb{R} \,|\, (x + r\Delta(x)/\|\Delta(x)\|) \in \mathcal{X}\}$. We have that

$$
\|\nabla f(x) - g_{\delta,\ell,\mu}(x)\| \leq
\begin{cases}
\sqrt{2\delta(\ell - \mu)}, & \text{when } \sqrt{\frac{2\delta}{\ell - \mu}} \leq r(x), \\
\frac{\ell - \mu}{2}r(x) + \frac{\delta}{r(x)}, & \text{otherwise.}
\end{cases}
\tag{7}
$$

Since we use $g(x) = g_{\delta,\ell,\mu}(x)$, the proof is complete. $\qquad\square$

Devolder et al. [27] prove that AGD equipped with $(\delta, \ell, \mu)$-oracle in Definition 8 converges to a $\mathcal{O}(\delta\sqrt{\ell/\mu})$-neighborhood of the optimal solution with accelerated rate $T = \tilde{\mathcal{O}}(\sqrt{\ell/\mu})$:

$$
f(x_T) - f^* \leq \mathcal{O}\left(\exp\left(-T\sqrt{\frac{\mu}{\ell}}\right) + \delta\sqrt{\frac{\ell}{\mu}}\right),
$$

where $x_T$ is the output of $T$-step AGD and $f^*$ is the optimal value. They further establish a lower-bound showing tightness of the $\mathcal{O}(\delta\sqrt{\ell/\mu})$ error for any first-order methods with accelerated rate.

Here, we are interested in the performance of AGD under the $\delta$-oracle in Definition 7. Motivated by the transformation in Lemma A.3, we choose the parameters in AGD as follows:

$$
\begin{aligned}
x_{t+1} &= \Pi_{\mathcal{X}}\left(y_t - \frac{1}{2\ell}g(y_t)\right), \\
y_{t+1} &= x_{t+1} + \frac{2 - \sqrt{\mu/\ell}}{2 + \sqrt{\mu/\ell}}(x_{t+1} - x_t).
\end{aligned}
\tag{8}
$$

The results can be implied by Devolder et al. [27] together with Lemma A.3. We provide detailed proof in the following for completeness of the paper.

**Lemma A.4.** *Under Assumption A.2. Let $x^*$ be the unique minimizer of $f(x)$ and $\kappa = \ell/\mu$ be the condition number. Given an inexact $\delta$-oracle in Definition 7. Starting from $y_0 = x_0$, AGD with updates (8) for $t = 0, 1, \cdots, T-1$ converges with*

$$f(x_T) - f(x^*) \leq \exp\left(-\frac{T}{2\sqrt{\kappa}}\right)\left(f(x_0) - f(x^*) + \frac{\mu}{4}\|x_0 - x^*\|^2\right) + \sqrt{\kappa}\left(\frac{1}{\ell} + \frac{2}{\mu}\right)\delta^2.$$

*Proof.* By (5) in the proof of Lemma A.3, we have that

$$f(x_{t+1}) \leq f(y_t) + g(y_t)^\top(x_{t+1} - y_t) + \ell\|x_{t+1} - y_t\|^2 + \frac{\delta^2}{2\ell}.$$

Similarly by (6), we know for any $x \in \mathcal{X}$,

$$f(x) \geq f(y_t) + g(y_t)^\top(x - y_t) + \frac{\mu}{4}\|x - y_t\|^2 - \frac{\delta^2}{\mu}.$$

Combing the above two results, for any $x \in \mathcal{X}$, we have

$$f(x_{t+1}) - f(x) = f(x_{t+1}) - f(y_t) + f(y_t) - f(x)$$

$$\leq g(y_t)^\top(x_{t+1} - x) + \ell\|x_{t+1} - y_t\|^2 - \frac{\mu}{4}\|x - y_t\|^2 + \left(\frac{1}{2\ell} + \frac{1}{\mu}\right)\delta^2$$

$$\leq -2\ell(x_{t+1} - y_t)^\top(x_{t+1} - x) + \ell\|x_{t+1} - y_t\|^2 - \frac{\mu}{4}\|x - y_t\|^2 + \left(\frac{1}{2\ell} + \frac{1}{\mu}\right)\delta^2$$

$$= -\ell\|x_{t+1} - y_t\|^2 + 2\ell(x_{t+1} - y_t)^\top(x - y_t) - \frac{\mu}{4}\|x - y_t\|^2 + \left(\frac{1}{2\ell} + \frac{1}{\mu}\right)\delta^2,$$

where in the last inequality we use the optimality condition of the projection step such that $\forall\, x \in \mathcal{X}$,

$$\left(x_{t+1} - y_t + \frac{1}{2\ell}g(y_t)\right)^\top(x - x_{t+1}) \geq 0.$$

Let $\theta := 1/(2\sqrt{\kappa}) = \sqrt{\mu/(4\ell)}$. Setting $x = x_t$ and $x = x^*$ in the above equation, we get

$$(1-\theta)(f(x_{t+1}) - f(x_t)) \leq -\ell(1-\theta)\|x_{t+1} - y_t\|^2 + 2\ell(1-\theta)(x_{t+1} - y_t)^\top(x_t - y_t)$$

$$- \frac{\mu}{4}(1-\theta)\|x_t - y_t\|^2 + (1-\theta)\left(\frac{1}{2\ell} + \frac{1}{\mu}\right)\delta^2,$$

$$\theta(f(x_{t+1}) - f(x^*)) \leq -\ell\theta\|x_{t+1} - y_t\|^2 + 2\ell\theta(x_{t+1} - y_t)^\top(x^* - y_t)$$

$$- \frac{\mu}{4}\theta\|x^* - y_t\|^2 + \theta\left(\frac{1}{2\ell} + \frac{1}{\mu}\right)\delta^2.$$

Let $\Delta_t := f(x_t) - f(x^*) \geq 0$. Summing the above two up, by fact $(i)$ in Lemma A.1, we obtain

$$\Delta_{t+1} - (1-\theta)\Delta_t \leq -\ell\|x_{t+1} - y_t\|^2 + 2\ell(x_{t+1} - y_t)^\top((1-\theta)x_t + \theta x^* - y_t) - \frac{\mu}{4}\theta\|x^* - y_t\|^2$$

$$- \frac{\mu}{4}(1-\theta)\|x_t - y_t\|^2 + \left(\frac{1}{2\ell} + \frac{1}{\mu}\right)\delta^2,$$

$$= \ell\|y_t - (1-\theta)x_t - \theta x^*\|^2 - \ell\|x_{t+1} - (1-\theta)x_t - \theta x^*\|^2 - \frac{\mu}{4}\theta\|x^* - y_t\|^2$$

$$- \frac{\mu}{4}(1-\theta)\|x_t - y_t\|^2 + \left(\frac{1}{2\ell} + \frac{1}{\mu}\right)\delta^2.$$

Let $\theta u_t := x_t - (1-\theta)x_{t-1}$ for $t \geq 1$. From the update (8) of AGD, we observe

$$(1+\theta)y_t = (1+\theta)x_t + (1-\theta)(x_t - x_{t-1})$$

$$= 2x_t - (1-\theta)x_{t-1}$$

$$= x_t + \theta u_t.$$

Rearranging terms, we can get $x_t = (1+\theta)y_t - \theta u_t$ and thus

$$
\begin{aligned}
y_t - (1-\theta)x_t &= y_t - (1-\theta)((1+\theta)y_t - \theta u_t) \\
&= y_t - ((1-\theta^2)y_t - (1-\theta)\theta u_t) \\
&= \theta(\theta y_t + (1-\theta)u_t).
\end{aligned}
$$

It is easy to verify that the above also holds when $u_0 = x_0 = y_0$. Since $\ell\theta^2 = \mu/4$, we have that

$$
\begin{aligned}
&\Delta_{t+1} - (1-\theta)\Delta_t \\
&\leq \frac{\mu}{4}\|\theta y_t + (1-\theta)u_t - x^*\|^2 - \frac{\mu}{4}\|u_{t+1} - x^*\|^2 - \frac{\mu}{4}\theta\|x^* - y_t\|^2 + \left(\frac{1}{2\ell} + \frac{1}{\mu}\right)\delta^2 \\
&= \frac{\mu}{4}\|(1-\theta)(u_t - x^*) + \theta(y_t - x^*)\|^2 - \frac{\mu}{4}\|u_{t+1} - x^*\|^2 - \frac{\mu}{4}\theta\|x^* - y_t\|^2 + \left(\frac{1}{2\ell} + \frac{1}{\mu}\right)\delta^2 \\
&\leq \frac{\mu}{4}(1-\theta)\|u_t - x^*\|^2 - \frac{\mu}{4}\|u_{t+1} - x^*\|^2 + \left(\frac{1}{2\ell} + \frac{1}{\mu}\right)\delta^2,
\end{aligned}
$$

where we use fact $(iv)$ in Lemma A.1. Rearranging terms, we then obtain

$$
\Delta_{t+1} + \frac{\mu}{4}\|u_{t+1} - x^*\|^2 \leq (1-\theta)\left(\Delta_t + \frac{\mu}{4}\|u_t - x^*\|^2\right) + \left(\frac{1}{2\ell} + \frac{1}{\mu}\right)\delta^2.
$$

Unrolling the recursion, we have that

$$
\begin{aligned}
f(x_T) - f(x^*) &\leq \Delta_T + \frac{\mu}{4}\|u_T - x^*\|^2 \\
&\leq (1-\theta)^T\left(\Delta_0 + \frac{\mu}{4}\|u_0 - x^*\|^2\right) + ((1-\theta)^{T-1} + \cdots + (1-\theta) + 1)\left(\frac{1}{2\ell} + \frac{1}{\mu}\right)\delta^2 \\
&\leq \exp(-\theta T)\left(f(x_0) - f(x^*) + \frac{\mu}{4}\|u_0 - x^*\|^2\right) + \frac{1}{\theta}\left(\frac{1}{2\ell} + \frac{1}{\mu}\right)\delta^2 \\
&= \exp\left(-\frac{T}{2\sqrt{\kappa}}\right)\left(f(x_0) - f(x^*) + \frac{\mu}{4}\|x_0 - x^*\|^2\right) + \sqrt{\kappa}\left(\frac{1}{\ell} + \frac{2}{\mu}\right)\delta^2,
\end{aligned}
$$

where we use the fact that $1 + \eta \leq e^\eta, \forall \eta \in \mathbb{R}$. $\qquad\square$

Lemma 3.4 immediately follows from Lemma A.4. With the above results at hand, we are ready to show proof of Theorem 3.5 below.

*Proof of Theorem 3.5.* For the convergence guarantee, similarly to the perturbed initialization case in (4), for $x^* \in \arg\min_{x \in \mathcal{X}} F(x)$ and $x_r^* = \arg\min_{x \in \mathcal{X}} F_r(x)$, we have that

$$
F(x_r) - F(x^*) \leq F_r(x_r) - F_r(x_r^*) + \frac{rD^2}{2}.
$$

For the reproducibility guarantee, using $r$-strong-convexity of $F_r(x)$, we can obtain that

$$
\begin{aligned}
\|x_r - x_r'\| &\leq \|x_r - x_r^*\| + \|x_r^* - x_r'\| \\
&\leq \sqrt{\frac{2(F_r(x_r) - F_r(x_r^*))}{r}} + \sqrt{\frac{2(F_r(x_r') - F_r(x_r^*))}{r}}.
\end{aligned}
$$

Applying Lemma 3.4, if (Inexact-AGD) is used as the base algorithm $\mathcal{A}$ and $x_r$ is the output given initialization $y_0 = x_0$ after $T$ iterations, since $r = \epsilon/D^2 \leq \ell$, we know that

$$
\begin{aligned}
&F_r(x_r) - F_r(x_r^*) \\
&\leq \exp\left(-\frac{T}{2}\sqrt{\frac{r}{\ell+r}}\right)\left(F_r(x_0) - F_r(x_r^*) + \frac{r}{4}\|x_0 - x_r^*\|^2\right) + \sqrt{\frac{\ell+r}{r}}\left(\frac{1}{\ell+r} + \frac{2}{r}\right)\delta^2 \\
&\leq \exp\left(-\frac{T}{2}\sqrt{\frac{r}{2\ell}}\right)\left(F_r(x_0) - F_r(x_r^*) + \frac{r}{4}\|x_0 - x_r^*\|^2\right) + 5\delta^2\sqrt{\frac{\ell}{2r^3}}.
\end{aligned}
$$

When setting $T = \mathcal{O}(\sqrt{\ell/r}\log(r^{3/2}/\delta^2))$, this means the algorithm converges to $F_r(x_r) - F_r(x_r^*) \le 6\delta^2\sqrt{\ell/(2r^3)}$ and $\|x_r - x_r^*\|^2 \le 12\delta^2\sqrt{\ell/(2r^5)}$. Therefore, since $r = \epsilon/D^2$, we have that

$$F(x_r) - F(x^*) \le \mathcal{O}\left(\frac{\delta^2}{\epsilon^{3/2}} + \epsilon\right),$$

and the reproducibility is $\|x_r - x_r'\|^2 \le \mathcal{O}(\delta^2/\epsilon^{5/2})$. $\qquad\square$

The results suggest that to achieve $\epsilon$-approximation error on the function value gap, we need to set $\delta \le \mathcal{O}(\epsilon^{5/4})$, which is a smaller regime compared to $\delta \le \mathcal{O}(\epsilon)$ in the previous work [1] when $\epsilon \le 1$. Furthermore, optimal reproducibility $\mathcal{O}(\delta^2/\epsilon^2)$ is not attained. We observe from the proof that the additional $\mathcal{O}(\sqrt{\kappa})$-factor in the last term of the error bound in Lemma A.4 leads to this degradation. Since we set $r = \mathcal{O}(\epsilon)$ to balance the convergence rate and approximation error introduced through regularization, this factor can be $\mathcal{O}(\sqrt{1/\epsilon})$. Based on the lower-bound in Devolder et al. [27] for $(\delta, \ell, \mu)$-oracle such that this $\mathcal{O}(\sqrt{\kappa})$-factor is unavoidable for an accelerated convergence rate and the transformation between the two inexact oracles in Lemma A.3, we thus make the conjecture here that the above results cannot be further improved. Algorithms that achieve optimal convergence and reproducibility under this setting require better designs and we leave it for future work.

## B  Preliminary Results in the Minimax Case

In this section, we provide proof of some preliminary results in the minimax setting. We start with a proof of the lower-bounds in Lemma B.3. Sub-optimal guarantees of gradient descent ascent (GDA) in the deterministic case, as well as optimal guarantees of stochastic gradient descent ascent (SGDA), are provided in Section B.2. Sub-optimal results of Extragradient (EG) are proved in Section B.3.

Before that, we introduce some notations and helpful lemmas that will be used in the analysis. We let $z = (x, y)$ and $\tilde{\nabla}F(z) = (\nabla_x F(x, y), -\nabla_y F(x, y))$ for simplicity of the notation in the remaining of the paper. The following results will be frequently used.

**Lemma B.1.** *Under Assumption 4.1, the operator $\tilde{\nabla}F$ is monotone and $\ell$-Lipschitz. That is, $\forall z_1, z_2 \in \mathcal{X} \times \mathcal{Y}$, $\|\tilde{\nabla}F(z_1) - \tilde{\nabla}F(z_2)\| \le \ell\|z_1 - z_2\|$ and $(\tilde{\nabla}F(z_1) - \tilde{\nabla}F(z_2))^\top(z_1 - z_2) \ge 0$. Moreover, $\forall z \in \mathcal{X} \times \mathcal{Y}$, $\|\tilde{\nabla}F(z)\| \le L$ where we define $L := \min\|\tilde{\nabla}F(z^*)\| + \sqrt{2}\ell D$ for minimum taking w.r.t. any saddle point $z^* = (x^*, y^*)$ of $F(x, y)$.*

*Proof.* Lipschitzness of $\tilde{\nabla}F$ directly follows from $\ell$-smoothness of $F(x, y)$. The fact that $\tilde{\nabla}F$ is monotone when $F(x, y)$ is convex-concave is well-known in the literature (e.g., see Theorem 1 in Rockafellar [65]). For the last statement, taking any saddle point $z^*$, we have that $\forall z \in \mathcal{X} \times \mathcal{Y}$,

$$\|\tilde{\nabla}F(z)\| \le \|\tilde{\nabla}F(z^*)\| + \|\tilde{\nabla}F(z) - \tilde{\nabla}F(z^*)\|$$
$$\le \|\tilde{\nabla}F(z^*)\| + \ell\|z - z^*\|.$$

The proof is complete since the domain $\mathcal{X}$ and $\mathcal{Y}$ have a diameter of $D$. $\qquad\square$

**Lemma B.2.** *Under Assumption 4.1. For some integer $T \ge 1$, let $z_t = (x_t, y_t)$ for $t = 0, 1, \cdots, T-1$ and $\bar{z}_T = (\bar{x}_T, \bar{y}_T) = (1/T)\sum_{t=0}^{T-1}(x_t, y_t)$. If $\forall z \in \mathcal{X} \times \mathcal{Y}$, $(1/T)\sum_{t=0}^{T-1}\tilde{\nabla}F(z_t)^\top(z_t - z) \le \epsilon$, then it satisfies that $\max_{y \in \mathcal{Y}} F(\bar{x}_T, y) - \min_{x \in \mathcal{X}} F(x, \bar{y}_T) \le \epsilon$.*

*Proof.* Since $F(x, y)$ is convex-concave, we get that $\forall z = (x, y) \in \mathcal{X} \times \mathcal{Y}$,

$$F(x_t, y) - F(x, y_t) = F(x_t, y) - F(x_t, y_t) + F(x_t, y_t) - F(x, y_t)$$
$$\le \nabla_x F(x_t, y_t)^\top(x_t - x) - \nabla_y F(x_t, y_t)^\top(y_t - y)$$
$$= \tilde{\nabla}F(z_t)^\top(z_t - z).$$

Summing up from $t = 0$ to $T - 1$ and dividing both sides by $T$, by Jensen's inequality, we have that

$$F(\bar{x}_T, y) - F(x, \bar{y}_T) \le \frac{1}{T}\sum_{t=0}^{T-1}\tilde{\nabla}F(z_t)^\top(z_t - z)$$
$$\le \epsilon.$$

Taking $y = \arg\max_{v \in \mathcal{Y}} F(\bar{x}_T, v)$ and $x = \arg\min_{u \in \mathcal{X}} F(u, \bar{y}_T)$, we conclude the proof. $\qquad\square$

### B.1 Lower-bounds for Reproducibility

The lower-bounds follow from the minimization setting [1].

**Lemma B.3.** *For smooth convex-concave minimax optimization under Assumption 4.1, the reproducibility, i.e., $(\epsilon, \delta)$-deviation, of any algorithm $\mathcal{A}$ is at least $(i)$ $\Omega(\delta^2)$ for the inexact initialization oracle; $(ii)$ $\Omega(\delta^2/\epsilon^2)$ for the deterministic inexact gradient oracle; $(iii)$ $\Omega(\delta^2/(T\epsilon^2))$ for the stochastic gradient oracle, where $T$ is the total number of iterations of the algorithm.*

*Proof.* The lower-bound of reproducibility in Ahn et al. [1] for smooth convex minimization problems is also a valid lower-bound for smooth convex-concave minimax problems. To show this, we consider a special case of the minimax problem (1) where the domain $\mathcal{Y}$ is a singleton, i.e., $\mathcal{Y} = \{y_0\}$ for some $y_0$. Then the original smooth convex-concave minimax problem $\min_{x \in \mathcal{X}} \max_{y \in \mathcal{Y}} F(x, y)$ is equivalent to the smooth convex minimization problem $\min_{x \in \mathcal{X}} F(x, y_0)$. For all three inexact oracles, let $(\hat{x}, \hat{y})$ and $(\hat{x}', \hat{y}')$ be the $\epsilon$-approximate outputs of independent two runs of the same algorithm, i.e., the duality gap can be upper-bounded by $\epsilon$, then the reproducibility $\|\hat{x} - \hat{x}'\|^2 + \|\hat{y} - \hat{y}'\|^2 = \|\hat{x} - \hat{x}'\|^2$ since $\hat{y} = \hat{y}' = y_0$. Moreover, $\hat{x}$ and $\hat{x}'$ are also $\epsilon$-approximate solutions of the function $F(x, y_0)$ based on the definition of duality gap. Thus the lower-bound in the minimization setting [1] directly implies the lower-bound in the minimax setting. To be specific, the lower-bound is: $(i)$ $\Omega(\delta^2)$ for the inexact initialization case; $(ii)$ $\Omega(\delta^2/\epsilon^2)$ for the inexact deterministic gradient case; and $(iii)$ $\Omega(\delta^2/(T\epsilon^2))$ for the stochastic gradient case. $\qquad\square$

### B.2 Guarantees of Gradient Descent Ascent

This section provides proof of Theorem 4.2 for the sub-optimal guarantees of GDA in the deterministic setting and Theorem 5.1 for the optimal guarantees of SGDA in the stochastic setting. We first provide a general analysis and then expand it for three different inexact oracles in subsequent sections.

#### B.2.1 General Analysis

---
**Algorithm 4** Gradient Descent Ascent
---
**Input:** Stepsize $\alpha > 0$, initialization $(x_0, y_0)$, number of iterations $T > 0$.
**for** $t = 0, 1, \cdots T - 1$ **do**
$\quad y_{t+1} = \Pi_{\mathcal{Y}}(y_t + \alpha \nabla_y F(x_t, y_t))$,
$\quad x_{t+1} = \Pi_{\mathcal{X}}(x_t - \alpha \nabla_x F(x_t, y_t))$.
**Output:** $(\bar{x}_T, \bar{y}_T) = (1/T)\sum_{t=0}^{T-1}(x_t, y_t)$.

---

We consider (stochastic) gradient descent ascent (GDA/SGDA) outlined in Algorithm 4 for solving minimax problems (1) or (3). The algorithm iteratively updates the variables $x_t$ and $y_t$ using exact gradients $\nabla F(x_t, y_t)$, or inexact gradients $G(x_t, y_t)$, or stochastic gradients $\nabla f(x_t, y_t; \xi_t)$ based on different types of the inexact oracles in Definition 4.

We first analyze the behavior of GDA with access to exact gradients. It is well-known that the last iterate of GDA can diverge even for bilinear functions [55, 9, 36], and the average iterates converge with a sub-optimal rate $\mathcal{O}(1/\sqrt{T})$. We provide proof for completeness.

**Lemma B.4.** *Under Assumption 4.1. When setting the stepsize to $\alpha = 1/(\ell\sqrt{T})$, the average iterates $(\bar{x}_T, \bar{y}_T)$ of GDA converges with*

$$\max_{y \in \mathcal{Y}} F(\bar{x}_T, y) - \min_{x \in \mathcal{X}} F(x, \bar{y}_T) \leq \frac{\ell D^2 + L^2/(2\ell)}{\sqrt{T}},$$

*This suggests $\mathcal{O}(1/\epsilon^2)$ gradient complexity is required to achieve $\epsilon$-saddle point.*

*Proof.* Recall $z_t = (x_t, y_t)$ and $\tilde{\nabla}F(z_t) = (\nabla_x F(x_t, y_t), -\nabla_y F(x_t, y_t))$. The GDA updates in Algorithm 4 can be simplified to

$$z_{t+1} = \Pi_{\mathcal{X} \times \mathcal{Y}}(z_t - \alpha \tilde{\nabla}F(z_t)). \tag{9}$$

Since the projection step is nonexpansive [12], we have that $\forall z = (x, y) \in \mathcal{X} \times \mathcal{Y}$,

$$\|z_{t+1} - z\|^2 \leq \|z_t - \alpha \tilde{\nabla} F(z_t) - z\|^2$$
$$= \|z_t - z\|^2 - 2\alpha \tilde{\nabla} F(z_t)^\top (z_t - z) + \alpha^2 \|\tilde{\nabla} F(z_t)\|^2.$$

Rearranging terms and using Lemma B.1, we can obtain that

$$\tilde{\nabla} F(z_t)^\top (z_t - z) \leq \frac{1}{2\alpha} \big(\|z_t - z\|^2 - \|z_{t+1} - z\|^2\big) + \frac{\alpha L^2}{2}.$$

Taking summation from $t = 0$ to $T - 1$ and dividing both sides by $T$, by Lemma B.2, we thus have

$$\max_{y \in \mathcal{Y}} F(\bar{x}_T, y) - \min_{x \in \mathcal{X}} F(x, \bar{y}_T) \leq \frac{D^2}{\alpha T} + \frac{\alpha L^2}{2}.$$

When setting $\alpha = 1/(\ell \sqrt{T})$, this means the complexity is required to be $T \geq (\ell D^2 + L^2/(2\ell))^2/\epsilon^2$ to achieve an $\epsilon$-saddle point such that $\max_{y \in \mathcal{Y}} F(\bar{x}_T, y) - \min_{x \in \mathcal{X}} F(x, \bar{y}_T) \leq \epsilon$. $\qquad \square$

**Lemma B.5.** *Under Assumption 4.1, the GDA update* (9) *is* $(1 + \alpha^2 \ell^2)$*-expansive. That is, if* $(x_{t+1}, y_{t+1})$ *is obtained through 1-step of the update given* $(x_t, y_t)$*, and* $(x'_{t+1}, y'_{t+1})$ *is obtained given* $(x'_t, y'_t)$*, we have that*

$$\|x_{t+1} - x'_{t+1}\|^2 + \|y_{t+1} - y'_{t+1}\|^2 \leq (1 + \alpha^2 \ell^2)\big(\|x_t - x'_t\|^2 + \|y_t - y'_t\|^2\big).$$

*Proof.* Recall $z_t = (x_t, y_t)$ and $z'_t = (x'_t, y'_t)$. By the updates of GDA (9), we get that

$$\|z_{t+1} - z'_{t+1}\|^2 \leq \|(z_t - z'_t) - \alpha(\tilde{\nabla} F(z_t) - \tilde{\nabla} F(z'_t))\|^2$$
$$\leq \|z_t - z'_t\|^2 + \alpha^2 \|\tilde{\nabla} F(z_t) - \tilde{\nabla} F(z'_t)\|^2 - 2\alpha(\tilde{\nabla} F(z_t) - \tilde{\nabla} F(z'_t))^\top (z_t - z'_t)$$
$$\leq (1 + \alpha^2 \ell^2)\big(\|x_t - x'_t\|^2 + \|y_t - y'_t\|^2\big),$$

where we use the fact that the projection step is nonexpansive and Lemma B.1. $\qquad \square$

### B.2.2 Inexact Initialization Oracle

**Theorem B.6** (Restate Theorem 4.2, part $(i)$)**.** *Under Assumptions 4.1. The average iterate* $(\bar{x}_T, \bar{y}_T)$ *of GDA satisfies* $\max_{y \in \mathcal{Y}} F(\bar{x}_T, y) - \min_{x \in \mathcal{X}} F(x, \bar{y}_T) \leq \mathcal{O}(\epsilon)$ *with complexity* $T = \mathcal{O}(1/\epsilon^2)$ *if setting stepsize* $\alpha = 1/(\ell \sqrt{T})$*. The reproducibility, i.e.,* $(\epsilon, \delta)$*-deviation between outputs* $(\bar{x}_T, \bar{y}_T)$ *and* $(\bar{x}'_T, \bar{y}'_T)$ *of two independent runs given different initialization is* $\|\bar{x}_T - \bar{x}'_T\|^2 + \|\bar{y}_T - \bar{y}'_T\|^2 \leq \mathcal{O}(\delta^2)$*.*

*Proof.* The convergence analysis directly follows from Lemma B.4. For the reproducibility analysis, by Lemma B.5 and the choice that $\alpha = 1/(\ell \sqrt{T})$, we have that for $t = 1, 2, \cdots, T - 1$,

$$\|x_t - x'_t\|^2 + \|y_t - y'_t\|^2 \leq (1 + \alpha^2 \ell^2)\big(\|x_{t-1} - x'_{t-1}\|^2 + \|y_{t-1} - y'_{t-1}\|^2\big)$$
$$= \left(1 + \frac{1}{T}\right)\big(\|x_{t-1} - x'_{t-1}\|^2 + \|y_{t-1} - y'_{t-1}\|^2\big)$$
$$\leq \left(1 + \frac{1}{T}\right)^t \big(\|x_0 - x'_0\|^2 + \|y_0 - y'_0\|^2\big).$$

The above also trivially holds for $t = 0$. Therefore, by Jensen's inequality, we can obtain that

$$\|\bar{x}_T - \bar{x}'_T\|^2 + \|\bar{y}_T - \bar{y}'_T\|^2 \leq \frac{1}{T} \sum_{t=0}^{T-1} \big(\|x_t - x'_t\|^2 + \|y_t - y'_t\|^2\big)$$
$$\leq \delta^2 \cdot \frac{1}{T} \sum_{t=0}^{T-1} \left(1 + \frac{1}{T}\right)^t$$
$$\leq e\delta^2.$$

The choice of $\alpha$ is to avoid exponential dependence on $\ell$ in the reproducibility bound. $\qquad \square$

### B.2.3 Inexact Deterministic Gradient Oracle

When only given an inexact gradient oracle in Definition 4, the updates of GDA become

$$z_{t+1} = \Pi_{\mathcal{X} \times \mathcal{Y}}(z_t - \alpha \tilde{G}(z_t)), \tag{10}$$

where we let $\tilde{G}(z_t) = (G_x(x_t, y_t), -G_y(x_t, y_t))$ for the inexact gradients.

**Theorem B.7** (Restate Theorem 4.2, part $(ii)$). *Under Assumptions 4.1. Given an inexact deterministic gradient oracle in Definition 4 with $\delta \leq \mathcal{O}(\epsilon)$. The average iterate $(\bar{x}_T, \bar{y}_T)$ of GDA satisfies $\max_{y \in \mathcal{Y}} F(\bar{x}_T, y) - \min_{x \in \mathcal{X}} F(x, \bar{y}_T) \leq \mathcal{O}(\epsilon)$ with complexity $T = \mathcal{O}(1/\epsilon^2)$ if setting stepsize $\alpha = 1/(\ell\sqrt{T})$. Furthermore, the reproducibility is $\|\bar{x}_T - \bar{x}_T'\|^2 + \|\bar{y}_T - \bar{y}_T'\|^2 \leq \mathcal{O}(\delta^2/\epsilon^2)$.*

*Proof.* We first show the optimization guarantee. By the GDA updates in (10) and Definition 4 such that $\|\tilde{\nabla} F(z_t) - \tilde{G}(z_t)\|^2 \leq \delta^2$, we have that for any $z = (x, y) \in \mathcal{X} \times \mathcal{Y}$,

$$
\begin{aligned}
\|z_{t+1} - z\|^2 &\leq \|z_t - z\|^2 - 2\alpha \tilde{G}(z_t)^\top (z_t - z) + \alpha^2 \|\tilde{G}(z_t)\|^2 \\
&\leq \|z_t - z\|^2 - 2\alpha \tilde{\nabla} F(z_t)^\top (z_t - z) + 2\alpha^2 \|\tilde{\nabla} F(z_t)\|^2 \\
&\quad + 2\alpha(\tilde{\nabla} F(z_t) - \tilde{G}(z_t))^\top (z_t - z) + 2\alpha^2 \|\tilde{G}(z_t) - \tilde{\nabla} F(z_t)\|^2 \\
&\leq \|z_t - z\|^2 - 2\alpha \tilde{\nabla} F(z_t)^\top (z_t - z) + 2\alpha^2 L^2 + 2\sqrt{2}\alpha \delta D + 2\alpha^2 \delta^2.
\end{aligned} \tag{11}
$$

Taking summation from $t = 0$ to $T - 1$, we obtain that

$$\frac{1}{T} \sum_{t=0}^{T-1} \tilde{\nabla} F(z_t)^\top (z_t - z) \leq \frac{\|z_0 - z\|^2}{2\alpha T} + \alpha(L^2 + \delta^2) + \sqrt{2}\delta D.$$

Supposing $\delta \leq \epsilon/(2\sqrt{2}D)$ and setting $\alpha = 1/(\ell\sqrt{T})$, by Lemma B.2, this means

$$\max_{y \in \mathcal{Y}} F(\bar{x}_T, y) - \min_{x \in \mathcal{X}} F(x, \bar{y}_T) \leq \frac{\ell D^2 + (L^2 + \delta^2)/\ell}{\sqrt{T}} + \frac{\epsilon}{2}.$$

$\epsilon$-saddle point is guaranteed when $T = c/\epsilon^2$ for some constant $c \geq 4(\ell D^2 + (L^2 + \delta^2)/\ell)^2$.

We then prove the reproducibility guarantee. Let $\{z_t\}_{t=1}^T$ and $\{z_t'\}_{t=1}^T$ be the trajectories of two independent runs of GDA with the same initial point $z_0 \in \mathcal{X} \times \mathcal{Y}$ and stepsize $\alpha > 0$. By the GDA updates (10) and Lemma B.5, we have that

$$
\begin{aligned}
\|z_{t+1} - z_{t+1}'\| &\leq \|(z_t - z_t') - \alpha(\tilde{G}(z_t) - \tilde{G}(z_t'))\| \\
&\leq \|(z_t - z_t') - \alpha(\tilde{\nabla} F(z_t) - \tilde{\nabla} F(z_t'))\| + 2\alpha\delta \\
&\leq \sqrt{1 + \alpha^2 \ell^2} \|z_t - z_t'\| + 2\alpha\delta.
\end{aligned} \tag{12}
$$

Since the initialization $z_0 = z_0'$ is the same, we obtain that for any $t = 1, 2, \cdots, T - 1$,

$$
\begin{aligned}
\|z_t - z_t'\| &\leq \left(\sqrt{1 + \alpha^2 \ell^2}\right)^t \|z_0 - z_0'\| + 2\alpha\delta \left(1 + \sqrt{1 + \alpha^2 \ell^2} + \cdots + \left(\sqrt{1 + \alpha^2 \ell^2}\right)^{t-1}\right) \\
&= 2\alpha\delta \sum_{i=0}^{t-1} (1 + \alpha^2 \ell^2)^{i/2} \\
&\leq 2\alpha\delta \cdot t(1 + \alpha^2 \ell^2)^{T/2}.
\end{aligned}
$$

The above also holds for $t = 0$ denoting $\sum_{i=0}^{-1} = 0$. Setting $\alpha = 1/(\ell\sqrt{T})$, the reproducibility is

$$
\begin{aligned}
\|\bar{x}_T - \bar{x}_T'\|^2 + \|\bar{y}_T - \bar{y}_T'\|^2 &\leq \frac{1}{T} \sum_{t=0}^{T-1} \left(\|x_t - x_t'\|^2 + \|y_t - y_t'\|^2\right) \\
&\leq \frac{4\alpha^2 \delta^2}{T} (1 + \alpha^2 \ell^2)^T \sum_{t=0}^{T-1} t^2 \\
&\leq \frac{4e}{3\ell^2} \cdot \delta^2 T,
\end{aligned}
$$

which is $\mathcal{O}(\delta^2/\epsilon^2)$ when $T = c/\epsilon^2$ as required in the convergence analysis of GDA. $\qquad \square$

### B.2.4 Stochastic Gradient Oracle

For the stochastic minimax problem (3), with access to a stochastic gradient oracle in Definition 4, SGDA updates for $t = 0, 1, \cdots, T - 1$,

$$z_{t+1} = \Pi_{\mathcal{X} \times \mathcal{Y}}(z_t - \alpha \tilde{\nabla} f(z_t; \xi_t)), \tag{13}$$

where $\tilde{\nabla} f(z_t; \xi_t) = (\nabla_x f(x_t, y_t; \xi_t), -\nabla_y f(x_t, y_t; \xi_t))$ and $\{\xi_t\}_{t=0}^{T-1}$ are i.i.d. samples.

*Proof of Theorem 5.1.* We first show the convergence guarantee. By the SGDA updates in (13), given all the information up to iteration $t$ and taking expectation with respect to $\xi_t$, we have $\forall z \in \mathcal{X} \times \mathcal{Y}$,

$$\mathbb{E}_{\xi_t} \|z_{t+1} - z\|^2 \leq \|z_t - z\|^2 - 2\alpha \mathbb{E}[\tilde{\nabla} f(z_t; \xi_t)^\top (z_t - z)] + \alpha^2 \mathbb{E} \|\tilde{\nabla} f(z_t; \xi_t)\|^2$$
$$= \|z_t - z\|^2 - 2\alpha \tilde{\nabla} F(z_t)^\top (z_t - z) + \alpha^2 \mathbb{E} \|\tilde{\nabla} f(z_t; \xi_t)\|^2.$$

Taking full expectation, rearranging terms, and summing up from $t = 0$ to $T - 1$, we have that

$$\frac{1}{T} \sum_{t=0}^{T-1} \mathbb{E} \left[ \tilde{\nabla} F(z_t)^\top (z_t - z) \right] \leq \frac{\|z_0 - z\|^2}{2\alpha T} + \frac{\alpha(L^2 + \delta^2)}{2}.$$

Therefore, by slightly modifying the proof of Lemma B.2 through taking expectations, and then setting $x = \arg\min_{u \in \mathcal{X}} \mathbb{E}[F(u, \bar{y}_T)]$ and $y = \arg\max_{v \in \mathcal{Y}} \mathbb{E}[F(\bar{x}_T, v)]$, we get

$$\max_{y \in \mathcal{Y}} \mathbb{E}[F(\bar{x}_T, y)] - \min_{x \in \mathcal{X}} \mathbb{E}[F(x, \bar{y}_T)] \leq \frac{D^2}{\alpha T} + \frac{\alpha(L^2 + \delta^2)}{2}.$$

We obtain that $\max_{y \in \mathcal{Y}} \mathbb{E}[F(\bar{x}_T, y)] - \min_{x \in \mathcal{X}} \mathbb{E}[F(x, \bar{y}_T)] \leq (\ell D^2 + (L^2 + \delta^2)/(2\ell))\epsilon$ if the inexactness $\delta = \mathcal{O}(1)$, and we set $\alpha = 1/(\ell \epsilon T)$, $T \geq 1/\epsilon^2$.

We then show the reproducibility guarantee. For two independent runs of SGDA (13) with output $\{z_t\}_{t=1}^T$ and $\{z'_t\}_{t=1}^T$, by Lemma B.5, we have that for any $t = 0, 1, \cdots, T - 1$,

$$\mathbb{E}_{\xi_t, \xi'_t} \|z_{t+1} - z'_{t+1}\|^2$$
$$\leq \mathbb{E} \|(z_t - z'_t) - \alpha(\tilde{\nabla} f(z_t; \xi_t) - \tilde{\nabla} f(z'_t; \xi'_t))\|^2$$
$$= \|z_t - z'_t\|^2 - 2\alpha(\tilde{\nabla} F(z_t) - \tilde{\nabla} F(z'_t))^\top (z_t - z'_t) + \alpha^2 \mathbb{E} \|\tilde{\nabla} f(z_t; \xi_t) - \tilde{\nabla} f(z'_t; \xi'_t)\|^2$$
$$= \|z_t - z'_t\|^2 - 2\alpha(\tilde{\nabla} F(z_t) - \tilde{\nabla} F(z'_t))^\top (z_t - z'_t) + \alpha^2 \mathbb{E} \|\tilde{\nabla} F(z_t) - \tilde{\nabla} F(z'_t)\|^2$$
$$\quad + \alpha^2 \mathbb{E} \|(\tilde{\nabla} f(z_t; \xi_t) - \tilde{\nabla} f(z'_t; \xi'_t)) - (\tilde{\nabla} F(z_t) - \tilde{\nabla} F(z'_t))\|^2$$
$$\leq (1 + \alpha^2 \ell^2) \|z_t - z'_t\|^2 + 4\alpha^2 \delta^2.$$

Unrolling the recursion, noticing $z_0 = z'_0$, we have that for any $t = 0, 1, \cdots, T - 1$,

$$\mathbb{E} \|z_t - z'_t\|^2 \leq 4\alpha^2 \delta^2 \sum_{i=0}^{t-1} (1 + \alpha^2 \ell^2)^i.$$

Since $T \geq 1/\epsilon^2$, we know $\alpha = 1/(\ell \epsilon T) \leq 1/(\ell \sqrt{T})$. The reproducibility is thus

$$\mathbb{E} \left[ \|\bar{x}_T - \bar{x}'_T\|^2 + \|\bar{y}_T - \bar{y}'_T\|^2 \right] \leq \frac{1}{T} \sum_{t=0}^{T-1} \mathbb{E} \left[ \|x_t - x'_t\|^2 + \|y_t - y'_t\|^2 \right]$$
$$\leq \frac{4\alpha^2 \delta^2}{T} \sum_{t=1}^{T-1} \sum_{i=0}^{t-1} (1 + \alpha^2 \ell^2)^i$$
$$\leq \frac{4\alpha^2 \delta^2}{T} \sum_{t=1}^{T-1} t \left( 1 + \frac{1}{T} \right)^T$$
$$\leq 2e\delta^2 \alpha^2 T$$
$$= \frac{2e}{\ell^2} \cdot \frac{\delta^2}{\epsilon^2 T}.$$

The last step uses the choice of $\alpha$ such that $\alpha^2 T = 1/(\ell^2 \epsilon^2 T)$. $\qquad\square$

## B.3 Guarantees of Extragradient

This section provides proof of Theorem 4.3 for the sub-optimal guarantees of Extragradient (EG).

### B.3.1 General Analysis

---

**Algorithm 5** Extragradient

---

**Input:** Stepsize $\alpha > 0$, initialization $(x_0, y_0)$, number of iterations $T > 0$.
**for** $t = 0, 1, \cdots T - 1$ **do**
$\quad y_{t+1/2} = \Pi_{\mathcal{Y}}(y_t + \alpha \nabla_y F(x_t, y_t))$,
$\quad x_{t+1/2} = \Pi_{\mathcal{X}}(x_t - \alpha \nabla_x F(x_t, y_t))$.
$\quad y_{t+1} = \Pi_{\mathcal{Y}}(y_t + \alpha \nabla_y F(x_{t+1/2}, y_{t+1/2}))$,
$\quad x_{t+1} = \Pi_{\mathcal{X}}(x_t - \alpha \nabla_x F(x_{t+1/2}, y_{t+1/2}))$.
**Output:** $(\bar{x}_{T+1/2}, \bar{y}_{T+1/2}) = (1/T) \sum_{t=0}^{T-1} (x_{t+1/2}, y_{t+1/2})$.

---

For deterministic smooth convex-concave minimax optimization, Extragradient [48, 72] (EG), summarized in Algorithm 5, achieves the optimal $\mathcal{O}(1/\epsilon)$ convergence rate. When only given inexact gradients or stochastic gradients, the true gradients are just replaced by $G(x_t, y_t)$ or $\nabla f(x_t, y_t; \xi_t)$.

We provide proof of its $\mathcal{O}(1/\epsilon)$ convergence for completeness. The proof is standard in the literature, e.g., see Nemirovski [59] or Section 4.5 of Bubeck [20].

**Lemma B.8.** *Under Assumption 4.1. When setting the stepsize to $\alpha = 1/\ell$, the average iterates $(\bar{x}_{T+1/2}, \bar{y}_{T+1/2})$ of EG converges with*

$$\max_{y \in \mathcal{Y}} F(\bar{x}_{T+1/2}, y) - \min_{x \in \mathcal{X}} F(x, \bar{y}_{T+1/2}) \leq \frac{\ell D^2}{T}.$$

*This suggests $\mathcal{O}(1/\epsilon)$ gradient complexity is required to achieve $\epsilon$-saddle point.*

*Proof.* Recall $z_t = (x_t, y_t)$ and $\tilde{\nabla} F(z_t) = (\nabla_x F(x_t, y_t), -\nabla_y F(x_t, y_t))$. The EG updates in Algorithm 5 can be simplified to

$$\begin{aligned} z_{t+1/2} &= \Pi_{\mathcal{X} \times \mathcal{Y}}(z_t - \alpha \tilde{\nabla} F(z_t)), \\ z_{t+1} &= \Pi_{\mathcal{X} \times \mathcal{Y}}(z_t - \alpha \tilde{\nabla} F(z_{t+1/2})). \end{aligned} \tag{14}$$

By fact $(i)$ in Lemma A.1, we have that for any $z \in \mathcal{X} \times \mathcal{Y}$,

$$\begin{aligned} \|z_{t+1} - z\|^2 + \|z_{t+1} - z_t\|^2 - \|z_t - z\|^2 &= 2(z_{t+1} - z_t)^\top (z_{t+1} - z) \\ &\leq 2\alpha \tilde{\nabla} F(z_{t+1/2})^\top (z - z_{t+1}), \end{aligned}$$

where we use the optimality condition of the projection step such that $(\Pi_{\mathcal{C}}(u) - u)^\top (v - \Pi_{\mathcal{C}}(u)) \geq 0, \forall v \in \mathcal{C}$. For the same reason, we can obtain that

$$\begin{aligned} \|z_{t+1/2} - z_t\|^2 + \|z_{t+1/2} - z_{t+1}\|^2 - \|z_t - z_{t+1}\|^2 &= 2(z_{t+1/2} - z_t)^\top (z_{t+1/2} - z_{t+1}) \\ &\leq 2\alpha \tilde{\nabla} F(z_t)^\top (z_{t+1} - z_{t+1/2}). \end{aligned}$$

Summing up the above two inequalities, we get

$$\begin{aligned} \|z_{t+1} - z\|^2 &\leq \|z_t - z\|^2 - \|z_{t+1/2} - z_t\|^2 - \|z_{t+1/2} - z_{t+1}\|^2 + 2\alpha \tilde{\nabla} F(z_{t+1/2})^\top (z - z_{t+1}) \\ &\quad + 2\alpha \tilde{\nabla} F(z_t)^\top (z_{t+1} - z_{t+1/2}) \\ &= \|z_t - z\|^2 - \|z_{t+1/2} - z_t\|^2 - \|z_{t+1/2} - z_{t+1}\|^2 + 2\alpha \tilde{\nabla} F(z_{t+1/2})^\top (z - z_{t+1/2}) \\ &\quad + 2\alpha (\tilde{\nabla} F(z_t) - \tilde{\nabla} F(z_{t+1/2}))^\top (z_{t+1} - z_{t+1/2}). \end{aligned} \tag{15}$$

According to Lemma B.1, we can obtain

$$\begin{aligned} (\tilde{\nabla} F(z_t) - \tilde{\nabla} F(z_{t+1/2}))^\top (z_{t+1} - z_{t+1/2}) &\leq \ell \|z_t - z_{t+1/2}\| \|z_{t+1} - z_{t+1/2}\| \\ &\leq \frac{\ell}{2} \|z_t - z_{t+1/2}\|^2 + \frac{\ell}{2} \|z_{t+1} - z_{t+1/2}\|^2. \end{aligned}$$

Therefore, rearranging terms, by the choice of stepsize $\alpha \leq 1/\ell$, we have $\forall z \in \mathcal{X} \times \mathcal{Y}$,

$$
\begin{aligned}
\tilde{\nabla} F(z_{t+1/2})^\top (z_{t+1/2} - z) & \\
\leq \frac{1}{2\alpha} & (\|z_t - z\|^2 - \|z_{t+1} - z\|^2) - \frac{1}{2\alpha}\|z_{t+1/2} - z_t\|^2 - \frac{1}{2\alpha}\|z_{t+1/2} - z_{t+1}\|^2 \\
& + (\tilde{\nabla} F(z_t) - \tilde{\nabla} F(z_{t+1/2}))^\top (z_{t+1} - z_{t+1/2}) \\
\leq \frac{1}{2\alpha} & (\|z_t - z\|^2 - \|z_{t+1} - z\|^2) - \left(\frac{1}{2\alpha} - \frac{\ell}{2}\right)\|z_{t+1/2} - z_t\|^2 - \left(\frac{1}{2\alpha} - \frac{\ell}{2}\right)\|z_{t+1/2} - z_{t+1}\|^2 \\
\leq \frac{1}{2\alpha} & (\|z_t - z\|^2 - \|z_{t+1} - z\|^2).
\end{aligned}
\tag{16}
$$

Taking summation from $t = 0$ to $T - 1$, by Lemma B.2, we have

$$
\max_{y \in \mathcal{Y}} F(\bar{x}_{T+1/2}, y) - \min_{x \in \mathcal{X}} F(x, \bar{y}_{T+1/2}) \leq \frac{\|z_0 - z\|^2}{2\alpha T}.
$$

Since $\|z_0 - z\|^2 \leq 2D^2$ and $\alpha = 1/\ell$, the proof is complete. $\qquad\square$

The following results are motivated from Boob and Guzmán [16].

**Lemma B.9.** *Under Assumption 4.1. Let $z_{t+1} = (x_{t+1}, y_{t+1})$ be obtained through 1-step of EG update* (14) *given $z_t = (x_t, y_t)$, and $z'_{t+1}$ is obtained given $z'_t$. Setting $\alpha \leq 1/\ell$, then we have*

$$
\|z_{t+1} - z'_{t+1}\| \leq \|z_t - z'_t\| + 2L\ell^2\alpha^3.
$$

*Proof.* For any $z = (x, y) \in \mathcal{X} \times \mathcal{Y}$, we define an operator $P_{z_t}(\cdot) : \mathcal{X} \times \mathcal{Y} \to \mathcal{X} \times \mathcal{Y}$ as $P_{z_t}(z) = \Pi_{\mathcal{X} \times \mathcal{Y}}(z_t - \alpha\tilde{\nabla} F(z))$, and the EG updates can be written as $z_{t+1} = P_{z_t}(P_{z_t}(z_t))$. When the stepsize $\alpha \leq 1/\ell$, the operator $P_{z_t}(\cdot)$ is nonexpansive, i.e., $\forall z_1, z_2 \in \mathcal{X} \times \mathcal{Y}$,

$$
\begin{aligned}
\|P_{z_t}(z_1) - P_{z_t}(z_2)\| & \leq \alpha\|\tilde{\nabla} F(z_1) - \tilde{\nabla} F(z_2)\| \\
& \leq \alpha\ell\|z_1 - z_2\| \\
& \leq \|z_1 - z_2\|.
\end{aligned}
$$

Since the domain $\mathcal{X} \times \mathcal{Y}$ is a nonempty bounded closed convex set, by Theorem 4.19 in Bauschke et al. [12], the nonexpansive operator $P_{z_t}(\cdot)$ admits fixed points. Denote one fixed point as $u_t \in \mathcal{X} \times \mathcal{Y}$ such that $u_t = \Pi_{\mathcal{X} \times \mathcal{Y}}(z_t - \alpha\tilde{\nabla} F(u_t)) = P_{z_t}(u_t)$. The nonexpansiveness of $P_{z_t}(\cdot)$ implies

$$
\begin{aligned}
\|z_{t+1} - u_t\| = \|P_{z_t}(P_{z_t}(z_t)) - P_{z_t}(P_{z_t}(u_t))\| & \\
& \leq (\alpha\ell)^2\|z_t - u_t\| \\
& \leq \alpha^2\ell^2 \cdot \alpha\|\tilde{\nabla} F(u_t)\| \\
& \leq \alpha^3\ell^2 L.
\end{aligned}
\tag{17}
$$

The same holds true for $z'_{t+1}$ and $u'_t = P_{z'_t}(u'_t)$ defined for $z'_t$. As a result, we can obtain that

$$
\begin{aligned}
\|z_{t+1} - z'_{t+1}\| & \leq \|z_{t+1} - u_t\| + \|u_t - u'_t\| + \|u'_t - z'_{t+1}\| \\
& \leq \|u_t - u'_t\| + 2L\ell^2\alpha^3.
\end{aligned}
\tag{18}
$$

By optimality conditions of $u_t = \Pi_{\mathcal{X} \times \mathcal{Y}}(z_t - \alpha\tilde{\nabla} F(u_t))$ and $u'_t = \Pi_{\mathcal{X} \times \mathcal{Y}}(z'_t - \alpha\tilde{\nabla} F(u'_t))$, we obtain that for any $z, z' \in \mathcal{X} \times \mathcal{Y}$,

$$
\begin{aligned}
(u_t - z_t + \alpha\tilde{\nabla} F(u_t))^\top (z - u_t) & \geq 0, \\
(u'_t - z'_t + \alpha\tilde{\nabla} F(u'_t))^\top (z' - u'_t) & \geq 0.
\end{aligned}
$$

Taking $z = u'_t$ and $z' = u_t$ and using the fact that $\tilde{\nabla} F$ is monotone by Lemma B.1, we obtain that

$$
\begin{aligned}
\|u_t - u'_t\|^2 & \leq (u_t - u'_t)^\top (z_t - z'_t) - \alpha(\tilde{\nabla} F(u_t) - \tilde{\nabla} F(u'_t))^\top (u_t - u'_t) \\
& \leq \|u_t - u'_t\|\|z_t - z'_t\|.
\end{aligned}
$$

Combined with (18), the proof is complete since $\|u_t - u'_t\| \leq \|z_t - z'_t\|$. $\qquad\square$

**Remark 6.** We can alternatively derive the relation between $\|z_{t+1} - z'_{t+1}\|$ and $\|z_t - z'_t\|$ as follows:

$$
\begin{aligned}
\|z_{t+1} - &z'_{t+1}\|^2 \\
&\leq \|z_t - z'_t\|^2 - 2\alpha(z_t - z'_t)^\top(\tilde\nabla F(z_{t+1/2}) - \tilde\nabla F(z'_{t+1/2})) + \alpha^2\|\tilde\nabla F(z_{t+1/2}) - \tilde\nabla F(z'_{t+1/2})\|^2 \\
&\leq \|z_t - z'_t\|^2 + 2\alpha\ell\|z_t - z'_t\|\|z_{t+1/2} - z'_{t+1/2}\| + \alpha^2\ell^2\|z_{t+1/2} - z'_{t+1/2}\|^2 \\
&\leq (1 + 2\alpha\ell\sqrt{1 + \alpha^2\ell^2} + \alpha^2\ell^2(1 + \alpha^2\ell^2))\|z_t - z'_t\|^2 \\
&= \left(1 + \alpha\ell\sqrt{1 + \alpha^2\ell^2}\right)^2 \|z_t - z'_t\|^2.
\end{aligned}
$$

Here, we use Lemma B.1 and B.5. The above results will lead to reproducibility that grows with $\mathcal{O}(e^T)$, which is similar to the results of AGD for the minimization setting [6].

### B.3.2 Inexact Initialization Oracle

**Theorem B.10** (Restate Theorem 4.3, part $(i)$). *Under Assumptions 4.1. The average iterate $(\bar{x}_{T+1/2}, \bar{y}_{T+1/2})$ of EG satisfies $\max_{y\in\mathcal{Y}} F(\bar{x}_{T+1/2}, y) - \min_{x\in\mathcal{X}} F(x, \bar{y}_{T+1/2}) \leq \mathcal{O}(\epsilon)$ with complexity $T = \mathcal{O}(1/\epsilon)$ if setting stepsize $\alpha = 1/\ell$. Furthermore, the reproducibility, i.e., $(\epsilon, \delta)$-deviation between outputs of two independent runs of EG given different initialization is $\|\bar{x}_{T+1/2} - \bar{x}'_{T+1/2}\|^2 + \|\bar{y}_{T+1/2} - \bar{y}'_{T+1/2}\|^2 \leq \mathcal{O}(\min\{\delta^2 e^{1/\epsilon}, \delta^2 + 1/\epsilon^2, D^2\})$.*

*Proof.* The convergence part directly follows from Lemma B.8 with $T = c/\epsilon$ for some constant $c \geq \ell D^2$. For reproducibility, by Lemma B.5, B.9 and the stepsize $\alpha = 1/\ell$, we have that for $t = 1, 2, \cdots, T - 1$,

$$
\begin{aligned}
\|x_{t+1/2} - x'_{t+1/2}\|^2 + \|y_{t+1/2} - y'_{t+1/2}\|^2 &\leq (1 + \alpha^2\ell^2)\left(\|x_t - x'_t\|^2 + \|y_t - y'_t\|^2\right) \\
&\leq 2(\|z_0 - z'_0\| + 2L\ell^2\alpha^3 t)^2 \\
&\leq 2\left(\delta + \frac{2L}{\ell}t\right)^2.
\end{aligned}
$$

The above also holds for $t = 0$. Therefore, by Jensen's inequality, we obtain

$$
\begin{aligned}
\|\bar{x}_{T+1/2} - \bar{x}'_{T+1/2}\|^2 + \|\bar{y}_{T+1/2} - \bar{y}'_{T+1/2}\|^2 &\leq \frac{1}{T}\sum_{t=0}^{T-1}\left(\|x_{t+1/2} - x'_{t+1/2}\|^2 + \|y_{t+1/2} - y'_{t+1/2}\|^2\right) \\
&\leq \frac{2}{T}\sum_{t=0}^{T-1}\left(\delta + \frac{2L}{\ell}t\right)^2 \\
&\leq 4\delta^2 + \frac{16L^2}{3\ell^2}T^2.
\end{aligned}
$$

Alternatively, by Remark 6, we know that $\|z_{t+1/2} - z'_{t+1/2}\|^2 \leq 2(1 + \sqrt{2})^{2t}\delta^2$, and thus the reproducibility is $\|\bar{x}_{T+1/2} - \bar{x}'_{T+1/2}\|^2 + \|\bar{y}_{T+1/2} - \bar{y}'_{T+1/2}\|^2 \leq \mathcal{O}(e^T\delta^2)$. The proof is complete by taking the minimum between the two results and replacing $T$ with $c/\epsilon$. $\qquad\square$

### B.3.3 Inexact Deterministic Gradient Oracle

When only given inexact gradient $(G_x(x_t, y_t), G_y(x_t, y_t))$, the updates of EG becomes

$$
\begin{aligned}
z_{t+1/2} &= \Pi_{\mathcal{X}\times\mathcal{Y}}(z_t - \alpha\tilde{G}(z_t)), \\
z_{t+1} &= \Pi_{\mathcal{X}\times\mathcal{Y}}(z_t - \alpha\tilde{G}(z_{t+1/2})),
\end{aligned}
$$

where exact gradients $\tilde\nabla F(z_t)$ in (14) are replaced by $\tilde{G}(z_t) = (G_x(x_t, y_t), -G_y(x_t, y_t))$.

**Theorem B.11** (Restate Theorem 4.3, part $(ii)$). *Under Assumptions 4.1. Given an inexact deterministic gradient oracle in Definition 4 with $\delta \leq \mathcal{O}(\epsilon)$. The average iterate $(\bar{x}_{T+1/2}, \bar{y}_{T+1/2})$ of EG satisfies $\max_{y\in\mathcal{Y}} F(\bar{x}_{T+1/2}, y) - \min_{x\in\mathcal{X}} F(x, \bar{y}_{T+1/2}) \leq \mathcal{O}(\epsilon)$ with complexity $T = \mathcal{O}(1/\epsilon)$ if setting stepsize $\alpha = 1/\ell$. Furthermore, the reproducibility is $\mathcal{O}(\min\{\delta^2 e^{1/\epsilon}, 1/\epsilon^2, D^2\})$.*

*Proof.* Let $\Delta(z_t) = \tilde{G}(z_t) - \tilde{\nabla}F(z_t)$. We know $\|\Delta(z_t)\| \leq \delta$ by Definition 4. Using (15) in the proof of Lemma B.8, we have that $\forall z \in \mathcal{X} \times \mathcal{Y}$,

$$
\begin{aligned}
\|z_{t+1} - z\|^2 &\leq \|z_t - z\|^2 - \|z_{t+1/2} - z_t\|^2 - \|z_{t+1/2} - z_{t+1}\|^2 + 2\alpha \tilde{G}(z_{t+1/2})^\top (z - z_{t+1/2}) \\
&\quad + 2\alpha(\tilde{G}(z_t) - \tilde{G}(z_{t+1/2}))^\top (z_{t+1} - z_{t+1/2}) \\
&= \|z_t - z\|^2 - \|z_{t+1/2} - z_t\|^2 - \|z_{t+1/2} - z_{t+1}\|^2 + 2\alpha \tilde{\nabla}F(z_{t+1/2})^\top (z - z_{t+1/2}) \\
&\quad + 2\alpha(\tilde{\nabla}F(z_t) - \tilde{\nabla}F(z_{t+1/2}))^\top (z_{t+1} - z_{t+1/2}) + 2\alpha \Delta(z_{t+1/2})^\top (z - z_{t+1/2}) \\
&\quad + 2\alpha(\Delta(z_t) - \Delta(z_{t+1/2}))^\top (z_{t+1} - z_{t+1/2}) \\
&\leq \|z_t - z\|^2 - \|z_{t+1/2} - z_t\|^2 - \|z_{t+1/2} - z_{t+1}\|^2 + 2\alpha \tilde{\nabla}F(z_{t+1/2})^\top (z - z_{t+1/2}) \\
&\quad + 2\alpha(\tilde{\nabla}F(z_t) - \tilde{\nabla}F(z_{t+1/2}))^\top (z_{t+1} - z_{t+1/2}) + 6\sqrt{2}\alpha\delta D.
\end{aligned}
\tag{19}
$$

The above is the same as (15) up to an additional error in $\mathcal{O}(\delta)$. Following the same proof after (15), with $\alpha = 1/\ell$, we obtain that

$$
\max_{y \in \mathcal{Y}} F(\bar{x}_{T+1/2}, y) - \min_{x \in \mathcal{X}} F(x, \bar{y}_{T+1/2}) \leq \frac{\ell D^2}{T} + 3\sqrt{2}\delta D.
$$

When $\delta \leq \epsilon/(6\sqrt{2}D)$ and $T = c/\epsilon$ for some constant $c \geq 2\ell D^2/\epsilon$, we get $\epsilon$-saddle point.

We then show the reproducibility guarantee. Let $u_t = \Pi_{\mathcal{X} \times \mathcal{Y}}(z_t - \alpha \tilde{\nabla}F(u_t))$ be the same as in the proof of Lemma B.9. Similarly to (17), we have that

$$
\begin{aligned}
\|z_{t+1} - u_t\| &\leq \alpha\|\tilde{G}(z_{t+1/2}) - \tilde{\nabla}F(u_t)\| \\
&\leq \alpha\|\tilde{\nabla}F(z_{t+1/2}) - \tilde{\nabla}F(u_t)\| + \alpha\|\tilde{G}(z_{t+1/2}) - \tilde{\nabla}F(z_{t+1/2})\| \\
&\leq \alpha\ell\|z_{t+1/2} - u_t\| + \alpha\delta \\
&\leq \alpha^2\ell\|\tilde{G}(z_t) - \tilde{\nabla}F(u_t)\| + \alpha\delta \\
&\leq \alpha^2\ell^2\|z_t - u_t\| + (1 + \alpha\ell)\alpha\delta \\
&\leq \alpha^3\ell^2 L + (1 + \alpha\ell)\alpha\delta.
\end{aligned}
$$

As a result, the same as (18), since $\alpha = 1/\ell$, we can obtain that $\forall t = 0, 1, \cdots, T-1$,

$$
\begin{aligned}
\|z_t - z_t'\| &\leq \|z_{t-1} - z_{t-1}'\| + 2\alpha^3\ell^2 L + 2(1 + \alpha\ell)\alpha\delta \\
&\leq t(2\alpha^3\ell^2 L + 2(1 + \alpha\ell)\alpha\delta) \\
&\leq \frac{2t}{\ell}(L + 2\delta).
\end{aligned}
$$

Therefore, by Jensen's inequality and (12) in Section B.2.3 for the guarantee of GDA, we know

$$
\begin{aligned}
\|\bar{x}_{T+1/2} - \bar{x}_{T+1/2}'\|^2 + \|\bar{y}_{T+1/2} - \bar{y}_{T+1/2}'\|^2 &\leq \frac{1}{T} \sum_{t=0}^{T-1} \left( \|x_{t+1/2} - x_{t+1/2}'\|^2 + \|y_{t+1/2} - y_{t+1/2}'\|^2 \right) \\
&\leq \frac{1}{T} \sum_{t=0}^{T-1} 2\left( (1 + \alpha^2\ell^2)\|z_t - z_t'\|^2 + 4\alpha^2\delta^2 \right) \\
&\leq \frac{1}{T} \sum_{t=0}^{T-1} \frac{8}{\ell^2} \left( 2t^2(L + 2\delta)^2 + \delta^2 \right) \\
&\leq \frac{128}{3\ell^2}\delta^2 T^2 + \frac{32L^2}{3\ell^2}T^2 + \frac{8}{\ell^2}\delta^2.
\end{aligned}
$$

Note that $T = c/\epsilon$ and $\delta \leq \mathcal{O}(\epsilon)$. Thus the reproducibility is $\mathcal{O}(1/\epsilon^2)$. Alternatively, by Remark 6 and similarly to (12), we have that

$$
\begin{aligned}
\|z_{t+1} &- z'_{t+1}\| \\
&\leq \sqrt{\|z_t - z'_t\|^2 + 2\alpha\ell\|z_t - z'_t\|\|z_{t+1/2} - z'_{t+1/2}\| + \alpha^2\ell^2\|z_{t+1/2} - z'_{t+1/2}\| + 2\alpha\delta} \\
&\leq \sqrt{\left(1 + \alpha\ell\sqrt{1 + \alpha^2\ell^2}\right)^2\|z_t - z'_t\|^2 + 4\alpha^2\ell\delta\left(1 + \alpha\ell\sqrt{1 + \alpha^2\ell^2}\right)\|z_t - z'_t\| + 4\alpha^4\ell^2\delta^2} + 2\alpha\delta \\
&= \left(1 + \alpha\ell\sqrt{1 + \alpha^2\ell^2}\right)\|z_t - z'_t\| + 2\alpha\delta(1 + \alpha\ell) \\
&= (1 + \sqrt{2})\|z_t - z'_t\| + \frac{4\delta}{\ell}.
\end{aligned}
$$

Thus $\|z_{t+1/2} - z'_{t+1/2}\| \leq \sqrt{2}\|z_t - z'_t\| + 2\delta/\ell \leq \mathcal{O}(e^T\delta/\ell)$ and the reproducibility is $\mathcal{O}(\delta^2 e^{1/\epsilon})$. The proof is complete by taking the minimum between the two results. $\qquad\square$

### B.3.4 More Discussions

In this section, we show that Extragradient can also be optimally reproducible by a different selection of parameters. Although it will suffer from a sub-optimal convergence rate $\mathcal{O}(1/\epsilon^{3/2})$ instead of $\mathcal{O}(1/\epsilon)$, this is still an improvement on the $\mathcal{O}(1/\epsilon^2)$ rate of GDA.

**Theorem B.12.** *Under Assumptions 4.1. The average iterate $(\bar{x}_{T+1/2}, \bar{y}_{T+1/2})$ of EG satisfies $\max_{y\in\mathcal{Y}} F(\bar{x}_{T+1/2}, y) - \min_{x\in\mathcal{X}} F(x, \bar{y}_{T+1/2}) \leq \mathcal{O}(\epsilon)$ with complexity $T = \mathcal{O}(1/(\delta^{1/2}\epsilon^{3/2}))$ if setting stepsize $\alpha = \min\{1/\ell, (\delta/(2\ell^2 T))^{1/3}\}$. The reproducibility is $\mathcal{O}(\delta^2)$.*

*Proof.* The same as Section B.3.2, by the choice of stepsize $\alpha$ such that $\alpha^3 T \leq \delta/2\ell^2$, we obtain

$$
\begin{aligned}
\|\bar{x}_{T+1/2} - \bar{x}'_{T+1/2}\|^2 + \|\bar{y}_{T+1/2} - \bar{y}'_{T+1/2}\|^2 &\leq \frac{2}{T}\sum_{t=0}^{T-1}\left(\delta + 2L\ell^2\alpha^3 t\right)^2 \\
&\leq 4\delta^2 + 4(2L\ell^2\alpha^3 T)^2 \\
&\leq 4(L^2 + 1)\delta^2.
\end{aligned}
$$

By Lemma B.8, when the stepsize $\alpha \leq 1/\ell$, we have that

$$
\begin{aligned}
\max_{y\in\mathcal{Y}} F(\bar{x}_{T+1/2}, y) - \min_{x\in\mathcal{X}} F(x, \bar{y}_{T+1/2}) &\leq \frac{D^2}{\alpha T} \\
&\leq \frac{\ell D^2}{T} + \frac{D^2(2\ell^2/\delta)^{1/3}}{T^{2/3}}.
\end{aligned}
$$

This means a $\mathcal{O}(1/(\delta^{1/2}\epsilon^{3/2}))$ convergence rate with reproducibility $\mathcal{O}(\delta^2)$. In the case $\delta = \mathcal{O}(1)$, the gradient complexity is $\mathcal{O}(1/\epsilon^{3/2})$. $\qquad\square$

**Theorem B.13.** *Under Assumptions 4.1. Given an inexact deterministic gradient oracle in Definition 4 with $\delta \leq \mathcal{O}(\epsilon)$. The average iterate $(\bar{x}_{T+1/2}, \bar{y}_{T+1/2})$ of EG satisfies $\max_{y\in\mathcal{Y}} F(\bar{x}_{T+1/2}, y) - \min_{x\in\mathcal{X}} F(x, \bar{y}_{T+1/2}) \leq \mathcal{O}(\epsilon)$ with complexity $T = \mathcal{O}(1/(\epsilon\sqrt{\delta}))$ if setting stepsize $\alpha = \min\{1/\ell, (\delta/(2\ell^2))^{1/2}\}$. The reproducibility is $\mathcal{O}(\delta^2/\epsilon^2)$.*

*Proof.* The same as Section B.3.3, since $\alpha\ell \leq 1$ and $\alpha^2 \leq \delta/(2\ell^2)$, we have that

$$
\begin{aligned}
\|\bar{x}_{T+1/2} - \bar{x}'_{T+1/2}\|^2 + \|\bar{y}_{T+1/2} - \bar{y}'_{T+1/2}\|^2 &\leq 8\alpha^2\delta^2 + \frac{4}{T}\sum_{t=0}^{T-1}(2\alpha^3\ell^2 L + 4\alpha\delta)^2 t^2 \\
&\leq 8\left((2L\ell^2\alpha^2)^2\alpha^2 T^2 + 8\delta^2\alpha^2 T^2 + \delta^2\alpha^2\right) \\
&\leq 8(L^2 + 9)\delta^2(\alpha T)^2.
\end{aligned}
$$

When the stepsize $\alpha \leq 1/\ell$, we also have

$$\max_{y \in \mathcal{Y}} F(\bar{x}_{T+1/2}, y) - \min_{x \in \mathcal{X}} F(x, \bar{y}_{T+1/2}) \leq \frac{D^2}{\alpha T} + 3\sqrt{2}\delta D$$

$$\leq \frac{\ell D^2}{T} + \frac{\sqrt{2}\ell D^2}{T\sqrt{\delta}} + 3\sqrt{2}\delta D.$$

To guarantee $\mathcal{O}(\epsilon)$-saddle point, we need to ensure $\delta \leq \mathcal{O}(\epsilon)$ and $\alpha T = c/\epsilon$ for some constant $c$. This means a $\mathcal{O}(1/(\epsilon\sqrt{\delta}))$ convergence rate with reproducibility $\mathcal{O}(\delta^2/\epsilon^2)$. Since $\delta \leq \mathcal{O}(\epsilon)$, the gradient complexity is $\mathcal{O}(1/\epsilon^{3/2})$. $\qquad\square$

Finally, we want to mention that the analysis can also be extended to reproducibility under stochastic gradient oracle and stability of Extragradient [16] that matches with SGDA. We will not provide all details here. The key is to select stepsize $\alpha$ to balance the convergence $\mathcal{O}(1/(\alpha T))$ in Lemma B.8 and the error term $\mathcal{O}(\alpha^3 T)$ that appears according to Lemma B.9. Moreover, we also acknowledge that it is unclear whether the analysis of EG is tight since the specific lower-bound is unknown. We leave this problem for future exploration.

## C    Near-optimal Guarantees in the Minimax Case

This section discusses near-optimal guarantees for algorithmic reproducibility and gradient complexity in smooth convex-concave minimax optimization.

### C.1    Useful Lemmas

We first establish the convergence behavior of gradient descent ascent (GDA) and Extragradient (EG) [48] for smooth and strongly-convex–strongly-concave (SC-SC) functions under the inexact gradient oracle in Definition 4. For the sake of simplicity and to enable a general analysis, we slightly abuse notation here to consider the minimax optimization problem

$$\min_{x \in \mathcal{X}} \max_{y \in \mathcal{Y}} f(x, y),$$

where $f : \mathcal{X} \times \mathcal{Y} \to \mathbb{R}$ satisfies the following assumption.

**Assumption C.1.** *The function $f(x, y)$ is $\ell$-smooth and $\mu$–strongly-convex–strongly-concave on the closed convex domain $\mathcal{X} \times \mathcal{Y}$.*

**Assumption C.2.** *We assume the existence of an inexact gradient oracle that returns a vector $g(x, y) = (g_x(x, y), g_y(x, y))$ at any querying point $(x, y) \in \mathcal{X} \times \mathcal{Y}$ such that $\|\nabla f(x, y) - g(x, y)\|^2 \leq \delta^2$ where $\nabla f(x, y) = (\nabla_x f(x, y), \nabla_y f(x, y))$ is the true gradient at $(x, y)$.*

The lemma below shows the convergence behavior of GDA under the inexact gradient oracle presented above, also referred to as Inexact-GDA.

**Lemma C.3.** *Under Assumption C.1. Let $z^* = (x^*, y^*) \in \mathcal{X} \times \mathcal{Y}$ be the unique saddle point of $f(x, y)$ and $\kappa := \ell/\mu$ be the condition number. Given an inexact gradient oracle in Assumption C.2. Denote $z_t = (x_t, y_t)$ and $\tilde{g}(z_t) = (g_x(x_t, y_t), -g_y(x_t, y_t))$. Starting from $z_0 \in \mathcal{X} \times \mathcal{Y}$, GDA that updates for $t = 0, 1, \cdots, T - 1$,*

$$z_{t+1} = \Pi_{\mathcal{X} \times \mathcal{Y}}(z_t - \alpha\tilde{g}(z_t)), \qquad \text{(Inexact-GDA)}$$

*with stepsize $\alpha = \mu/(4\ell^2)$ converges with*

$$\|z_T - z^*\|^2 \leq \exp\left(-\frac{T}{8\kappa^2}\right) \|z_0 - z^*\|^2 + \left(\frac{1}{\ell^2} + \frac{2}{\mu^2}\right) \delta^2.$$

*Proof.* Let $\tilde{\nabla} f(z_t) = (\nabla_x f(x_t, y_t), -\nabla_y f(x_t, y_t))$. It holds that $z^* = \Pi_{\mathcal{X} \times \mathcal{Y}}(z^* - \alpha\tilde{\nabla} f(z^*))$ since the saddle point problem and the projection problem share the same optimality condition when $f(x, y)$ is convex-concave (see Proposition 1.4.2 in Facchinei and Pang [32]) such that

$$\tilde{\nabla} f(z^*)^\top (z - z^*) \geq 0, \quad \forall z = (x, y) \in \mathcal{X} \times \mathcal{Y}.$$

Therefore, similarly to (11), by the GDA updates, we have

$$\|z_{t+1} - z^*\|^2 = \|\Pi_{\mathcal{X}\times\mathcal{Y}}(z_t - \alpha\tilde{g}(z_t)) - \Pi_{\mathcal{X}\times\mathcal{Y}}(z^* - \alpha\tilde{\nabla}f(z^*))\|^2$$
$$\leq \|(z_t - z^*) - \alpha(\tilde{g}(z_t) - \tilde{\nabla}f(z^*))\|^2$$
$$= \|z_t - z^*\|^2 - 2\alpha(\tilde{g}(z_t) - \tilde{\nabla}f(z^*))^\top(z_t - z^*) + \alpha^2\|\tilde{g}(z_t) - \tilde{\nabla}f(z^*)\|^2.$$

Since $\tilde{\nabla}f$ is $\mu$–strongly-monotone if $f(x,y)$ is $\mu$–strongly-convex–strongly-concave [66, 35], i.e., $\forall z_1, z_2 \in \mathcal{X} \times \mathcal{Y}, (\tilde{\nabla}f(z_1) - \tilde{\nabla}f(z_2))^\top(z_1 - z_2) \geq \mu\|z_1 - z_2\|^2$, we have that

$$(\tilde{g}(z_t) - \tilde{\nabla}f(z^*))^\top(z_t - z^*) = (\tilde{\nabla}f(z_t) - \tilde{\nabla}f(z^*))^\top(z_t - z^*) + (\tilde{g}(z_t) - \tilde{\nabla}f(z_t))^\top(z_t - z^*)$$
$$\geq \mu\|z_t - z^*\|^2 - \delta\|z_t - z^*\|$$
$$\geq \frac{\mu}{2}\|z_t - z^*\|^2 - \frac{\delta^2}{2\mu},$$

where we use Assumption C.2 such that $\|\tilde{g}(z_t) - \tilde{\nabla}f(z_t)\| \leq \delta$ and fact $(iii)$ in Lemma A.1. Then by $\ell$-smoothness of $f(x,y)$, we can obtain that

$$\|\tilde{g}(z_t) - \tilde{\nabla}f(z^*)\|^2 \leq 2\|\tilde{g}(z_t) - \tilde{\nabla}f(z_t)\|^2 + 2\|\tilde{\nabla}f(z_t) - \tilde{\nabla}f(z^*)\|^2$$
$$\leq 2\delta^2 + 2\ell^2\|z_t - z^*\|^2.$$

Combining all three results together, when choosing the stepsize $\alpha = \mu/(4\ell^2)$, we get that

$$\|z_{t+1} - z^*\|^2 \leq (1 - \alpha\mu + 2\alpha^2\ell^2)\|z_t - z^*\|^2 + \left(\frac{\alpha}{\mu} + 2\alpha^2\right)\delta^2$$
$$= \left(1 - \frac{1}{8\kappa^2}\right)\|z_t - z^*\|^2 + \frac{\delta^2}{4\ell^2}\left(1 + \frac{1}{2\kappa^2}\right). \tag{20}$$

Unrolling the recursion, we thus obtain

$$\|z_T - z^*\|^2$$
$$\leq \left(1 - \frac{1}{8\kappa^2}\right)^T\|z_0 - z^*\|^2 + \frac{\delta^2}{4\ell^2}\left(1 + \frac{1}{2\kappa^2}\right)\left(1 + \left(1 - \frac{1}{8\kappa^2}\right) + \cdots + \left(1 - \frac{1}{8\kappa^2}\right)^{T-1}\right)$$
$$\leq \exp\left(-\frac{T}{8\kappa^2}\right)\|z_0 - z^*\|^2 + \left(\frac{1}{\ell^2} + \frac{2}{\mu^2}\right)\delta^2.$$

This means a $\mathcal{O}(\kappa^2)$ convergence rate to a $\mathcal{O}(\delta^2)$ neighborhood, where $\kappa = \ell/\mu$ is the condition number. □

The lemma below establishes the convergence performance of EG under Assumption C.1 and C.2.

**Lemma C.4.** *Under Assumption C.1. Let $z^* = (x^*, y^*) \in \mathcal{X} \times \mathcal{Y}$ be the unique saddle point of $f(x,y)$ and $\kappa := \ell/\mu$ be the condition number. Given an inexact gradient oracle in Assumption C.2. Denote $z_t = (x_t, y_t)$ and $\tilde{g}(z_t) = (g_x(x_t, y_t), -g_y(x_t, y_t))$. Starting from $z_0 \in \mathcal{X} \times \mathcal{Y}$, Extragradient that updates for $t = 0, 1, \cdots, T - 1$,*

$$z_{t+1/2} = \Pi_{\mathcal{X}\times\mathcal{Y}}(z_t - \alpha\tilde{g}(z_t)),$$
$$z_{t+1} = \Pi_{\mathcal{X}\times\mathcal{Y}}(z_t - \alpha\tilde{g}(z_{t+1/2})), \tag{Inexact-EG}$$

*with stepsize $\alpha = 1/(2\ell)$ converges with*

$$\|z_T - z^*\|^2 \leq \exp\left(-\frac{T}{8\kappa}\right)\|z_0 - z^*\|^2 + \frac{8\delta^2}{\mu}\left(\frac{2}{\ell} + \frac{1}{\mu}\right).$$

*Proof.* Let $\tilde{\nabla}f(z_t) = (\nabla_x f(x_t, y_t), -\nabla_y f(x_t, y_t))$ and $\Delta(z_t) = \tilde{g}(z_t) - \tilde{\nabla}f(z_t)$. By (15) in the proof of Lemma B.8, setting $z = z^*$, we have that,

$$\|z_{t+1} - z^*\|^2 \leq \|z_t - z^*\|^2 - \|z_{t+1/2} - z_t\|^2 - \|z_{t+1/2} - z_{t+1}\|^2 + 2\alpha\tilde{\nabla}f(z_{t+1/2})^\top(z^* - z_{t+1/2})$$
$$+ 2\alpha(\tilde{\nabla}f(z_t) - \tilde{\nabla}f(z_{t+1/2}))^\top(z_{t+1} - z_{t+1/2}) + 2\alpha\Delta(z_{t+1/2})^\top(z^* - z_{t+1/2})$$
$$+ 2\alpha(\Delta(z_t) - \Delta(z_{t+1/2}))^\top(z_{t+1} - z_{t+1/2}). \tag{21}$$

By strong-convexity-strong-concavity of the function $f(x,y)$, we know that

$$f(x^*, y_{t+1/2}) \geq f(x_{t+1/2}, y_{t+1/2}) + \nabla_x f(x_{t+1/2}, y_{t+1/2})^\top (x^* - x_{t+1/2}) + \frac{\mu}{2}\|x_{t+1/2} - x^*\|^2,$$

$$-f(x_{t+1/2}, y^*) \geq -f(x_{t+1/2}, y_{t+1/2}) - \nabla_y f(x_{t+1/2}, y_{t+1/2})^\top (y^* - y_{t+1/2}) + \frac{\mu}{2}\|y_{t+1/2} - y^*\|^2.$$

Summing up the above two inequalities, using the definition of saddle points, we have

$$
\begin{aligned}
&\tilde{\nabla} f(z_{t+1/2})^\top (z^* - z_{t+1/2}) + \Delta(z_{t+1/2})^\top (z^* - z_{t+1/2}) \\
&\quad \leq f(x^*, y_{t+1/2}) - f(x_{t+1/2}, y^*) - \frac{\mu}{2}\|z_{t+1/2} - z^*\|^2 + \Delta(z_{t+1/2})^\top (z^* - z_{t+1/2}) \\
&\quad \leq -\frac{\mu}{2}\|z_{t+1/2} - z^*\|^2 + \|\Delta(z_{t+1/2})\|\|z^* - z_{t+1/2}\| \\
&\quad \leq -\frac{\mu}{4}\|z_{t+1/2} - z^*\|^2 + \frac{\delta^2}{\mu} \\
&\quad \leq -\frac{\mu}{8}\|z_t - z^*\|^2 + \frac{\mu}{4}\|z_t - z_{t+1/2}\|^2 + \frac{\delta^2}{\mu},
\end{aligned}
\tag{22}
$$

where we use fact $(iii)$ in Lemma A.1 and $\|z_t - z^*\|^2 \leq 2\|z_t - z_{t+1/2}\|^2 + 2\|z_{t+1/2} - z^*\|^2$. By smoothness of $f(x,y)$ and fact $(iii)$ in Lemma A.1, we also have that

$$
\begin{aligned}
&(\tilde{\nabla} f(z_t) - \tilde{\nabla} f(z_{t+1/2}))^\top (z_{t+1} - z_{t+1/2}) + (\Delta(z_t) - \Delta(z_{t+1/2}))^\top (z_{t+1} - z_{t+1/2}) \\
&\quad \leq \ell\|z_t - z_{t+1/2}\|\|z_{t+1} - z_{t+1/2}\| + 2\delta \cdot \|z_{t+1} - z_{t+1/2}\| \\
&\quad \leq \frac{\ell}{2}\|z_t - z_{t+1/2}\|^2 + \ell\|z_{t+1} - z_{t+1/2}\|^2 + \frac{2\delta^2}{\ell}.
\end{aligned}
\tag{23}
$$

Plugging (22) and (23) back into (21), choosing $\alpha = 1/(2\ell)$, we obtain that

$$
\begin{aligned}
\|z_{t+1} - z^*\|^2 &\leq \left(1 - \frac{\mu\alpha}{4}\right)\|z_t - z^*\|^2 - \left(1 - \frac{\mu\alpha}{2} - \alpha\ell\right)\|z_{t+1/2} - z_t\|^2 \\
&\quad - (1 - 2\alpha\ell)\|z_{t+1/2} - z_{t+1}\|^2 + 2\alpha\delta^2\left(\frac{2}{\ell} + \frac{1}{\mu}\right) \\
&\leq \left(1 - \frac{\mu}{8\ell}\right)\|z_t - z^*\|^2 + \frac{\delta^2}{\ell}\left(\frac{2}{\ell} + \frac{1}{\mu}\right).
\end{aligned}
\tag{24}
$$

Unrolling the recursion, since $1 + \eta \leq e^\eta, \forall \eta \in \mathbb{R}$, we get that

$$
\begin{aligned}
\|z_T - z^*\|^2 &\leq \left(1 - \frac{\mu}{8\ell}\right)^T \|z_0 - z^*\|^2 + \frac{\delta^2}{\ell}\left(\frac{2}{\ell} + \frac{1}{\mu}\right)\left(1 + \left(1 - \frac{\mu}{8\ell}\right) + \cdots + \left(1 - \frac{\mu}{8\ell}\right)^{T-1}\right) \\
&\leq \left(1 - \frac{\mu}{8\ell}\right)^T \|z_0 - z^*\|^2 + \frac{8\delta^2}{\mu}\left(\frac{2}{\ell} + \frac{1}{\mu}\right).
\end{aligned}
$$

This means a $\mathcal{O}(\kappa)$ convergence rate to a $\mathcal{O}(\delta^2)$ neighborhood, where $\kappa = \ell/\mu$ is the condition number. $\qquad\square$

Lemma 4.5 directly follows from Lemma C.4 observing that $G(x, y) + r(x - x_0, y_0 - y)$ is a $\delta$-inexact gradient of $F_r(x, y)$. Next, we provide a useful lemma showing how to satisfy the stopping criteria for the auxiliary smooth SC-SC sub-problem in Algorithm 2 when presented with inexact gradients. The results are motivated from Yang et al. [74].

**Lemma C.5.** *Under Assumption C.1 and C.2. Suppose the domain $\mathcal{X}$ and $\mathcal{Y}$ have a diameter of $D$. Denote $z^* = (x^*, y^*)$ be the unique saddle point of $f(x, y)$. For any $\hat{z} = (\hat{x}, \hat{y}) \in \mathcal{X} \times \mathcal{Y}$, we let $\tilde{g}(\hat{z}) = (g_x(\hat{x}, \hat{y}), -g_y(\hat{x}, \hat{y}))$ and define $[\hat{z}]_\beta = ([\hat{x}]_\beta, [\hat{y}]_\beta)$ for $\beta \geq 2\ell$ to be*

$$[\hat{z}]_\beta = \Pi_{\mathcal{X} \times \mathcal{Y}}\left(\hat{z} - \frac{1}{\beta}\tilde{g}(\hat{z})\right),$$

*which is obtained through one step of GDA starting from $\hat{z}$ with inexact gradients. Denote the true gradient as $\tilde{\nabla} f([\hat{z}]_\beta) = (\nabla_x f([\hat{x}]_\beta, [\hat{y}]_\beta), -\nabla_y f([\hat{x}]_\beta, [\hat{y}]_\beta))$. Then we have that $\forall z = (x, y) \in \mathcal{X} \times \mathcal{Y}$,*

$$\tilde{\nabla} f([\hat{z}]_\beta)^\top ([\hat{z}]_\beta - z) \leq 2\sqrt{2}\beta D\|\hat{z} - z^*\| + \sqrt{2}\delta D\left((2 + \sqrt{2})\sqrt{\frac{\beta}{\mu}} + 3\right).$$

*Moreover, it also holds that $\|[\hat{z}]_\beta - z^*\| \le (1 + \sqrt{2}\ell/\beta)\|\hat{z} - z^*\| + \delta(1/\sqrt{\beta\mu} + \sqrt{2}/\beta)$.*

*Proof.* We construct a "ghost" point $\hat{z}_1 = (\hat{x}_1, \hat{y}_1) \in \mathcal{X} \times \mathcal{Y}$ to be

$$\hat{z}_1 = \Pi_{\mathcal{X} \times \mathcal{Y}}\left(\hat{z} - \frac{1}{\beta}\tilde{g}([\hat{z}]_\beta)\right).$$

$\hat{z}_1$ can be regarded as performing one update of inexact-EG with stepsize $1/\beta$ starting from $\hat{z}$. Therefore, by (16) and (19) in the convergence analysis of EG, since $1/\beta \le 1/\ell$, we obtain that $\forall z = (x, y) \in \mathcal{X} \times \mathcal{Y}$,

$$\begin{aligned}
\tilde{\nabla}f([\hat{z}]_\beta)^\top([\hat{z}]_\beta - z) &\le \frac{\beta}{2}\|\hat{z} - z\|^2 - \frac{\beta}{2}\|\hat{z}_1 - z\|^2 + 3\sqrt{2}\delta D \\
&= \frac{\beta}{2}(\hat{z} - \hat{z}_1)^\top(\hat{z} - z + \hat{z}_1 - z) + 3\sqrt{2}\delta D \\
&\le \frac{\beta}{2}(\|\hat{z} - z^*\| + \|\hat{z}_1 - z^*\|) \cdot \|\hat{z} - z + \hat{z}_1 - z\| + 3\sqrt{2}\delta D.
\end{aligned}$$

By (24) in the proof of Lemma C.4, since $\beta \ge 2\ell$ and $\mu \le \ell$, we have that

$$\|\hat{z}_1 - z^*\|^2 \le \|\hat{z} - z^*\|^2 + \frac{4\delta^2}{\beta\ell} + \frac{2\delta^2}{\beta\mu}.$$

Therefore, we can obtain that $\|\hat{z}_1 - z^*\| \le \|\hat{z} - z^*\| + 2\delta/\sqrt{\beta\ell} + \sqrt{2}\delta/\sqrt{\beta\mu}$, and thus,

$$\begin{aligned}
\tilde{\nabla}f([\hat{z}]_\beta)^\top([\hat{z}]_\beta - z) &\le \sqrt{2}\beta D(\|\hat{z} - z^*\| + \|\hat{z}_1 - z^*\|) + 3\sqrt{2}\delta D \\
&\le 2\sqrt{2}\beta D\|\hat{z} - z^*\| + \sqrt{2}\delta D\left(2\sqrt{\frac{\beta}{\ell}} + \sqrt{\frac{2\beta}{\mu}} + 3\right).
\end{aligned}$$

For the last statement, since $[\hat{z}]_\beta$ is obtained through 1-step of GDA with inexact gradients, by (20) in the proof of GDA for SC-SC problems before, we have that

$$\|[\hat{z}]_\beta - z^*\|^2 \le \left(1 + \frac{2\ell^2}{\beta^2}\right)\|\hat{z} - z^*\|^2 + \delta^2\left(\frac{1}{\beta\mu} + \frac{2}{\beta^2}\right).$$

Therefore, we obtain that $\|[\hat{z}]_\beta - z^*\| \le (1 + \sqrt{2}\ell/\beta)\|\hat{z} - z^*\| + \delta(1/\sqrt{\beta\mu} + \sqrt{2}/\beta)$. $\qquad\square$

The above lemma also applies to the case when exact gradients are available setting $\delta = 0$ and $[\hat{z}]_\beta = \Pi_{\mathcal{X} \times \mathcal{Y}}\left(\hat{z} - \frac{1}{\beta}\tilde{\nabla}f(\hat{z})\right)$ for the true gradients $\tilde{\nabla}f(\hat{z})$. This implies the stopping criteria $\tilde{\nabla}f(\hat{z})^\top(\hat{z} - z) \le \hat{\epsilon}, \forall z \in \mathcal{X} \times \mathcal{Y}$ in Algorithm 2 and 3 can be translated to $\|\hat{z} - z^*\|^2 \le \mathcal{O}(\hat{\epsilon}^2)$, which can be satisfied within $\mathcal{O}(\log(1/\hat{\epsilon}))$ complexity using Lemma C.3 and C.4 with $\delta = 0$ (or existing results in Tseng [72] or Facchinei and Pang [32]).

## C.2 Regularization Helps!

Proof of Theorem 4.4 and 4.6 for the near-optimal guarantees of Algorithm 2 is provided here.

### C.2.1 Inexact Initialization Oracle

We also use $(x_0, y_0)$ as the initialization point when solving the auxiliary strongly-convex problem. Note that the gradient steps starting from $(x_0, y_0)$ remain the same on $F(x, y)$ and $F_r(x, y)$.

*Proof of Theorem 4.4.* We first show the convergence guarantee. Let $z_r = (x_r, y_r)$. By fact $(i)$ in Lemma A.1, we have that $\forall z = (x, y) \in \mathcal{X} \times \mathcal{Y}$,

$$\begin{aligned}
\tilde{\nabla}F(z_r)^\top(z_r - z) &= \left(\tilde{\nabla}F_r(z_r) - r(z_r - z_0)\right)^\top(z_r - z) \\
&= \tilde{\nabla}F_r(z_r)^\top(z_r - z) + \frac{r}{2}\|z_0 - z\|^2 - \frac{r}{2}\|z_r - z_0\|^2 - \frac{r}{2}\|z_r - z\|^2 \quad (25) \\
&\le \epsilon_r + rD^2.
\end{aligned}$$

According to Lemma B.2, this means $\max_{y \in \mathcal{Y}} F(x_r, y) - \min_{x \in \mathcal{X}} F(x, y_r) \leq \epsilon_r + rD^2$.

We then show the reproducibility guarantee. Denote the saddle point of $F_r(x, y)$ given $(x_0, y_0)$ as $(x_r^*, y_r^*)$, and the saddle point of $F_r'(x, y) = F(x, y) + (r/2)\|x - x_0'\|^2 - (r/2)\|y - y_0'\|^2$ given $(x_0', y_0')$ as $((x_r^*)', (y_r^*)')$. By Lemma B.4 in Appendix B.3 of Zhang et al. [78], we have that

$$\|x_r^* - (x_r^*)'\|^2 + \|y_r^* - (y_r^*)'\|^2 \leq \|x_0 - x_0'\|^2 + \|y_0 - y_0'\|^2.$$

Let $z_r = (x_r, y_r)$, $z_r^* = (x_r^*, y_r^*)$ and $z_0 = (x_0, y_0)$ for simplicity of the notation. $z_r'$, $(z_r^*)'$ and $z_0'$ can be defined in the same way. Similarly to the minimization case, we have

$$\|z_r - z_r'\| \leq \|z_r - z_r^*\| + \|z_r^* - (z_r^*)'\| + \|(z_r^*)' - z_r'\|$$

$$\leq \delta + 2\sqrt{\frac{2\epsilon_r}{r}},$$

where we use $\|z_r^* - (z_r^*)'\| \leq \|z_0 - z_0'\| \leq \delta$ and optimality of $z_r^*$ by $r$ strong-convexity–strong-concavity (SC-SC) of $F_r(x, y)$ (the same holds true for $z_r'$ and $(z_r^*)'$ as well):

$$\frac{r}{2}\|x_r - x_r^*\|^2 + \frac{r}{2}\|y_r - y_r^*\|^2 \leq F_r(x_r, y_r^*) - F_r(x_r^*, y_r^*) + F_r(x_r^*, y_r^*) - F_r(x_r^*, y_r)$$

$$\leq \max_{y \in \mathcal{Y}} F_r(x_r, y) - \min_{x \in \mathcal{X}} F_r(x, y_r)$$

$$\leq \epsilon_r.$$

Thus setting $r = \epsilon/D^2$ and $\epsilon_r = \epsilon \cdot \min\{1, \delta^2/(8D^2)\}$, we guarantee that $\max_{y \in \mathcal{Y}} F(x_r, y) - \min_{x \in \mathcal{X}} F(x, y_r) \leq 2\epsilon$ and $\|x_r - x_r'\|^2 + \|y_r - y_r'\|^2 \leq 4\delta^2$. Applying Lemma C.5 with $\delta = 0$, the complexity using Extragradient (EG) [72, 57] to achieve $\epsilon_r$-error on $r$-SC–SC $(\ell + r)$-smooth minimax optimization is $\mathcal{O}((\ell/r+1)\log(1/\epsilon_r)) = \tilde{\mathcal{O}}(\ell D^2/\epsilon)$, where $\tilde{\mathcal{O}}$ hides logarithmic terms. $\square$

### C.2.2 Inexact Deterministic Gradient Oracle

This section contains proof of Theorem 4.6 for the near-optimal guarantees in the inexact deterministic gradient case. The proof is based on Lemma 4.5 (restated and proved as Lemma C.4 in Section C.1) and Lemma C.5.

*Proof of Theorem 4.6.* For the convergence guarantee, the same as (25), we have that

$$\max_{y \in \mathcal{Y}} F(x_r, y) - \min_{x \in \mathcal{X}} F(x, y_r) \leq \epsilon_r + rD^2.$$

For the reproducibility guarantee, we can obtain that

$$\|z_r - z_r'\| \leq \|z_r - z_r^*\| + \|z_r^* - z_r'\|.$$

Let $z_T$ be the output of $T$-step Extragradient with initialization $z_0$. By Lemma 4.5, we have that

$$\|z_T - z_r^*\|^2 \leq \exp\left(-\frac{T}{8}\frac{r}{\ell + r}\right)\|z_0 - z_r^*\|^2 + \frac{8\delta^2}{r}\left(\frac{2}{\ell + r} + \frac{1}{r}\right)$$

$$\leq \exp\left(-\frac{T}{16}\frac{r}{\ell}\right)\|z_0 - z_r^*\|^2 + \frac{16\delta^2}{r^2}.$$

Setting $T \geq (32\ell/r)\log(rD/\delta)$ and $r = \epsilon/D^2$, this means the algorithm converges to $\|z_T - z_r^*\| \leq 3\sqrt{2}D^2(\delta/\epsilon)$. Therefore, according to Lemma C.5, if we choose $z_r = [z_T]_{2\ell}$, since $1 \leq \ell D^2/\epsilon$, we can guarantee that $\|z_r - z_r^*\| \leq 3(2\sqrt{2}+1)D^2(\delta/\epsilon)$ and that

$$\max_{y \in \mathcal{Y}} F(x_r, y) - \min_{x \in \mathcal{X}} F(x, y_r) \leq \left(4(\sqrt{2}+7)\frac{\ell D^2}{\epsilon} + 3\sqrt{2}\right)\delta D + \epsilon.$$

The reproducibility is $\|z_r - z_r'\|^2 \leq 36(9 + 4\sqrt{2})D^4(\delta^2/\epsilon^2)$. $\square$

### C.3 Inexact Proximal Point Method

Proof of Theorem 4.7 and 4.8 for the guarantees of Algorithm 3 is provided in this section.

### C.3.1 General Analysis

We first analyze the convergence of the inexact proximal point method (Inexact-PPM). Given initialization $(x_0, y_0)$ and $\alpha > 0$, for $t = 0, 1, \cdots, T - 1$, each step of Inexact-PPM is

$(x_{t+1}, y_{t+1})$ is an inexact solution to $\min_{x \in \mathcal{X}} \max_{y \in \mathcal{Y}} \hat{F}_t(x, y) = F(x, y) + \frac{1}{2\alpha} \|x - x_t\|^2 - \frac{1}{2\alpha} \|y - y_t\|^2$.

**Lemma C.6.** *If we run Inexact-PPM and make sure that for each sub-problem $\tilde{\nabla} \hat{F}_t(z_{t+1})^\top (z_{t+1} - z) \leq \hat{\epsilon}$ for all $z = (x, y) \in \mathcal{X} \times \mathcal{Y}$, where $z_{t+1} = (x_{t+1}, y_{t+1})$ and $\tilde{\nabla} \hat{F}_t(z_{t+1}) = (\nabla_x \hat{F}_t(x_{t+1}, y_{t+1}), -\nabla_y \hat{F}_t(x_{t+1}, y_{t+1}))$, then we have $\forall z \in \mathcal{X} \times \mathcal{Y}$,*

$$\max_{y \in \mathcal{Y}} F(\bar{x}_{T+1}, y) - \min_{x \in \mathcal{X}} F(x, \bar{y}_{T+1}) \leq \frac{\|z_0 - z\|^2}{2\alpha T} + \hat{\epsilon}.$$

*Proof.* The proof is similar to Proposition 7 in Mokhtari et al. [58]. The same as (25), for any $z = (x, y) \in \mathcal{X} \times \mathcal{Y}$ and any $t = 0, 1, \cdots, T - 1$, we have that

$$\tilde{\nabla} F(z_{t+1})^T (z_{t+1} - z) = \left( \tilde{\nabla} \hat{F}_t(z_{t+1}) - \frac{1}{\alpha} (z_{t+1} - z_t) \right)^\top (z_{t+1} - z)$$

$$= \frac{1}{2\alpha} \|z_t - z\|^2 - \frac{1}{2\alpha} \|z_{t+1} - z\|^2 - \frac{1}{2\alpha} \|z_{t+1} - z_t\|^2 + \tilde{\nabla} \hat{F}_t(z_{t+1})^\top (z_{t+1} - z)$$

$$\leq \frac{1}{2\alpha} \|z_t - z\|^2 - \frac{1}{2\alpha} \|z_{t+1} - z\|^2 + \hat{\epsilon}.$$

Taking summation from $t = 0$ to $T - 1$ and dividing both sides by $T$, we conclude that

$$\frac{1}{T} \sum_{t=0}^{T-1} \tilde{\nabla} F(z_{t+1})^\top (z_{t+1} - z) \leq \frac{\|z_0 - z\|^2}{2\alpha T} + \hat{\epsilon}.$$

The proof is completed by Lemma B.2. $\qquad\square$

### C.3.2 Inexact Initialization Oracle

This section provides proof of Theorem 4.7.

*Proof of Theorem 4.7.* Let $\bar{z}_{T+1} = (\bar{x}_{T+1}, \bar{y}_{T+1}) = (1/T) \sum_{t=0}^{T-1} (x_{t+1}, y_{t+1})$. By Lemma C.6 and the choice that $\alpha = 1/\ell$, $\hat{\epsilon} = \delta^2/(2\alpha T^2)$, we immediately have

$$\max_{y \in \mathcal{Y}} F(\bar{x}_{T+1}, y) - \min_{x \in \mathcal{X}} F(x, \bar{y}_{T+1}) \leq \frac{\ell D^2}{T} + \frac{\ell \delta^2}{2T^2}.$$

$\mathcal{O}(1/T)$ convergence rate is guaranteed for $\delta \leq \mathcal{O}(\sqrt{T})$. Note that the condition number of $\hat{F}_t(x, y)$ is $\mathcal{O}(1)$ when $\alpha = 1/\ell$. Therefore, to guarantee an $\epsilon$-saddle point of $F(x, y)$, a total complexity of $\mathcal{O}(T \log(1/\hat{\epsilon})) = \mathcal{O}((1/\epsilon) \log(1/(\epsilon \delta)))$ is sufficient for various algorithms including GDA [32] and EG [72] applying Lemma C.5 with $\delta = 0$.

Let $z_t^* = (x_t^*, y_t^*)$ be the unique saddle point of $\hat{F}_t(x, y)$ with proximal center $z_t$, and $(z_t^*)'$ be the saddle point when the proximal center is $z_t'$. For the reproducibility guarantee, similarly to Section C.2.1, we can obtain that

$$\|z_{t+1} - z_{t+1}'\| \leq \|z_{t+1} - z_t^*\| + \|z_t^* - (z_t^*)'\| + \|(z_t^*)' - z_{t+1}'\|$$

$$\leq \|z_t - z_t'\| + 2\sqrt{\frac{2\hat{\epsilon}}{\ell}}, \tag{26}$$

where we use Lemma B.4 in Zhang et al. [78] and $(1/\alpha)$-SC–SC of $\hat{F}_t(x, y)$:

$$\tilde{\nabla} \hat{F}_t(z_{t+1})^\top (z_{t+1} - z_t^*) \geq \hat{F}_t(x_{t+1}, y_t^*) - \hat{F}_t(x_t^*, y_{t+1}) + \frac{1}{2\alpha} \|z_{t+1} - z_t^*\|^2$$

$$\geq \frac{\ell}{2} \|z_{t+1} - z_t^*\|^2.$$

Therefore, by induction, we have that for any $t = 1, 2, \cdots, T$,

$$\|z_t - z_t'\| \leq \|z_0 - z_0'\| + 2t\sqrt{\frac{2\hat{\epsilon}}{\ell}}$$

$$\leq \delta + 2\delta\frac{t}{T}$$

$$\leq 3\delta.$$

The reproducibility is then $\|\bar{z}_{T+1} - \bar{z}_{T+1}'\|^2 \leq 9\delta^2$ using Jensen's inequality. $\qquad\square$

### C.3.3  Inexact Deterministic Gradient Oracle

For Theorem 4.8, we provide proof when using GDA as the base algorithm. According to Lemma C.4, EG can also be applied here with a similar argument.

*Proof of Theorem 4.8.* When setting $\alpha = 1/\ell$, the auxiliary problem is $\ell$–strongly-convex–strongly-concave and $2\ell$-smooth. Let $z_t^K$ be the output of $K$-step GDA with initialization $z_t^0$ on the minimax problem $\min_{x \in \mathcal{X}} \max_{y \in \mathcal{Y}} \hat{F}_t(x, y)$ at iteration $t$. Denote its saddle point as $z_t^*$. By Lemma C.3, if $K \geq 32 \log(8\ell^2 D^2/(3\delta^2))$, we have that

$$\|z_t^K - z_t^*\|^2 \leq \exp\left(-\frac{K}{32}\right) \|z_t^0 - z_t^*\|^2 + \frac{9\delta^2}{4\ell^2}$$

$$\leq \frac{3\delta^2}{\ell^2}.$$

By Lemma C.5, we can thus set $z_{t+1} = [z_t^K]_{2\ell}$ and guarantee that

$$\tilde{\nabla} \hat{F}_t(z_{t+1})^\top (z_{t+1} - z) \leq (4\sqrt{6} + 5\sqrt{2} + 4)\delta D, \quad \forall z \in \mathcal{X} \times \mathcal{Y}.$$

According to Lemma C.6, we then have

$$\max_{y \in \mathcal{Y}} F(\bar{x}_{T+1}, y) - \min_{x \in \mathcal{X}} F(x, \bar{y}_{T+1}) \leq \frac{\ell D^2}{T} + 21\delta D.$$

When $\delta \leq \epsilon/(42D)$, $T \geq 2\ell D^2/\epsilon$ is required to obtain an $\epsilon$-saddle point, and the total gradient complexity is $TK = (64\ell D^2/\epsilon)\log(8\ell^2 D^2/(3\delta^2)) = \tilde{\mathcal{O}}(1/\epsilon)$ with $\tilde{\mathcal{O}}$ hiding logarithmic terms.

We then show the reproducibility guarantee. From Lemma C.5, we know that $\|z_{t+1} - z_t^*\| \leq (1 + \sqrt{2}/2)\|z_t^K - z_t^*\| + \sqrt{2}\delta/\ell \leq 4.5\delta/\ell$. By (26), we have that

$$\|z_{t+1} - z_{t+1}'\| \leq \|z_t - z_t'\| + \frac{9\delta}{\ell}.$$

By induction, we conclude that $\|z_t - z_t'\| \leq 9t(\delta/\ell)$, and thus the reproducibility is $\|\bar{z}_{T+1} - \bar{z}_{T+1}'\|^2 \leq 81\delta^2 T^2/\ell^2 = 324D^4(\delta^2/\epsilon^2)$. $\qquad\square$

## D  Numerical Experiments

Some numerical experiments that demonstrate the effectiveness of regularization to improve reproducibility are provided in this section. We test the algorithms on two problems: a minimization problem with a quadratic objective and a minimax problem with a bilinear objective. The experiments are conducted on a single local machine.

**Minimization.**  We first compare the performance of gradient descent (GD), accelerated gradient descent (AGD), Algorithm 1 with GD as the base algorithm (Reg-GD), and Algorithm 1 with AGD as the base algorithm (Reg-AGD) on a quadratic minimization problem

$$\min_{x \in \mathbb{R}^d} \frac{1}{2}\|Ax - b\|^2.$$

Here, $b \in \mathbb{R}^d$ with each entry sampled from the Gaussian distribution with mean 0 and standard deviation 10 and $A \in \mathbb{R}^{d \times d}$ is a random positive semi-definite matrix with rank $d - 1$ that makes

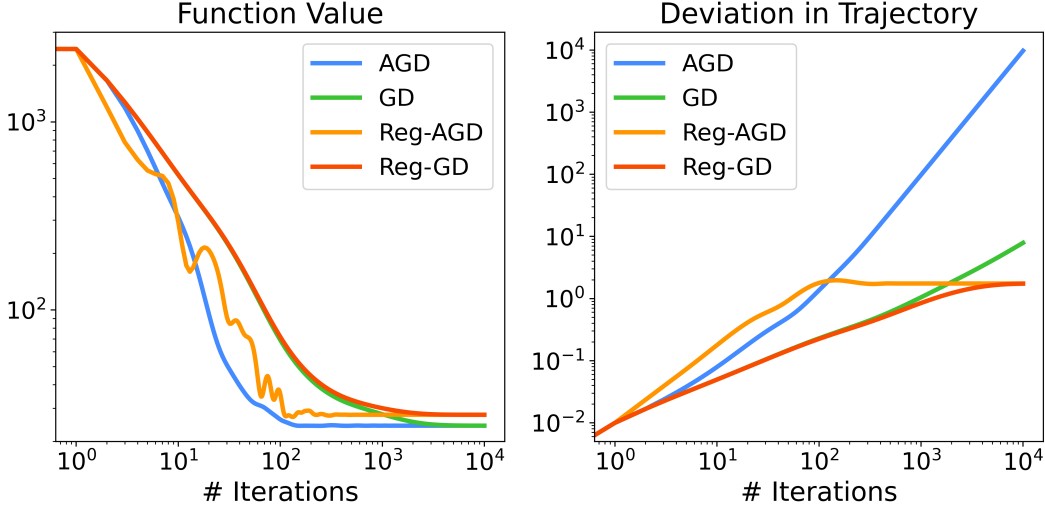

Figure 1: Comparisons among GD, AGD, and their regularized version on the quadratic minimization problem with $\delta$-inexact gradients. The left figure plots the convergence behavior and the right shows the reproducibility. Both axes are plotted utilizing a logarithmic scale.

sure the problem is convex but not strongly-convex. To be specific, we let $A = U\Sigma U^\top$ where $U$ is a random orthogonal matrix drawn from the Haar distribution, and $\Sigma$ is a diagonal matrix with 1 entry being 0 and the others uniformly sampled from $[0.1, 10]$. This ensures that the problem is smooth with a parameter smaller than $100$.

We implement an inexact gradient oracle that returns $A^\top(Ax - b) + \delta e$ where $e \in \mathbb{R}^d$ is an all-one vector and $\delta \in \mathbb{R}$ controls the inexactness level. We test the aforementioned four algorithms with this inexact gradient oracle on both convergence performance measured by function value and reproducibility performance measured by the deviation compared to the trajectory obtained from using the true gradient when $\delta = 0$. In the experiments, we let $d = 100$ and $\delta = 0.1$. For all four algorithms, we set the number of iterations to be $T = 10000$, and the stepsize to be 0.01 based on the fact that the smoothness parameter is at most 100. For the regularization-based methods, we set the regularization parameter of the auxiliary problem to 0.05. All other parameters are set according to the theoretically suggested values. The results are illustrated in Figure 1.

In Figure 1, we see AGD converges faster than GD, but the deviation in iterates is much larger. When introducing regularization, i.e., Reg-AGD, the reproducibility guarantee is greatly improved with only a small degradation in the convergence performance. It is worth mentioning that Reg-GD also has a smaller deviation bound compared to GD. All the results align with our theoretical analysis. Changing the inexactness level $\delta$ or the random seed for sampling the matrix $A$ and the vector $b$ does not influence the phenomenon too much, so we do not report the results with different selections.

**Minimax.** We also test the performance of gradient descent ascent (GDA), Extragradient (EG), and their regularized counterparts (Reg-GDA and Reg-EG) in Algorithm 2 on a bilinear matrix game

$$\min_{x \in \mathcal{X}} \max_{y \in \mathcal{Y}} \ x^\top A y.$$

Here, $A \in \mathbb{R}^{d \times d}$ is generated the same as in the quadratic minimization example, $\mathcal{X} = \{x \in \mathbb{R}^d \mid \|x\| \le D\}$ and $\mathcal{Y} = \{y \in \mathbb{R}^d \mid \|y\| \le D\}$ are $d$-dimensional balls with diameter $2D$ measured by the Euclidean norm. The projection onto these balls can be easily achieved. We implement an inexact gradient oracle that returns $Ay + \delta e$ and $A^\top x + \delta e$ for the partial gradients w.r.t. $x$ and $y$ respectively, where $e \in \mathbb{R}^d$ is an all-one vector and $\delta \in \mathbb{R}$ controls the inexactness level.

We test the aforementioned four algorithms with this inexact gradient oracle on both convergence performance measured by the duality gap (computable due to bounded domain) and reproducibility performance measured by the deviation compared to the trajectory obtained from using the true

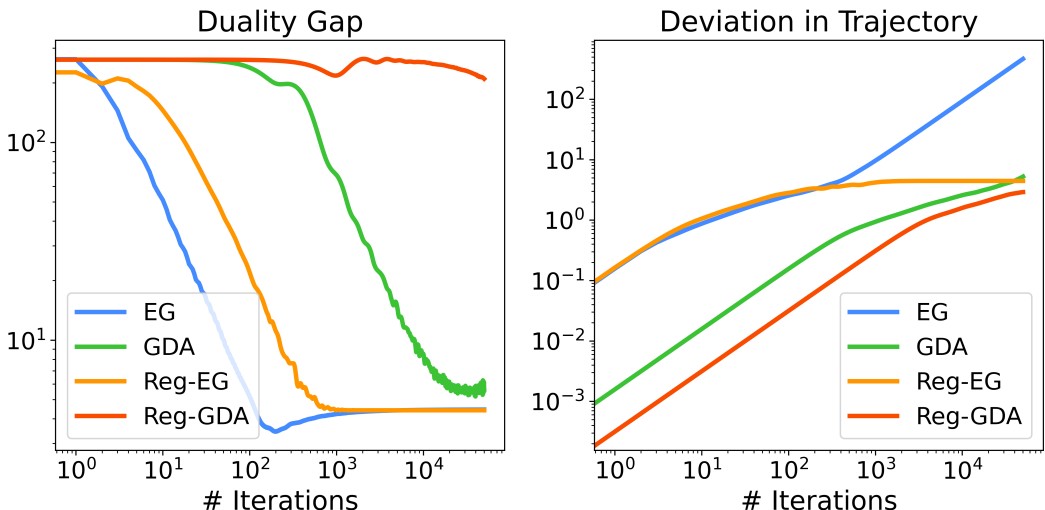

Figure 2: Comparisons among GDA, EG, and their regularized version on the bilinear matrix game with $\delta$-inexact gradients. The left figure plots the convergence behavior and the right shows the reproducibility. Both axes are plotted utilizing a logarithmic scale.

gradient when $\delta = 0$. In the experiments, we let $d = 500$ and $\delta = 0.1$. For all four algorithms, we set the number of iterations to be $T = 10000$, and the stepsize is 0.1 for EG, 0.05 for Reg-EG, 0.001 for GDA, and 0.0001 for Reg-GDA. The choice of stepsizes here adheres to our theoretical analysis noticing that the smoothness parameter is no larger than 10. Larger stepsize for GDA will make the trajectory easily diverge. For the regularization-based methods, we set the regularization parameter of the auxiliary problem to 0.05. The results are plotted in Figure 2. We see again the effectiveness of regularization to improve the reproducibility of the algorithms. Reg-EG largely reduces the deviation of EG even with respect to magnitude (note the figure is logarithmically scaled), with only a small degradation in terms of the convergence speed.

