# OpenReview forum: "Optimal Guarantees for Algorithmic Reproducibility and Gradient Complexity in Convex Optimization"
_NeurIPS.cc/2023/Conference — NeurIPS 2023 spotlight_

### Official Review · Reviewer_r2W7 · 2023-07-01

**Soundness:** 4 excellent
**Presentation:** 4 excellent
**Contribution:** 4 excellent
**Rating:** 7
**Confidence:** 5

**Summary:**

This work proposes optimization algorithms that achieve optimal convergence rate and reproducibility for the settings of convex optimization and convex-concave minimax settings. This work settled some of the open questions from the previous work, and extend the results to the minimax setting.

**Strengths:**

- Clear presentation.
- Novel technical contributions.
- Strong theoretical results and cleanly written proofs.

**Weaknesses:**

- Overall, the presentation is very clear, and the results are rigorous, and I didn't see any major weaknesses.
- Some details about Inexact-EG would have been helpful since it seems like a new method developed in this work. In particular, is it a direct extension of Devolder et al. to the minimax setting?


**Questions:**

- Regarding the O(delta^2/\eps^{2/5}) bound on the reproducibility, do you have some intuitions as to why improving upon this would be difficult? Even if it's a heuristic argument, it would help readers understand the main challenge.

**Limitations:**

- They discussed them well in the conclusion section.

---

> ### Author Rebuttal · Authors · 2023-08-09
>
> We thank the expert reviewer for recognizing our contribution and for the very positive feedback.
>
> 1. **Details about Inexact-EG.**
>
> > Inexact-EG seems like a new method developed in this work. Is it a direct extension of Devolder et al. to the minimax setting?
>
> Extragradient (EG) is proposed by Korpelevich in 1976 and becomes popular in the minimax literature. Devolder et al. introduce an inexact oracle in the context of smooth convex minimization problems that is different from the inexact gradient oracle considered in this work. We provide a detailed discussion about the relationship between the two oracles in Appendix A.2. In this work, we analyze the Extragradient method under the inexact gradient oracle. Since the method does not use the true gradients but some inexact gradients, we denote it as Inexact-EG. In this sense, Inexact-EG is neither a new method nor a direct extension of Devolder et al. to the minimax setting.
>
> 2. **Sub-optimal reproducibility of Inexact-AGD.**
>
> > Are there any intuitions why improving upon the $\mathcal{O}(\delta^2/\epsilon^{2.5})$ reproducibility of AGD would be difficult?
>
> The reproducibility of the proposed framework depends on how accurately the strongly-convex sub-problem is solved. According to the lower bounds in Devolder et al., it is only possible to guarantee convergence to a neighborhood of the optimal point when the oracle has certain inexactness, and the size of this neighborhood depends on the condition number $\kappa:=\ell/\mu$. Since the strongly-convex parameter $\mu$ of our sub-problems is in the order of $\epsilon$ to ensure convergence to the original convex problem, we introduce additional $\epsilon$-dependence in the reproducibility as well. We argue improving upon such dependence could be difficult as a result of the lower bounds provided in Devolder et al. More discussions can be found in Appendix A.2. Probably a different algorithm design is required to attain optimal reproducibility of AGD.

---

> > ### Comment · Reviewer_r2W7 · 2023-08-13
> >
> > I see, thank you for your response -- please reflect them in the final version.
> > This is a great work and technically sound.
> > Congrats, and good luck!

---

### Official Review · Reviewer_hiCP · 2023-07-04

**Soundness:** 3 good
**Presentation:** 4 excellent
**Contribution:** 4 excellent
**Rating:** 7
**Confidence:** 2

**Summary:**

The authors proposed and studied optimization algorithms in both the convex and the convex-concave minimax setting, where both the criteria of convergence and reproducibility are measured. The authors provided upper bounds on these criteria for the proposed algorithms that matches nearly all of the lower bounds.

While I'm not an expert on this subject (and I leave the judgement of correctness to other experts in this area), I believe this work has reached satisfying results, and I would recommend accept.

**Strengths:**

1. The upper bounds on convergence and reproducibility matches nearly all of the known lower bounds.
2. The presentation of the work is quite clean and readable.

**Weaknesses:**

N/A

**Questions:**

N/A

---

> ### Author Rebuttal · Authors · 2023-08-09
>
> We thank the reviewer for acknowledging the contribution of this work.

---

### Official Review · Reviewer_FHMd · 2023-07-09

**Soundness:** 3 good
**Presentation:** 2 fair
**Contribution:** 3 good
**Rating:** 7
**Confidence:** 2

**Summary:**

The authors investigate the reproducibility problem of algorithms that solve convex and convex-concave minimax optimization problems.  They propose new methods with better reproducibility guarantees while maintaining the theoretical state-of-the-art convergence rates.

**Strengths:**

*I want to acknowledge that I got this paper for review **after the deadline**, so I didn't have much time to check every detail, especially the proof.*

The idea of using regularization is good. It leads to new theoretical state-of-the-art convergence rates and reproducibility guarantees. This is a solid contribution to the NeurIPS community.



**Weaknesses:**

1. The new method requires $\epsilon$ and the distance $D$ between the starting point $x_0$ and $x^*.$ (e.g. Theorem 3.3.) I am not sure that the previous methods in Table 1 need these parameters. Do the authors discuss these important limitations?
2. Algorithm 1 is a well-known method in the optimization community. For instance, see https://arxiv.org/pdf/1603.05642.pdf. It is called "regularization technique" or "regularization reduction." I believe that the authors should cite the previous works that consider Algorithm 1.
3. Wrong citation [55] in Theorem 3.3. The paper [55] doesn't provide an analysis of AGD for strongly convex functions. It is better to cite any of Nesterov's books.

**Questions:**

.

**Limitations:**

.

---

> ### Author Rebuttal · Authors · 2023-08-09
>
> We thank the reviewer for positive feedback and valuable suggestions. We will mention the limitations that our algorithm requires knowledge of $\epsilon$ and $D$, whereas previous methods do not. We will also add and correct the citations following the reviewer's suggestion. Thanks for the pointers.

---

> > ### Comment · Reviewer_FHMd · 2023-08-10
> > **Respond**
> >
> > Can the authors also comment on the following weakness?
> >  > Algorithm 1 is a well-known method in the optimization community. For instance, see https://arxiv.org/pdf/1603.05642.pdf. It is called "regularization technique" or "regularization reduction." I believe that the authors should cite the previous works that consider Algorithm 1.
> >
> > Do the authors discuss the connection between these methods and their method?

---

> > > ### Author Response · Authors · 2023-08-14
> > >
> > > Thanks a lot for the follow-up discussion. We apologize for not clearly addressing your question before.
> > >
> > > Adding regularization is indeed a common and useful technique in the optimization literature. The work [AH16] mentioned by the reviewer is one important use case where regularization is added to **boost convergence analysis**, i.e., to leverage known and good convergence properties of algorithms on smooth strongly-convex functions and transform it to other functions including convex and nonsmooth cases. The algorithmic frameworks 1 and 2 in our paper only consider solving one auxiliary regularized strongly-convex problem, which is referred to as **classical regularization reduction** in [AH16]. The algorithm is *biased* and requires the knowledge of $\epsilon$ and $D$ to control the biased term introduced by the regularization term. The convergence guarantee also has an additional sub-optimal logarithmic term depending on $\epsilon$. In comparison, [AH16] propose to use a double-loop algorithm, where a sequence of auxiliary regularized strongly-convex problems with decreasing regularization parameters are solved. The decreasing regularization ensures the algorithm is *unbiased*, and the resulting convergence guarantee requires no knowledge of $\epsilon$ and does not have an additional logarithmic term. We realize that the same idea could apply to our case as well, where it is very possible to remove the additional sub-optimal logarithmic factor in our convergence rate as well as the requirement of knowing $\epsilon$. We want to thank the reviewer again for pointing this out. We will add this discussion in the conclusion section for our limitations and potential future work.
> > >
> > > In addition to boosting convergence such as [AH16] and Catalyst [LMH15], the regularization technique has also been demonstrated to be useful in improving stability and generalization [AK22, Zha+21], enhancing sensitivity and privacy guarantees [FKT20], etc. In this paper, we provide another use case by showing an improved convergence-reproducibility trade-off. We will add another paragraph in the related work section to discuss these related examples as well.
> > >
> > > **References**
> > >
> > > * **[AH16]** Zeyuan Allen-Zhu and Elad Hazan. Optimal black-box reductions between optimization objectives. Advances in Neural Information Processing Systems, 2016.
> > >
> > > * **[LMH15]** Hongzhou Lin, Julien Mairal, and Zaid Harchaoui. A universal catalyst for first-order optimization. Advances in Neural Information Processing Systems, 2015.
> > >
> > > * **[AK22]** Amit Attia and Tomer Koren. Uniform stability for first-order empirical risk minimization. Conference on Learning Theory, 2022.
> > >
> > > * **[Zha+21]** Junyu Zhang, Mingyi Hong, Mengdi Wang, and Shuzhong Zhang. Generalization bounds for stochastic saddle point problems. International Conference on Artificial Intelligence and Statistics, 2021.
> > >
> > > * **[FKT20]** Vitaly Feldman, Tomer Koren, and Kunal Talwar. Private stochastic convex optimization: optimal rates in linear time. Symposium on Theory of Computing, 2020.

---

### Official Review · Reviewer_jCKW · 2023-07-09

**Soundness:** 4 excellent
**Presentation:** 3 good
**Contribution:** 3 good
**Rating:** 7
**Confidence:** 4

**Summary:**

The paper considers the problem of ensuring reproducibility in convex optimization. It builds on a recent framework for understanding reproducibility initiated by Ahn et al. The paper considers both minimization and minimax optimization (the latter being a new setting investigated in this work). The key results of the paper are the following:

1. Minimization problems: The paper shows an improvement in the convergence-reproducibility tradeoffs under inexact initialization and inexact gradients compared to the results of Ahn et al. For inexact initialization, the paper shows that an L2 regularized version of AGD simultaneously obtains optimal convergence and optimal reproducibility. For inexact gradients, the same algorithm obtains sub-optimal reproducibility but with optimal convergence.

2. Minimax optimization: L2 regularized versions of existing algorithms achieve optimal reproducibility and near-optimal gradient complexity. Similar to Ahn et al., SGD attains optimal reproducibility and convergence under a stochastic gradient oracle.

**Strengths:**

1. The problem of developing reproducible optimization algorithms is well-motivated and relevant. Various empirical studies have shown that randomness in initialization, training, data augmentation and numerical instabilities can lead to models which make significantly different predictions on test points.

2. The paper demonstrates a valuable algorithmic principle of using L2 regularization to ensure reproducibility. All the results in the paper (apart from those for a stochastic gradient oracle) are obtained by incorporating L2 regularization into prior algorithms. The results strongly suggest that there could be deeper connections between stability and reproducibility, since L2 regularization is also a similarly useful techniques for ensuring algorithmic stability. Investigating this could be an interesting direction of future work.

3. The paper improves stronger bounds across a number of settings compared to prior work. It also demonstrates that suspected instability issues of AGD are not a barrier to obtaining reproducibility guarantees for it. The paper also broadens the study of reproducibility to minimax optimization, with a similar message for it.

**Weaknesses:**

1. The writing of the paper is decent overall, but could do with some improvements. The paper does not motivate reproducibility adequately on its own. Some of the comments seem to concern reproducibility in science broadly rather than the particular issues which concern reproducibility in modern ML. Since the paper has a lot of different results, some more intuition behind the specific bounds that are obtained could be useful. For example, the paper could comment on the reproducibility obtained in different settings and why these bounds arise from the algorithm.

2. Though I can understand that there is not too much space given the number of results, some intuition for the technical ideas which go into the bounds would be good as well. In particular, what are the main ideas behind extending reproducibility to minimax optimization? Does the intuition for why L2 regularization works for reproducibility in minimization mostly carry over and give the bounds for minimax optimization?

**Questions:**

Overall, this paper makes a good contribution on a relevant problem and I don't have any major concerns or questions. Some other suggestions are included above.

**Limitations:**

These are discussed adequately.

---

> ### Author Rebuttal · Authors · 2023-08-09
>
> We thank the reviewer for the positive feedback and insightful suggestions.
>
> 1. **Writing of the paper.**
>
> > The paper does not motivate reproducibility adequately on its own. Since the paper has a lot of different results, some more intuition behind the specific bounds and technical ideas obtained could be useful.
>
> We will try our best to motivate why reproducibility needs to be studied and provide more intuition about our results. For now, in addition to the theoretical work of Ahn et al. [1], we provide a set of previous empirical works on reproducibility to justify that reproducibility has become an important topic in modern machine learning, e.g., [40] for reproducibility issues in reinforcement learning and [59] for a report of NeurIPS 2019 reproducibility program.
>
> Although there are lots of different results in the paper, we think the main motivations and insights behind them are **consistent**.
>
> * For the smooth convex minimization setting, previous work suggests that GD is optimally reproducible but converges sub-optimally, while AGD converges optimally but is not reproducible. There seems to be a fundamental trade-off between convergence speed and reproducibility of algorithms. This motivates us to study whether it is possible to attain both optimal convergence and reproducibility at the same time.
>
> * For the smooth convex-concave minimax setting, we observe a similar behavior of GDA and EG that mirrors the minimization setting, and we also ask the same question here.
>
> The reason why GD/GDA can be optimally reproducible is that the gradient descent step is **non-expansive** when the objective is smooth and convex. When introducing certain momentum or extrapolation step to accelerate convergence speed, such non-expansiveness property often disappears, which makes the optimally convergent algorithms AGD/EG not reproducible. The main idea behind our (near)-optimal algorithmic framework to simultaneously attain the best of the two worlds is to add regularization and leverage **uniqueness and stability of the solutions to strongly-convex problems** (or the non-expansiveness of proximal point steps), e.g., Lemma 3.2. As a result, it is possible to improve the reproducibility of optimally convergent algorithms through convergence on the strongly-convex sub-problems while selecting the regularization parameter small enough to avoid too much approximation error introduced by the regularization term.
>
> 2. **Extension to minimax optimization.**
>
> > What are the main ideas behind extending reproducibility to minimax optimization? Does the intuition for why regularization works for reproducibility in minimization mostly carry over to minimax optimization?
>
> The intuition and technical ideas to use regularization mostly carry over from minimization to minimax optimization. In particular, similarly to Lemma 3.2, the saddle point $(x\_r^\*, y\_r^\*)$ of the strongly-convex-strongly-concave (SC-SC) function $F(x,y) + (r/2)\Vert x - x\_0\Vert^2 - (r/2)\Vert y - y\_0\Vert^2$ is also unique and satisfies that
>
> $$\Vert x\_r^\* - (x\_r^\*)'\Vert^2 + \Vert y\_r^\* - (y\_r^\*)'\Vert^2 \leq \Vert x\_0 - x\_0'\Vert^2 + \Vert y\_0 - y\_0'\Vert^2.$$
>
> As a result, by converging closely enough on the SC-SC sub-problem, this property of the optimal solution can be leveraged to obtain the optimal reproducibility guarantee. In addition, the smooth SC-SC minimax problems can be solved efficiently by a large class of algorithms, which at the same time maintains fast convergence guarantees. More interestingly, since the (inexact) proximal point method already attains the (near)-optimal convergence rate for the smooth convex-concave minimax problems, it is possible for Algorithm 3 to be optimally reproducible and convergent at the same time for minimax problems. The same framework cannot be used to improve the trade-offs in the minimization setting because of its sub-optimal convergence for smooth convex minimization problems.

---

> > ### Comment · Reviewer_jCKW · 2023-08-16
> > **Thanks for the rebuttal**
> >
> > Thank you, the proposed updates sound good and I'm happy to still recommend the paper for acceptance.

---

### Official Review · Reviewer_u7gX · 2023-07-11

**Soundness:** 3 good
**Presentation:** 3 good
**Contribution:** 3 good
**Rating:** 7
**Confidence:** 3

**Summary:**

The paper studies the problem of reproducibility in convex optimization. The notion of reproducibility, borrowed from prior work, measures the "stability" of the output of a procedure under noisy initialization or gradient computation. For the smooth convex setting, they design an algorithm based on running accelerated gradient descent on a regularized objective, which achieves optimal reproducibility and convergence rate. This answers an open question from prior work. The authors further extend their results to the minmax optimization setting deriving many new results.

**Strengths:**

1. Reproducibility has become an important topic in modern machine learning. Since (convex) optimization is the dominant algorithmic paradigm for modern ML, it is imporant to formulate and study reproducibility in optimization. The topic of the paper thus is important and timely.

2. The paper obtains the optimal bounds on convergence and reproducibility for the smooth convex setting, something which prior work conjectured to be unattainable. This is an important contribution. Further, they managed to also get optimal, and non-trivial rates for the minmax optimization, a setting which has received considerable attention lately.

 3. The paper is well written. Granted that it covers a lot of algorithms and results, the writing is to the point and the flow of ideas is natural.

**Weaknesses:**

1. About the definition of reproducibility: since this is a new field, I presume that the community has not yet agreed upon a definition. However, it seems to me that the only paper using the definition in this paper is the prior work of Ahn et al. Does adhering to this definition indeed reflect reprodubility in practice, in some sense? Even in optimization settings, there are other sources of instability not accounted for in the analysis, for instance, truncation and rounding due to the finite precision. Is adhering to reproducility (say with respect to initialization) and disregarding potential numerical instability arising in other steps give some meaningful in practice?

2. Related to the above, some experiments demonstrating usefulness of the framework would strengthen the paper. In the current version, there are no experiments.

3. The underlying idea is very simple and has been used many a times in (related) prior works -- regularization makes the problem strongly convex and thus aids leasds to (various forms of) stability. Nonetheless, the authors build on this to provide non-trivial bounds for many settings.

4. Some technical details, which I presume are in prior work,  are not covered in the main text. Something that confused me is how to define  "optimal reroducibility", which is referred many a times in the paper. Some text explaining it, perhaps in the preliminaries will be helpful.

5. What if we consider an inexact initialization as well as inexact grdient, with the same $\delta$ say -- is it possible to say something about this from the algorithms proposed?

6. The authors analyze a number of algorithms in the minmax setting, as a result this part of the paper looks rather dense with Thm statements.  Some organization of what is to come will help the reader. From my understanding, Alg3, Inexact Proximal Point Method, strictly improves over all others? If yes, this should be conveyed early on in this section.

**Questions:**

Please answer the questions posed in the weekaneses.

**Limitations:**

The work is limited to a certain notion of reproducility, used only in one prior work, and thus which may or may not be the definition as the area matures. Further, even though the authors identify two sources of instability: initialization and gradient computation, they are studied separately. A unified analysis can perhaps reflect more about the practical aspects.

---

> ### Author Rebuttal · Authors · 2023-08-09
>
> We thank the reviewer for the valuable suggestions.
>
> 1. **Definition of reproducibility.**
>
> > Does this definition indeed reflect reproducibility in practice? There are other sources of instability not accounted for in the analysis. Is disregarding potential numerical instability arising in other steps still meaningful in practice? Moreover, some experiments would strengthen the paper.
>
> We agree with the reviewer that there is no consensus on what is the right mathematical notion of reproducibility in the community. It would be impossible to find a perfect definition that accounts for all sources of instability in practice. In our humble opinion,  the current definition adopted in our paper is a meaningful one and at least partially reflects practical needs. Here, the source of irreproducibility is modeled by inexact oracles and reproducibility is defined to be the deviation in algorithms' outputs under such inexact oracles. The numerical instability in practice can be modeled as inexact updates $x\_{t+1} = x\_t - \alpha\nabla F(x\_t) + \delta$ where $\delta$ represents all errors coming from truncation and rounding due to the finite precision. This could fit into the inexact gradient oracle model that we considered as $x\_{t+1} = x\_t - \alpha(\nabla F(x\_t) + \delta')$. Hence, our analysis could also apply to such sources of numerical instability.
>
> 2. **Lack of experiments.**
>
> > Some experiments demonstrating the usefulness of the framework would strengthen the paper. In the current version, there are no experiments.
>
> We are afraid that the reviewer might have missed checking our appendix in the supplementary material, where we have already provided some toy numerical experiments in Appendix D (along with Python codes in the supplementary material) to showcase the effectiveness of regularization on improving reproducibility for both minimization and minimax optimization.  We kindly note that the supplementary material can be found in the zip folder.
>
>
> 3. **The underlying idea is simple.**
>
> > The underlying idea is very simple and has been used many times in related works. Nonetheless, the authors build on this to provide non-trivial bounds for many settings.
>
> We take simplicity rather as a compliment especially when the simple algorithm yields the optimal guarantees.  Although the idea of regularization has been developed before, they are often used for different purposes, e.g., to boost convergence [68], to improve stability [5], or to enhance privacy guarantees [71]. We provide an important use case of the regularization technique by showing an improved convergence-reproducibility trade-off.
>
> 4 and 6. **Organization of the paper.**
>
> > Some text explaining "optimal reproducibility" perhaps in the preliminaries will be helpful; The authors analyze a number of algorithms in the minimax setting. Some organization of what is to come will help the reader; Algorithm 3 should be conveyed early on in this section.
>
> Thanks for the suggestion. For "optimal reproducibility", we mean the algorithm attains the lower bound of reproducibility established in Ahn et al. We will add more discussions about it in the preliminaries section. In the minimax setting, the aim of providing analysis of GDA/EG is to mirror what is known in the minimization setting, i.e., the optimally reproducible algorithm GD converges sub-optimally, while the optimally convergent algorithm AGD is not reproducible. This motivates us to study whether both optimal results can be achieved at the same time. Following the same idea as the minimization setting, we also propose Algorithm 2 for minimax problems. Finally, since the inexact proximal point algorithm already achieves near-optimal convergence for smooth convex-concave minimax problems (but not for minimization problems), it is possible to have Algorithm 3 that strictly improves all the others. We will add organization paragraphs and more discussions at the beginning of sections.
>
> 5. **Combination of different oracle settings.**
>
> > What if we consider both the inexact initialization oracle as well as the inexact gradient oracle? A unified analysis can perhaps reflect more on the practical aspects.
>
> Thanks for the interesting question. It is possible and immediate to extend the current definition to consider both errors at the same time, and the resulting bounds will simply be the **summation of the two**. Taking gradient descent as a simple example, the deviation in iterates will be
>
> $$\Vert x\_t-x\_t'\Vert \leq \Vert x\_0 - x\_0'\Vert + 2\alpha\delta t,$$
>
> where $\Vert x\_0 - x\_0'\Vert$ is the inexactness of initialization, $\delta$ is the inexactness of gradients, and $\alpha$ is the stepsize. This expression successfully unifies both inexact oracles and recovers either one by setting the other source of error to 0. The same holds true for the proposed regularization framework. We will add a discussion about this in our revision.

---

> > ### Comment · Reviewer_u7gX · 2023-08-21
> > **Thanks!**
> >
> > I thank the authors for their detailed response and pointing to experiments in Appendix D. I encourage the authors to include some of the above discussion, especially those around definiton of optimal resproducibiliy from Ahn et al, as well as combination of the two settings, to the revised version. I increase my score to 7.

---

### Official Review · Reviewer_MNe7 · 2023-07-20

**Soundness:** 3 good
**Presentation:** 3 good
**Contribution:** 3 good
**Rating:** 7
**Confidence:** 3

**Summary:**

In this paper, the authors introduce a novel algorithmic framework that can achieve near optimal convergence while preserving optimal reproducibility, for minimizing smooth convex objectives and minimax optimization of convex-concave objectives. Here, reproducibility under inexact initialization oracles, inexact deterministic gradient oracles, and inexact stochastic gradient oracles are considered. The key idea is to optimize a regularized strongly convex objective ( or Strongly convex - strongly concave objective for minimax optimization) using a given base algorithm, and bounding the error introduced by the regularization. The authors derive convergence and reproducibility guarantees that match or improve existing guarantees for different base algorithms applied to the proposed framework.

**Strengths:**

* The paper discuss about theoretically improving the trade-off between optimal convergence and optimal reproducibility of algorithms, which is an important emerging research area.

* The paper is fairly easy to read, and the methods and results are presented in a clear manner.

* Using the proposed algorithmic framework, the authors show, contrary to what was previously believed, that accelerated gradient descent (AGD) method can achieve near optimal convergence preserving optimal reproducibility, which seems like a non-trivial and interesting result.

**Weaknesses:**

* The paper covers only convex (convex-concave) objective minimization (minimax optimization), which prevents the results of this paper being applied to many applications where reproducibility is a challenge (e.g. reinforcement learning) as mentioned in the introduction of the paper.

* The paper considers a constrained optimization setting for minimax optimization, while most applications of optimization, such as machine learning, will be deployed in an unconstrained setting. This might again prevent these results being applied to many practical applications.

* The dependence of the convergence bounds on the diameter of $\mathcal{X}$ and $\mathcal{Y}$ in Theorems 4.4. and 4.6 makes the corresponding bounds too loose when $\mathcal{X}$ and $\mathcal{Y}$ are significantly large.

Minor comments

* Using $x^*_{r’}$ to denote  $\underset{x\in\mathcal{X}}{\operatorname{argmax}}\\{ F(x) + (r/2) \Vert x - x_0'  \Vert^2\\}$, it might suggest $x^*_{r’}$ corresponds to the optimum when $r’$ is used as the regularization parameter, which can be a bit confusing.

* Introduction of Assumptions 3.1 and 4.1 seems abrupt, and some discussion on the assumption (e.g. their implications and how these assumptions compare with prior work) seems missing.

**Questions:**

* What kind of modifications to the proposed algorithmic framework or the assumptions on the objectives will allow this framework (or similar framework) being applied to the non-convex setting?

* Intuitively, why it is necessary to consider the constrained optimization setting for minimax optimization when obtaining these results given using the proposed framework?

**Limitations:**

This work only considers convex (convex-concave) setting, and some convergence results contain logarithmic factors, which are recognized by the authors in the conclusion. In addition to these, this work considers a constrained minimax optimization setting, which might limit the applicability of the corresponding results in practical applications.

---

> ### Author Rebuttal · Authors · 2023-08-09
>
> We thank the reviewer for the questions. Here are some clarifications.
>
> 1. **The convexity assumption on the objectives.**
>
> > The paper only covers convex objectives. What kind of modifications to the proposed algorithmic framework or the assumptions on the objectives will allow the results to be applied to the nonconvex setting?
>
> We focus on the convex case as a first step since it is the **most basic and fundamental** setting in optimization. We believe a solid understanding of the reproducibility in convex optimization will also shed insights for that of the more challenging nonconvex optimization. Note that some of the analysis and techniques used in this paper can be extended to the smooth nonconvex setting. For example, to track the difference between iterates along the trajectory of GD/GDA, we would obtain $\Vert z\_{t+1} - z\_{t+1}'\Vert \leq (1+\alpha\ell)\Vert z\_t - z\_t'\Vert$ without assuming convexity. The error can still be controlled when the stepsize $\alpha$ is small enough. For the proposed framework, extension to nonconvex functions is also possible. See [All18] and [Yan+20] for the convergence analysis of regularization/proximal point-based methods for nonconvex functions in the minimization and minimax settings respectively. However, for the reproducibility analysis, the non-expansiveness property, e.g., Lemma 3.2, will not hold any more without the convexity assumption. One potential way to alleviate it is to assume the negative comonotonicity [Gor+23] of the gradients. We leave a detailed study of the nonconvex setting to future work.
>
> 2. **The constrained optimization setting.**
>
> > The paper considers a constrained optimization setting for minimax optimization. Why is it necessary? Also, the dependence of the convergence bounds on the diameter of the constrained set makes the corresponding bounds too loose.
>
> The assumption that the domain $\mathcal{X}$ and $\mathcal{Y}$ are convex and compact for the minimax problems is to ensure the **existence of the saddle point**, as suggested by von Neumann's minimax theorem. The dependence on the diameters in the convergence bounds of convex-concave minimax problems is actually **inevitable** according to the lower bound in [OX21]. However, the reproducibility of the proposed framework under the inexact gradient setting crucially depends on the convergence guarantees of the sub-problems to the optimal solutions, which will introduce diameter dependence. It would be interesting to see whether such dependence in reproducibility can be relaxed.
>
> 3. **Other questions.**
>
> * **Notation.** Thanks for the suggestion. We will change the notation to $(x_r^*)'$ to avoid confusion.
>
> * **Assumptions 3.1 and 4.1.**  The assumptions (convexity, smoothness, and bounded initial solution/ bounded domain)  are **standard** in the convex optimization literature. We will add more discussions to introduce them.
>
> * **The logarithmic factors.** The logarithmic factor comes from the complexity of inexactly solving the strongly-convex sub-problems and is common for inexact proximal point type methods. It could be possible to remove such term by directly unwrapping the proposed framework to obtain a single-loop algorithm, with a much more involved and less intuitive analysis.
>
> **References**
>
> * **[All18]** Zeyuan Allen-Zhu. How to make the gradients small stochastically:
> Even faster convex and nonconvex SGD. Advances in Neural In-
> formation Processing Systems, 2018.
>
> * **[Yan+20]** Junchi Yang, Siqi Zhang, Negar Kiyavash, and Niao He. A catalyst framework for minimax optimization. Advances in Neural
> Information Processing Systems, 2020.
>
> * **[Gor+23]** Eduard Gorbunov, Adrien Taylor, Samuel Horváth, and Gauthier Gidel. Convergence of proximal point and extragradient-based methods beyond monotonicity: the case of negative comonotonicity. International Conference on Machine Learning, 2023.
>
> * **[OX21]** Yuyuan Ouyang and Yangyang Xu. Lower complexity bounds of first-order methods for convex-concave bilinear saddle-point problems. Mathematical Programming, 2021.

---

> > ### Comment · Reviewer_MNe7 · 2023-08-13
> > **Rebuttal acknowledgment**
> >
> > I thank the authors for the clarifications. In light of satisfactory clarifications, I raise my score.

---

### Decision · Program_Chairs · 2023-09-21

**Decision:**

Accept (spotlight)

**Comment:**

All reviewers are in unanimous agreement that this paper holds strong technical strength, leading them to recommend its acceptance. Given the compelling endorsement from the reviewers, I recommend accepting it as a spotlight paper.

In addition, I would like to encourage the authors to enhance the presentation of the paper by thoroughly incorporating the reviewers' suggestions and feedback.